# Reproducibility in Optimization: Theoretical Framework and Limits

**Kwangjun Ahn**[*]
MIT EECS
kjahn@mit.edu

**Prateek Jain**
Google Research
prajain@google.com

**Ziwei Ji**
Google Research
ziweiji@google.com

**Satyen Kale**
Google Research
satyenkale@google.com

**Praneeth Netrapalli**
Google Research
pnetrapalli@google.com

**Gil I. Shamir**
Google Research, Brain Team
gshamir@google.com

## Abstract

We initiate a formal study of reproducibility in optimization. We define a quantitative measure of reproducibility of optimization procedures in the face of noisy or error-prone operations such as inexact or stochastic gradient computations or inexact initialization. We then analyze several convex optimization settings of interest such as smooth, non-smooth, and strongly-convex objective functions and establish tight bounds on the limits of reproducibility in each setting. Our analysis reveals a fundamental trade-off between computation and reproducibility: more computation is necessary (and sufficient) for better reproducibility.

## 1 Introduction

Machine learned models are increasingly entering wider ranges of domains in our lives, driving a constantly increasing number of important systems. Large scale systems can be trained in highly parallel and distributed training environments, with a large amount of randomness in training the models. While some systems may tolerate such randomness leading to models that differ from one another every time a model retrains, for many applications, *reproducible* models are required, where slight changes in training do not lead to drastic differences in the model learned.

Beyond practical deployments of machine learned models, the reproducibility crisis in the machine learning academic world has also been well-documented: see [Pineau et al., 2021] and the references therein for an excellent discussion of the reasons for irreproducibility (insufficient exploration of hyperparameters and experimental setups, lack of sufficient documentation, inaccessible code, and different computational hardware) and for mitigation recommendations. Recent papers [Chen et al., 2020, D'Amour et al., 2020, Dusenberry et al., 2020, Snapp and Shamir, 2021, Summers and Dinneen, 2021, Yu et al., 2021] have also demonstrated that even when models are trained on identical datasets with identical optimization algorithms, architectures, and hyperparameters, they can produce significantly different predictions on the same example. This type of irreproducibility may be caused by multiple factors [D'Amour et al., 2020, Fort et al., 2020, Frankle et al., 2020, Shallue et al., 2018, Snapp and Shamir, 2021, Summers and Dinneen, 2021], such as non-convexity of the objective, random initialization, nondeterminism in training such as data shuffling, parallelism, random schedules, hardware used, and round off quantization errors. Perhaps surprisingly, even if we control for the randomness by using the same "seed" for model initialization, other factors such as numerical errors introduced due to nondeterminism of modern GPUs (see, e.g., [Zhuang et al., 2021]) may still lead to significant differences. It was empirically shown (see, e.g., Achille et al. [2017]) that

---

[*]Part of this work was done when Kwangjun Ahn was an intern at Google Research.

slight deviations early in training can lead to different optima, with substantial differences in resulting models. Thus we are forced to accept some *fundamental* level of irreproducibility that persists even after fixing all aspects of the training process under our control.

The goal of this paper is to initiate a formal study of the fundamental limits of irreproducibility. In particular, we focus on the most basic training process: *convex optimization*. At first glance, it might seem surprising that convex optimization procedures can exhibit irreproducibility since they're guaranteed to converge to an optimal solution. However, in practice, the default convex optimization algorithms are iterative first-order methods; methods that only use a first-order oracle to provide an approximate gradient of the function at a given point, and converge to an approximately optimal solution. The first-order oracle is a source of irreproducibility. In stochastic gradient descent, it returns a random vector whose expectation is the true gradient. The randomness in the stochastic gradients can lead to different outcomes of the optimization process. Similarly, there are numerical errors that can arise in the computation of the gradients due to inherent nondetermism in modern GPUs. Beyond the first-order oracle, irreproducibility may also arise in convex optimization procedures because the initial point is chosen randomly. Thus, we attempt to answer the following questions for convex optimization procedures operating with the above sources of irreproducibility:

- What are the fundamental limits of reproducibility for any convex optimization procedure?
- Can we design practical and efficient first-order methods that achieve these limits?

We study these questions in a variety of settings; including general non-smooth convex functions, smooth convex functions, strongly-convex functions, finite-sum functions, and stochastic convex optimization, under the different sources of irreproducibility mentioned above. To the best of our knowledge, no prior theoretical work considered such questions.

The primary contribution of this paper is conceptual: the development of a rigorous theoretical framework to study the fundamental limits of reproducibility in convex optimization. The concepts developed in this framework can be extended easily to other settings of interest such as non-convex optimization. The technical contribution of this paper is the development of lower bounds on the amount of reproducibility, and matching upper bounds via analysis of specific first-order algorithms in all the different settings of convex optimization described above. Detailed technical descriptions of the results appear below in Subsection 1.1.

At a high level, our study provides the same message for all the different optimization settings we consider. On the lower bound side, we find that *any* first-order method would need to trade-off convergence rate (computational complexity) for more reproduciblity. On the upper bound side, we find that various forms of gradient descent, when run with *lower* step-size (and correspondingly, *more* iterations) already achieve the fundamental limits of reproducibility. One (somewhat surprising) consequence of our results is that advanced techniques like regularization, variance reduction, and acceleration do not improve reproducibility over standard gradient descent methods.

We show, for example, that when optimizing a Lipschitz non-smooth convex function $f$ on a bounded domain using a first-order oracle that computes gradients with even *vanishingly small* error, two runs of *any* first-order method that obtains an $\varepsilon$ suboptimal solution of $f$ after $T$ iterations can generate solutions that are $\Omega(1/(\sqrt{T}\varepsilon))$ apart in $\ell_2$ distance. Thus, if we run the method for the standard $T = O(1/\varepsilon^2)$ iterations required to obtain $\varepsilon$-approximate solution for a non-smooth function, then obtained solutions can deviate by $\Omega(1)$ distance; i.e., the method is *maximally* irreproducible. To ensure that irreproducibility is small, i.e. that the solutions are within a small distance $\gamma$ of each other, we will need to run at least $\Omega(1/(\varepsilon^2\gamma^2))$ iterations. Interestingly, standard gradient descent with appropriately chosen learning rate and number of iterations already achieves this trade-off.

Our results demonstrate the challenge of reproducibility even for standard convex optimization. While we provide matching lower and upper bounds in certain general settings, in Section 7, we outline several important open directions. Solutions to these problems should enable better understanding of reproducibility even for deep learning.

## 1.1 Summary of results

Table 1 summarizes our key results for our measure of irreproducibility, $(\varepsilon, \delta)$-deviation (see Definition 3). The $(\varepsilon, \delta)$-deviation measures the amount of change between the outputs of two independent runs of an optimization algorithm, that is guaranteed to achieve $\varepsilon$-suboptimality after $T$ iterations,

Table 1: Summary of $(\varepsilon, \delta)$-deviation bounds for various convex optimization settings.

| | Stochastic Inexact Gradient Oracle **Theorem 1** | Non-stochastic Inexact Gradient Oracle **Theorem 2** | Inexact Initialization Oracle **Theorem 3** |
|---|---|---|---|
| Smooth | $\Theta(\delta^2/(T\varepsilon^2))$ | $\Theta(\delta^2/\varepsilon^2)$ | $\Theta(\delta^2)$ |
| Smooth Strongly-Cvx. | $\Theta(\delta^2/T \wedge \varepsilon)$ | $\Theta(\delta^2 \wedge \varepsilon)$ | $\Theta(e^{-\Omega(T)}\delta^2 \wedge \varepsilon)$ |
| Nonsmooth | $\Theta(1/(T\varepsilon^2))$ | $\Theta(1/(T\varepsilon^2) + \delta^2/\varepsilon^2)$ | $\Theta(1/(T\varepsilon^2) + \delta^2)$ |
| Nonsmooth Strongly-Cvx. | $\Theta(1/T \wedge \varepsilon)$ | $\Theta((1/T + \delta^2) \wedge \varepsilon)$ | $\Theta(1/T \wedge \varepsilon)$ |

when the computations of the algorithm incur errors of magnitude up to $\delta$. We specifically focus on three different sources of errors: i) stochastic gradient oracles, ii) gradient oracles with non-deterministic numerical errors (Definition 2), and iii) inexact initialization for the optimizer (Definition 1). We analyze the deviation under these sources of errors for four types of function classes: smooth convex functions, non-smooth but Lipschitz convex functions and strongly-convex restrictions of the two. Throughout the paper, $a \wedge b$ denotes the minimum between $a$ and $b$.

All lower bounds are for first-order iterative algorithms (*à la* Nesterov [2018]) that we formally define in (FOI). This is a large class of iterative optimization methods, including Stochastic Gradient Descent (SGD), which construct successive iterates *adaptively* in the linear span of previous iterates. Additionally, for smooth costs and stochastic inexact gradient oracle, we have an information theoretic lower bound of $\Omega(\delta^2/(T\varepsilon^2))$ when $\varepsilon \lesssim \delta^2$ (Theorem 6). We believe such informtation-theoretic lower bounds can be shown for all the settings in this paper. As for the upper bounds, they are all obtained using *slowed-down* SGD: i.e. SGD using smaller learning rates and more iterations.

For the non-strongly convex cases, one may expect to have high irreproducibility if the minima form a large flat region; however, surprisingly, our upper bounds show that we can always bound the extent of irreproducibility via slowed-down SGD. In the non-smooth cases, the main observation is that the deviation *does not* depend on scale of perturbation by the gradient oracle, i.e., any $\delta > 0$ can lead to fairly irreproducible solutions. The non-stochastic gradient oracle setting is strictly harder than the stochastic setting. Naturally, the lower and upper bounds on reproducibility are worse. Interestingly, even though strong convexity implies uniqueness of the global optimum, which intuitively should lead to highly reproducible solutions, we show that when faced with sources of error in computations, the deviation can still be significantly large for any algorithm.

Finally, we study reproducibility of optimization in machine learning settings. Here we have additional structure such as finite-sum minimization (for optimizing training loss) and stochastic convex optimization (for optimizing population loss). We define appropriate notions of errors for these problems and analyze two settings of particular interest. Our main results (Theorem 4 and Theorem 5) show that despite the additional structure in these problems, the bounds given by Table 1 for the specific settings are nonetheless *tight*. One consequence is that more sophisticated techniques for these problems such as variance reduction don't improve reproducibility.

## 1.2   Related work

**Related notions.**   In the scientific world, the terms *reproducibility* and *replicability* are often used interchangeably, but here we distinguish the two, following Pineau et al. [2021]. Reproducibility refers to the requirement that results obtained by a computational procedure (e.g. an experiment or a statistical analysis of a data set) should be the same (or largely similar) when the procedure is repeated using the same code on the same data, whereas replicability is a different notion that requires that results be reliably the same or similar when the data are changed. The field of statistical hypothesis testing [Lehmann and Romano, 2005] provides rigorous and principled techniques to minimize false discoveries and thereby promote replicability. The notion of *algorithmic stability* can also be seen as quantifying the amount of change in the output when a single data sample is changed. This notion has been extensively studied in the context of providing algorithm-dependent generalization bounds [Bousquet and Elisseeff, 2001, Kutin and Niyogi, 2002] and in developing differentially private algorithms [Dwork et al., 2006, McSherry and Talwar, 2007]. In very recent concurrent work, Impagliazzo et al. [2022] define a notion of *replicability* in learning that is quite different from ours: they aim to develop algorithms that generate the *exact same output* with reasonable probability given

a fresh sample. Note that despite the title of their paper, technically the notion studied is replicability, not reproducibility, since they study the output of algorithms when the input data are changed.

In this paper, we specifically focus on reproducibility: how much can the results of a computation differ when it is re-run on the same data with the same code? Hence, both hypothesis testing and algorithmic stability are orthogonal to the study in this paper, although some of our upper bounds use similar analysis techniques as algorithmic stability. On a different note, similar to replicability, the boundary between the notions of irreproducibility and *uncertainty* in deep models is rather blurred. Several papers considered different aspects of uncertainty (see, e.g., [Lakshminarayanan et al., 2017] and references therein), but this line of work has been empirical in nature.

**Inexact oracles in optimization.**    The optimization community has studied the consequences of using inexact or error-prone gradient oracles in optimization. Several papers (e.g. [Aybat et al., 2020, Devolder et al., 2014, d'Aspremont, 2008, Cohen et al., 2018] have developed bounds on the optimization error incurred due to the use of inexact oracles. While the sources of errors are similar to the ones studied in this paper, the quantities of interest in these papers are convergence rate and optimization error rather than reproducibility. Interestingly, despite the different objective, some of the high-level conclusions are similar to our paper: for example, accelerated gradient methods do not outperform standard classical methods when used with inexact gradient oracles.

**Techniques to improve reproducibility in practice.**    Several recent empirical papers considered methods that can reduce levels of irreproducibility in deep models despite nondeterminism in training. Smooth activations [Du, 2019, Mhaskar, 1997] have been shown [Shamir et al., 2020] to improve reproducibilty over popular activations, as the Rectified Linear Unit [Nair and Hinton, 2010]. Ensembles [Dietterich, 2000] leverage diversity of multiple different solutions to produce an average more reproducible one [Allen-Zhu and Li, 2020]. Co-distillation [Anil et al., 2018] and Anti-distillation [Shamir and Coviello, 2020] leverage ensembles to further push deployed models to be more reproducible. Imposing constraints [Bhojanapalli et al., 2021, Shamir, 2018] forces models to prefer some solutions over others, but may come at the cost of reducing model accuracy performance.

**Robustness of dynamical systems.**    The upper bound results in our paper can be interpreted as robustness results of the (sub)gradient descent dynamics against disturbances. In particular, our upper bounds can be viewed as some variants of the input-to-state stability [Sontag and Wang, 1995] results for the dynamics (see, e.g., [Tu et al., 2022, Definition 3.2]).

## 2    Problem Formulation

In this section, we define a quantitative measure of *irreproducibility* amenable to a theoretical analysis. Intuitively, a computation is *reproducible* if it generates the exact same output given the same inputs on two different runs. Irreproducibility arises because low-level operations of a computation produce different answers on two runs due to either *randomness* or *non-determinism*.

Our computation of interest is convex optimization via first-order methods, where *initialization* and *gradient computations* are the primary operations that constitute the computation and are subject to errors leading to inexact outputs. A natural measure of irreproducibility is the amount of change in the computed solution to the convex optimization problem under inexact gradient computations or inexact initialization. However, there are two nuances that must be carefully handled here. First, a trivial procedure which ignores its input and outputs a constant solution is perfectly reproducible! Unfortunately, it is perfectly useless as a convex optimization procedure as well. Thus, in order to compare different procedures by their reproducibility metrics, we must assume that the procedures are guaranteed to converge to an optimal solution. The second nuance is that we need to assume that the errors in the gradient or initialization computations are *bounded* in some manner. Evidently, without such an assumption, any non-trivial convex optimization procedure will be extremely irreproducible. We now use the above considerations to develop a precise definition of a measure of irreproducibility.

**Convex function classes.**    We assume that the function to be optimized is chosen from a certain class, $\mathcal{F}$, of convex functions, along with their domains, satisfying suitable *regularity* conditions (e.g. Lipschitzness, smoothness, strong-convexity, etc.) to develop convergence rates. For clarity, we will suppress exact dependence on smoothness, Lipschitzness, and strong-convexity parameters. In

particular, "smooth" will denote a convex function whose gradients are $O(1)$-Lipschitz continuous, "non-smooth" a convex function that is $O(1)$-Lipschitz continuous, and "strongly-convex" an $\Omega(1)$-strongly-convex function. Here, $O(1)$ and $\Omega(1)$ denote universal constants independent of the dimension or other problem dependent quantities, which we leave unspecified to ease the exposition.

**Convex optimization procedures.** A *first-order* convex optimization procedure for $\mathcal{F}$ is an algorithm that, given any function $f \in \mathcal{F}$, and access to two (potentially noisy) oracles – an initialization oracle, which generates the initial point, and a gradient oracle, which computes gradients for $f$ at any given query point – generates a candidate solution $\boldsymbol{x}_{\text{out}}$ for the problem of minimizing $f$ over its domain. Note that the algorithm can only access $f$ via the oracles provided. We call such an algorithm $\varepsilon$-*accurate* if it guarantees that $\mathbb{E} f(\boldsymbol{x}_{\text{out}}) - \inf_{\boldsymbol{x} \in \text{dom} f} f(\boldsymbol{x}) \leq \varepsilon$, where the expectation is over any randomness in the computation of $\boldsymbol{x}_{\text{out}}$. Several of our lower bounds require more structure for the algorithm: specifically, a *first-order iterative* (FOI) algorithm (*à la* Nesterov [2018]) is one that starting from the point $\boldsymbol{x}_0$ generated by the initialization oracle, constructs successive iterates

$$\boldsymbol{x}_t = \boldsymbol{x}_0 - \sum_{i=0}^{t-1} \lambda_i^{(t)} g(\boldsymbol{x}_i) \quad \text{for some } \lambda_i^{(t)}, \; i = 0, \ldots, t-1, \tag{FOI}$$

where $g(\boldsymbol{x}_i)$ is the output of the gradient oracle query at $\boldsymbol{x}_i$, and outputs $\boldsymbol{x}_T$ for some integer $T > 0$. We emphasize that for all $t$, the coefficients $\lambda_i^{(t)}$ can be chosen adaptively based on all the previous computations. The above class of algorithms is a canonical one to consider when proving lower bounds against gradient oracle based optimization algorithms. We refer readers to [Nesterov, 2018, §2.1.2] for more background. For the case of nonsmooth costs, we additionally assume that the coefficient of the latest gradient is nonzero, i.e., $\lambda_{t-1}^{(t)} \neq 0$ for all $t$. We also note that one of our lower bound results (Theorem 6) is *information-theoretic* (in the sense of Nemirovski and Yudin [1983]).

**Sources of errors in computation.** Errors arise due to inexactness in the outputs of the initialization or gradient oracles. Queries to these oracles on two different runs of the same algorithm might yield different outputs, but we will control the errors by assuming that the outputs are close to some reference point (that remains fixed over different runs) in a suitable metric.

**Definition 1** ($\delta$-bounded inexact initialization oracle)**.** *Given a function $f \in \mathcal{F}$ and a reference initialization point $\boldsymbol{x}_0^{ref} \in \text{dom} f$, a $\delta$-**bounded** inexact initialization oracle for $f$ is one that generates an initial point $\boldsymbol{x}_0 \in \text{dom} f$ such that $\|\boldsymbol{x}_0 - \boldsymbol{x}_0^{ref}\| \leq \delta$.*

The gradient computation oracle is said to be $\delta$-**bounded** if for any $f \in \mathcal{F}$ and any point $\boldsymbol{x} \in \text{dom} f$, it outputs a vector $g(\boldsymbol{x})$ such that $\mathbb{E} \|g(\boldsymbol{x}) - \nabla f(\boldsymbol{x})\|^2 \leq \delta^2$ for some $\nabla f(\boldsymbol{x}) \in \partial f(\boldsymbol{x})$, where the expectation is over any randomness in the computation of $g(\boldsymbol{x})$. We consider both *stochastic* and *non-stochastic* inexact $\delta$-bounded gradient oracles. A stochastic gradient oracle has the additional property that its output $g(\boldsymbol{x})$ is a random vector such that $\mathbb{E} g(\boldsymbol{x}) = \nabla f(\boldsymbol{x})$, with different queries being independent of each other. Stochastic inexact gradient oracles arise naturally in machine learning applications due to randomness in minibatching. Non-stochastic inexact gradient oracles model non-deterministic numerical errors due to the accumulation of floating point errors; for giant machine learning models with billions of parameters, individual floating point errors could add up to a noticeable large error. We formally define the two types of inexact oracles below.

**Definition 2** ($\delta$-bounded inexact gradient oracle)**.** *Given a function $f \in \mathcal{F}$, and $\boldsymbol{x} \in \text{dom} f$, let $\partial f(\boldsymbol{x})$ denote the sub-differential of $f$ at $\boldsymbol{x}$.*

*(a) A* **stochastic inexact** *$\delta$-bounded gradient oracle outputs a random vector $g(\boldsymbol{x})$ such that $\mathbb{E} g(\boldsymbol{x}) = \nabla f(\boldsymbol{x})$ and $\mathbb{E} \|g(\boldsymbol{x}) - \nabla f(\boldsymbol{x})\|^2 \leq \delta^2$ for some $\nabla f(\boldsymbol{x}) \in \partial f(\boldsymbol{x})$. The expectation is over the randomness in $g(\boldsymbol{x})$ which is also assumed to be independent for each oracle call.*

*(b) A* **non-stochastic inexact** *$\delta$-bounded gradient oracle outputs a non-deterministic vector $g(\boldsymbol{x})$ such that $\|g(\boldsymbol{x}) - \nabla f(\boldsymbol{x})\|^2 \leq \delta^2$ for some $\nabla f(\boldsymbol{x}) \in \partial f(\boldsymbol{x})$.*

**Measure of irreproducibility.** Let $\mathcal{A}$ be a first-order, $\varepsilon$-accurate convex optimization procedure for $\mathcal{F}$ with access to either a $\delta$-bounded initialization oracle (and an exact gradient oracle), or a $\delta$-bounded gradient oracle (and an exact initialization oracle). The $(\varepsilon, \delta)$-**deviation** is

**Definition 3** ($(\varepsilon, \delta)$-deviation)**.** *Given a function $f \in \mathcal{F}$, let $\boldsymbol{x}_f$ and $\boldsymbol{x}_f'$ denote the outputs of $\mathcal{A}$ on two independent runs of $\mathcal{A}$ that result in $\varepsilon$-accurate solutions.*

*(a) If a stochastic inexact $\delta$-bounded gradient oracle is used, the $(\varepsilon, \delta)$-deviation of $\mathcal{A}$ is defined as $\sup_{f \in \mathcal{F}} \mathbb{E} \|\boldsymbol{x}_f - \boldsymbol{x}_f'\|^2$ where the randomness is over the stochastic oracle in the two runs.*

*(b) If either an inexact $\delta$-bounded initialization oracle or a $\delta$-bounded non-stochastic inexact gradient oracle is used, the $(\varepsilon, \delta)$-deviation of $\mathcal{A}$ is defined as $\sup_{f \in \mathcal{F}} \sup \|\boldsymbol{x}_f - \boldsymbol{x}'_f\|^2$, where the inner supremum is over the two runs.*

Here we note that $\boldsymbol{x}_f$ is not necessarily the last iterate of a first-order algorithm. Depending on cases, sometimes we choose $\boldsymbol{x}_f$ to be the (weighted) average iterate. We consider $\|x_f - x'_f\|^2$ for the notion of deviation instead of $\|x_f - x^*\|^2$; note that $x^*$ may not even be unique without strong convexity.

**Other notions?** Alternate definitions are certainly plausible. For example, in machine learning applications, the computed solution $\boldsymbol{x}$ may be used as a parameter vector for a predictor function $g_{\boldsymbol{x}}$ which maps inputs $z$ to real-valued predictions $\hat{y}$ whose quality is measured by a loss function $\ell(\hat{y}, y)$ where $y$ is the true label of $z$. Let $\boldsymbol{x}$ and $\boldsymbol{x}'$ be outputs of two different runs of the optimization algorithm. Then one can also define $(\varepsilon, \delta)$-deviation based on *prediction reproducibility*: e.g. $\sup_z |g_{\boldsymbol{x}}(z) - g_{\boldsymbol{x}'}(z)|$ or $\mathbb{E}_z |g_{\boldsymbol{x}}(z) - g_{\boldsymbol{x}'}(z)|$, or *loss reproducibility*: e.g. $\sup_{(z,y)} |\ell(g_{\boldsymbol{x}}(z), y) - \ell(g_{\boldsymbol{x}'}(z), y)|$ or $\mathbb{E}_{(z,y)} |\ell(g_{\boldsymbol{x}}(z), y) - \ell(g_{\boldsymbol{x}'}(z), y)|$, where the expectation is over the distribution of the examples. Nevertheless, we adopt Definition 3, which is based on *parameter reproducibility*, in this paper for the following reasons:

- The convex optimization problems studied in this paper are **more basic/fundamental** than more structured ML optimization. To the best of our knowledge, there is no pre-existing theory even for this basic setting. Parameter reproducibility is a more natural definition here since there is no notion of prediction or loss.

- In many ML applications, the predictor function $g_{\boldsymbol{x}}$ is Lipschitz in $\boldsymbol{x}$ for any input $z$. In such cases, $(\varepsilon, \delta)$-deviation bounds for parameter reproducibility immediately transform into $(\varepsilon, \delta)$-deviation bounds for prediction reproducibility. Similar transformations are also generally possible $(\varepsilon, \delta)$-deviation bounds for loss reproducibility.

- Without knowledge of how a learned parametric function is deployed for making predictions, it is difficult to analyze prediction reproducibility even in ML settings. Hence, parameter deviations provide a reasonable first approximation. Furthermore, in real systems, we often optimize multiple metrics (e.g. performance on sub-segments of populations, for fairness). Parameter reproducibility gives greater assurance on all metrics. A similar argument applies when the test distribution is different from the training distribution.

- Finally, several recent works [Shamir et al., 2020, D'Amour et al., 2020] have empirically observed that even if two different runs of the algorithm resulted in parameters that have nearly the same loss, the predictions on test examples could be very different. One reason this happens is because *surrogate* losses are used in place of the true metric for optimization. So simply achieving loss reproducibility may not be sufficient for practical applications.

## 3 Reproducibility with Stochastic Inexact Gradient Oracles

In this section we consider reproducibility of optimizing a convex function $f \in \mathcal{F}$ where we can access $f$ only via a stochastic gradient oracle (see Definition 2). This setting covers several important ML optimization scenarios, e.g., when the training data is randomly sampled from a population or when the selection of mini-batches is randomized. Our main result is the following theorem.

**Theorem 1.** *For any $\varepsilon, \delta > 0$, and number of iterations $T$, the $(\varepsilon, \delta)$-deviation for optimizing convex functions with a stochastic inexact gradient oracle is as follows. Unless indicated otherwise, the lower bounds hold for any FOI algorithm, and the upper bound is achieved by stochastic gradient descent for $T = \Omega(1/\varepsilon^2)$ in the non-strongly-convex settings and $T = \Omega(1/\varepsilon)$ in the strongly-convex settings.*

- *Smooth functions: $(\varepsilon, \delta)$-deviation is $\Theta(\delta^2/(T\varepsilon^2))$. Furthermore, for the case $\varepsilon \lesssim \delta^2$ there is a matching* **information theoretic lower bound**.
- *Smooth and strongly convex functions: $(\varepsilon, \delta)$-deviation is $\Theta(\delta^2/T \wedge \varepsilon)$.*
- *Lipschitz (non-smooth) functions: $(\varepsilon, \delta)$-deviation is $\Theta(1/(T\varepsilon^2))$.*
- *Lipschitz (non-smooth) and strongly convex functions: $(\varepsilon, \delta)$-deviation is $\Theta(1/T \wedge \varepsilon)$.*

This theorem is proved in several pieces, one corresponding to each setting (i.e., smooth, smooth & strongly convex etc.) and type of bound (i.e., upper or lower). The precise details are given in Appendix A. A few remarks are in order.

**Smoothness**: Intuitively, smoothness would ensure that a slight error in gradient computation need not imply catastrophic deviation in the iterates. Our matching upper and lower bounds confirm this intuition.

**Non-smoothness**: Our results show that even a slight amount of noise in the gradients can lead to drastic irreproducibility in the non-smooth case. Intuitively, the reason for this phenomenon is that non-smooth functions have non-differentiable points where the slightest amount of noise in the gradients can lead to drastically different behavior. This is in line with the empirical observations of Shamir et al. [2020].

**Strong convexity**: Note that the deviation is smaller than other cases due to the existence of a unique minimizer; the $\varepsilon$-accuracy in the cost already implies an $O(\varepsilon)$ upper bound on the deviation.

Furthermore, in all the settings, our results show that using an algorithm with a larger number of iterations is more helpful; intuitively that is because more gradient samples from the oracle can help reduce the sample noise as an averaging. For example, for smooth functions, if a gradient descent type method is run for the standard $T = O(1/\varepsilon^2)$ iterations, then the solution obtained might still suffer deviation of $\delta^2$, which is independent of $\varepsilon$. Thus, in order to obtain low deviation, we are forced to run the method for $\omega(1/\varepsilon^2)$ iterations. We also note that by relying on the standard convergence rate lower bounds [Nemirovski and Yudin, 1983], the requirement on $T$ e.g., $T = \Omega(1/\varepsilon^2)$ for the non-strongly convex setting, is without loss of generality. Finally, we remark that the lower bound for smooth functions is information theoretic (i.e., holds against *any* algorithm) for $\varepsilon \lesssim \delta^2$.

## 4 Reproducibility with Non-Stochastic Inexact Gradient Oracles

In this section, we study reproducibility for optimizing a function $f \in \mathcal{F}$ with non-stochastic inexact gradient oracle access (Definition 2). In particular, we establish lower and upper bounds for first-order algorithms (FOI) on the $(\varepsilon, \delta)$-deviation (Definition 3) despite initialization with the same point $x_0 \in \text{dom} f$. Recall that this setting allows us to capture reproducibility challenges due to non-deterministic numerical errors introduced by a computing device (like GPUs) during floating point computations. The main high-level message here is that unlike in the stochastic gradient oracle case, in the non-stochastic gradient oracle setting the iteration complexity $T$ has little to no effect on the $(\varepsilon, \delta)$-deviation: intuitively this is because unlike the stochastic setting, it is not possible to reduce the error in the gradients by taking more samples and averaging.

**Theorem 2.** *For $D = O(1)$, $\varepsilon, \delta > 0$ such that $\delta \leq \varepsilon/(2D)$, and number of iterations $T$, the $(\varepsilon, \delta)$-deviation for optimizing convex functions whose optimum has norm at most $D$ with a non-stochastic inexact gradient oracle is as follows. Unless indicated otherwise, the lower bounds hold for any FOI algorithm, and the upper bound is achieved by projected gradient descent for $T = \Omega(1/\varepsilon)$ in the smooth settings and non-smooth strongly convex setting, and $T = \Omega(1/\varepsilon^2)$ in the non-smooth setting.*

- *Smooth functions: $(\varepsilon, \delta)$-deviation is $\Theta(\delta^2/\varepsilon^2)$.*[2]
- *Smooth and strongly convex functions: $(\varepsilon, \delta)$-deviation is $\Theta(\delta^2 \wedge \varepsilon)$.*
- *Lipschitz (non-smooth) functions: $(\varepsilon, \delta)$-deviation is $\Theta(1/(T\varepsilon^2) + \delta^2/\varepsilon^2)$.*
- *Lipschitz (non-smooth) and strongly convex functions: $(\varepsilon, \delta)$-deviation is $\Theta((1/T + \delta^2) \wedge \varepsilon)$.*

Like Theorem 1, this theorem is proved in several pieces; see Appendix A. Some remarks follow:

**Acceleration v.s. reproducibility?** Note that for smooth functions, accelerated methods can get $\varepsilon$-suboptimality in only $T = O(1/\sqrt{\varepsilon})$ iterations. A natural question is if we can achieve similar $(\varepsilon, \delta)$-deviation for such accelerated methods. However, it is known that accelerated methods are unstable (see e.g., [Devolder et al., 2014, Attia and Koren, 2021]), and cannot even achieve the desired $\varepsilon$-accuracy under the inexact oracle model. Hence, we conjecture that iteration complexity of $\Omega(1/\varepsilon)$ is necessary to achieve the desired reproducibility.

---

[2]Note that Theorem 21 provides a similar upper bound result without the assumption that the optimum lies in a ball of radius $O(1)$.

**Nonsmoothness:** Note that the lower bound shows that if gradient descent style methods are run for the standard $T = 1/\varepsilon^2$ iterations, then the solutions can be forced to have $\Omega(1)$ deviation. Hence, to ensure reproducibility, any method would need to have a slower convergence rate than $T = 1/\varepsilon^2$.

## 5 Reproducibility with Inexact Initialization Oracles

We now study reproducibility for optimizing a function $f \in \mathcal{F}$ with an inexact initialization oracle (Definition 1) and establish tight bounds on the $(\varepsilon, \delta)$-deviation for FOI algorithms. An inexact initialization oracle models situations where the initial values of the parameters are set randomly, or incur some non-deterministic numerical error.

**Theorem 3.** *For any $\varepsilon, \delta > 0$ and number of iterations $T$, the $(\varepsilon, \delta)$-deviation for optimizing convex functions with an inexact initialization oracle is as follows. In all cases, the lower bounds hold for any FOI algorithm, and the upper bound is achieved by gradient descent for $T = \Omega(1/\varepsilon)$ in the smooth settings and non-smooth strongly convex setting, and $T = \Omega(1/\varepsilon^2)$ in the non-smooth setting.*
- *■ **Smooth functions:** $(\varepsilon, \delta)$-deviation is $\Theta(\delta^2)$.*
- *■ **Smooth and strongly convex functions:** $(\varepsilon, \delta)$-deviation is $\Theta((\exp(-\Omega(T))\delta^2 \wedge \varepsilon)$.*
- *■ **Lipschitz (non-smooth) functions:** $(\varepsilon, \delta)$-deviation is $\Theta(1/(T\varepsilon^2) + \delta^2)$.*
- *■ **Lipschitz (non-smooth) and strongly convex functions:** $(\varepsilon, \delta)$-deviation is $\Theta(1/T \wedge \varepsilon)$.*

Like Theorem 1, this theorem is proved in several pieces; see Appendix A. A remark follows:

**Nonsmoothness:** As in the case of non-smooth optimization with inexact gradient oracles, here as well we see that even the slightest amount of inexactness in initialization can lead to non-negligible irreproducibility. Intuitively this is a consequence of non-smoothness: If the function is not differentiable at the reference point, even a slight bit of inexactness can lead to drastically different trajectories right from the beginning.

## 6 Reproducibility in Optimization for Machine Learning

So far, we have studied reproducibility in general convex optimization with different sources of perturbation. In machine learning, however, optimization problems come with more structure, and hence a more nuanced analysis of reproducibility is called for. In this section, we study reproducibility in two specific optimization settings of interest in machine learning: finite sum minimization which corresponds to minimizing training loss, and stochastic convex optimization (SCO), which corresponds to minimizing population (or test) loss. Instead of a detailed study of different types of convex functions as done in the previous sections, here we focus on a few specific cases that yield particularly interesting insights.

### 6.1 Optimizing Training Loss (Finite Sum Minimization)

Optimizing the training loss in ML can be cast as minimizing a function that can be written as a finite sum of component functions: $f(\boldsymbol{x}) := \frac{1}{m} \sum_{i=1}^{m} f_i(\boldsymbol{x})$. Typical optimization methods such as stochastic gradient descent can be implemented to solve such problems by iteratively sampling one of the component functions and taking its gradient. So randomness in sampling as well as non-deterministic numerical errors in computing of gradient can lead to irreproducibility. Randomness in sampling is a property of the algorithm and hence under the control of the algorithm designer; however non-deterministic numerical errors in gradient computations are beyond the control of the designer, and hence we aim to quantify irreproducibility caused by this specific source (numerical errors). To capture this intuition, we consider the following inexact gradient oracle.

**Definition 4** ($\delta$-bounded inexact **component** gradient oracle)**.** *A $\delta$-bounded inexact **component** gradient oracle takes as input $i \in \{1, 2, \ldots, m\}$ and a point $x$, and outputs a vector $g_i(\boldsymbol{x})$ such that*

$$\|g_i(\boldsymbol{x}) - \nabla f_i(\boldsymbol{x})\|^2 \leq \delta^2 \text{ for some } \nabla f_i(\boldsymbol{x}) \in \partial f_i(\boldsymbol{x}).$$

The number of calls to the component gradient oracle is now our primary measure of complexity. Then, a natural question that arises is whether we can we tightly characterize reproduciblity of typical first-order methods when given access to an inexact component gradient oracle. We first observe

that in the special case that all $f_i$ are identical, the inexact component gradient oracle reduces to the non-stochastic inexact oracle (Definition 2). Hence, for the smooth and non-smooth function settings, we immediately obtain lower bounds on the $(\varepsilon, \delta)$-deviation via our previous analysis; see Theorem 8 and Theorem 14, respectively.

The non-smooth case in particular is interesting, as the deviation lower bound is $\Omega(1/(T\varepsilon^2) + \delta^2/\varepsilon^2)$, where $T$ is the number of oracle calls. Now, if we use projected (full-batch) gradient descent (as used for the matching upper bound in Theorem 27), then each iteration would require $m$ oracle calls, hence the $(\varepsilon, \delta)$-deviation for such a method is $O(m/(T\varepsilon^2) + \delta^2/\varepsilon^2)$, which is worse than the lower bound mentioned above. Now, the key question is whether we can reduce this deviation by other means. It turns out that *stochastic* gradient descent (SGD), where in each step we sample a component function to query randomly, when run with an appropriate learning rate and using averaging, matches the optimal deviation up to constant factors. We provide a detailed proof of the result in Appendix G. This provides another justification for using SGD instead of gradient descent in practice.

**Theorem 4.** *For $G = O(1)$ and $D = O(1)$, let $f_i$ be an $G$-Lipschitz convex cost function for each $i \in [m]$, and assume that the optimum of $f$ lies in a ball of radius $D$. Let $\varepsilon, \delta > 0$ be given parameters such that $\delta \leq \varepsilon/(2D)$, and $T = \Omega(1/\varepsilon^2)$ be a given number of iterations. Define the SGD updates as follows: initialize $\boldsymbol{x}_0 = 0$, and for $t = 0, 1, \ldots, T-1$, set $\boldsymbol{x}_{t+1} = \boldsymbol{x}_t - \eta_t g_{i_t}(\boldsymbol{x}_t)$ where $i_t \sim [n]$ uniformly at random. Under the inexact component gradient oracle (Definition 4), the average iterate $\bar{\boldsymbol{x}}_T$ of SGD with stepsize $\eta = \Theta(1/(\varepsilon T))$ satisfies $\mathbb{E} f(\bar{\boldsymbol{x}}_T) - \inf_{\boldsymbol{x} \in dom f} f(\boldsymbol{x}) \leq \varepsilon$ and $\mathbb{E} \|\bar{\boldsymbol{x}}_T - \bar{\boldsymbol{x}}'_T\|^2 = O(1/(T\varepsilon^2) + \delta^2/\varepsilon^2)$, where $\bar{\boldsymbol{x}}'_T$ is the output of an independent run of SGD.*

## 6.2 Optimizing Population Loss (Stochastic Convex Optimization)

The fundamental problem of machine learning is to find the solution to the following *population* (or *test*) loss minimization problem: minimize $F(\boldsymbol{x}) = \mathbb{E}_{\xi \sim \Xi} f(\boldsymbol{x}, \xi)$, where $\Xi$ is an unknown distribution on the examples $\xi$, given access to an oracle that sample from $\Xi$. When the function $F$ is convex, this is also known as Stochastic Convex Optimization (SCO).

In this setting, the sampling oracle is a natural source of irreproducibility of optimization methods. To model this, we consider the *stochastic global oracle*, inspired by Foster et al. [2019].

**Definition 5** ($\delta$-bounded stochastic global oracle). *Given a function $F$, a $\delta$-bounded stochastic global oracle for $F$ is an algorithm that, on each query, draws an independent sample $\xi \sim \Xi$ from some distribution $\Xi$, and outputs a function $f(\cdot, \xi) : dom F \to \mathbb{R}$ such that for all $x \in dom F$, we have $F(\boldsymbol{x}) = \mathbb{E}_{\xi \sim \Xi} f(\boldsymbol{x}, \xi)$, and, $\mathbb{E}_{\xi \sim \Xi} \|\nabla f(\boldsymbol{x}, \xi) - \nabla F(\boldsymbol{x})\|^2 \leq \delta^2$ .*

Since in each query, the stochastic global oracle returns the complete specification of $f(\cdot, \xi)$, an algorithm using such an oracle has access to $(\boldsymbol{x}, f(\boldsymbol{x}, \xi), \nabla f(\boldsymbol{x}, \xi), \nabla^2 f(\boldsymbol{x}, \xi), \cdots)$ for all $\boldsymbol{x}$.

Note that the inexact stochastic gradient oracle of Definition 2 is a weaker form of the stochastic global oracle. Thus, the key question is: *under the powerful stochastic global oracle, can we provide better reproducibility than the lower bounds in Theorem 1?* Surprisingly, the answer to the above question is negative. That is, despite the stronger oracle setting, there is a function class for which the lower bound on $(\varepsilon, \delta)$-deviation matches that of Theorem 1.

**Theorem 5.** *Assume that $\varepsilon < 1/200$ and $\varepsilon \lesssim \delta^2$. Then there exists a family of population costs $F_\theta(\boldsymbol{x}) = \mathbb{E}_{\xi \sim P_\theta} f_\theta(\boldsymbol{x}, \xi)$ parametrized by $\theta \in [1, 2]$, and a $\delta$-bounded stochastic global oracle for $F_\theta$, satisfying the following property. Suppose that $A$ is any algorithm that for each $\theta \in [1, 2]$ uses at most $T$ queries to the stochastic global oracle and outputs $\boldsymbol{x}_T^\theta$ that is $\varepsilon$-accurate, i.e., $\mathbb{E} F_\theta(\boldsymbol{x}_T^\theta) - \inf_{\boldsymbol{x}} F_\theta(\boldsymbol{x}) \leq \varepsilon$. Then, there exists $\theta_{bad} \in [1, 2]$ such that $\mathrm{Var}(\boldsymbol{x}_T^{\theta_{bad}}) \geq \Omega(\delta^2/(T\varepsilon^2))$.*

See Subsection B.2 for a detailed proof. On the other hand, Theorem 1 also shows that standard SGD with the weaker inexact stochastic gradient oracle achieves this lower bound.

# 7 Conclusions

We presented a framework for studying reproducibility in convex optimization, and explored limits of reproducibility for optimizing various function classes under different sources of errors. For each of these settings, we provide tight lower and upper bounds on reproducibility of first order iterative algorithms. Overall, our results provide the following insights: a) Non-smooth functions

can be highly susceptible to even tiny errors which can creep in due to say numerical errors in GPU. Thus introducing smoothness in deep learning models might help with reproducibility. b) Generally, gradient descent type methods with small learning rate are more reproducible. c) In finite-sum settings, despite more randomness, SGD is more reproducible than full-batch gradient descent.

The study in this paper is a first step towards addressing the challenging problem of reproducibility in a rigorous framework. So, many important directions remain unexplored. For example, study of reproducibility of adaptive methods like AdaGrad that are not captured by (FOI) is interesting. We study reproducibility in a strict form where we measure deviation in terms of the learned parameters of the model. However, in practice, one might care for reproducibility in model predictions only. So, extending our results to such ML-driven scenarios should be relevant in practice. Finally, while our lower bounds on reproducibility obviously also hold for non-convex optimization, extension of the upper bounds to non-convex settings and potentially, designing novel rigorous methods in this setting is a fascinating direction for future work.

## Acknowledgments and Disclosure of Funding

Kwangjun Ahn was supported by graduate assistantship from the NSF Grant (CAREER: 1846088), the ONR grant (N00014-20-1-2394) and MIT-IBM Watson as well as a Vannevar Bush fellowship from Office of the Secretary of Defense. Part of this work was done while Kwangjun Ahn was visiting the Simons Institute for the Theory of Computing, Berkeley, California.

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
