# Appendix

## Table of Contents

# A  Summary of results in Appendix

In order to help readers navigates the results in the appendix, we summarize the results in the appendix in the following table. In the main paper, we have three main theorems: Theorems 1, 2 and 3, each corresponding to one column in the following table. Further each of these theorems have four components corresponding to four settings: smooth, smooth & strongly convex, nonsmooth and finally nonsmooth & strongly convex. Each cell in the below table lists the corresponding theorems which prove the lower and upper bounds corresponding to a given setting. Theorems 1, 2 and 3 follow immediately from the constituent theorems. In each cell of results, "Theorems **X** ‖ **Y**" indicates that the lower bound appears in Theorem **X**, and the upper bound in Theorem **Y**. All lower bounds are for first-order iterative algorithms (*à la* Nesterov [2018]) that we formally defined in (FOI).

| | Stochastic Inexact Gradient Oracle (Theorem 1) | Non-stochastic Inexact Gradient Oracle (Theorem 2) | Inexact Initialization Oracle (Theorem 3) |
|---|---|---|---|
| Smooth | $\Theta(\delta^2/T\varepsilon^2)$ 
 Theorems 6$^\dagger$&7 ‖ 19 | $\Theta(\delta^2/\varepsilon^2)$ 
 Theorems 8 ‖ 20 | $\Theta(\delta^2)$ 
 Theorems 9 ‖ 22 |
| Smooth Strongly-Convex | $\Theta(\delta^2/T \wedge \varepsilon)$ 
 Theorems 10 ‖ 23 | $\Theta(\delta^2 \wedge \varepsilon)$ 
 Theorems 11 ‖ 24 | $\Theta(e^{-\Omega(T)}\delta^2 \wedge \varepsilon)$ 
 Theorems 12 ‖ 25 |
| Nonsmooth | $\Theta(1/T\varepsilon^2)$ 
 Theorems 13 ‖ 26 | $\Theta(1/T\varepsilon^2 + \delta^2/\varepsilon^2)$ 
 Theorems 14 ‖ 27 | $\Theta(1/T\varepsilon^2 + \delta^2)$ 
 Theorems 15 ‖ 28 |
| Nonsmooth Strongly-Convex | $\Theta(1/T \wedge \varepsilon)$ 
 Theorems 16 ‖ 29 | $\Theta((1/T + \delta^2) \wedge \varepsilon)$ 
 Theorems 17 ‖ 30 | $\Theta(1/T \wedge \varepsilon)$ 
 Theorems 18 ‖ 31 |

† For smooth costs and stochastic inexact gradient oracle, we also have an additional **information-theoretic** lower bound of $\Omega(\delta^2/T\varepsilon^2)$ when $\varepsilon \lesssim \delta^2$ in Theorem 6.

## A.1  General guidance for navigating Appendix

- ■ The **information-theoretic lower bounds** are presented in Appendix B and can be read independently:
  - • The lower bound proof for the smooth costs (Theorem 6) and is presented in Subsection B.1.
  - • The lower bound proof for the stochastic global oracle case (Theorem 5) is similar to that of Theorem 6 and is presented in Subsection B.2.
- ■ The **FOI lower bounds** are quite technical and often rely on delicate constructions. Hence, we present the proofs as follows:
  - • In Subsection C.1, to help readers understand the general proof strategy, we first present the lower bound proof for smooth costs with the stochastic inexact oracle (Theorem 7).
  - • In Appendix C, we present the proofs of other FOI lower bounds for smooth costs.
  - • In Appendix D, we present the proofs of FOI lower bounds for nonsmooth costs. The lower bound constructions are more complicated than the case of smooth costs. Hence, to help reader understand the crux of arguments, we first present the proof of a (weaker) lower bound against gradient descent (as opposed to the entire class of FOI) in Subsection D.1.
- ■ The **upper bounds** are presented in Appendix E (smooth costs), Appendix F (non-smooth costs), and Appendix G (finite-sum setting).

# B  Information-theoretic lower bounds

## B.1  Information-theoretic lower bound for stochastic inexact gradient model

We state and prove the information-theoretic lower bound for smooth costs.

**Theorem 6.** **(Information-theoretic Lower Bound)** *Assume that $\varepsilon < 1/200$ and $\varepsilon \lesssim \delta^2$. Then there exists a family of smooth cost functions $\{F_\theta : \mathbb{R} \to \mathbb{R}\}$ parameterized by $\theta \in [-1, 1]$ with the following property. Suppose A is any algorithm that for each $\theta \in [-1, 1]$ uses at most T queries to*

*stochastic inexact gradient oracle and outputs $\boldsymbol{x}_T^\theta$ that is $\varepsilon$-accurate, i.e., $\mathbb{E}\, F_\theta(\boldsymbol{x}_T^\theta) - \inf_{\boldsymbol{x}} F_\theta(\boldsymbol{x}) \le \varepsilon$. Then, there exists $\theta_{\mathsf{bad}} \in [-1, 1]$ such that $(\varepsilon, \delta)$-deviation (Definition 3 (a)) is lower bounded by $\Omega(\frac{\delta^2}{T\varepsilon^2})$.*

*Proof.* Let $\varepsilon > 0$ be a fixed small constant. We consider the following family of cost functions $\{F_\theta : \mathbb{R} \to \mathbb{R}\}$ parametrized by $\theta \in [-1, 1]$:

$$F_\theta(x) = \begin{cases} 100\varepsilon \cdot (x - \theta)^2, & \text{for } x \in [-1, 1], \\ 100\varepsilon \cdot (1 - \theta)^2 + 200\varepsilon(1 - \theta)(x - 1) & \text{for } x > 1 \\ 100\varepsilon \cdot (-1 - \theta)^2 + 200\varepsilon(-1 - \theta)(x + 1) & \text{for } x < -1. \end{cases} \tag{B.1}$$

Note that $F_\theta$ is $200\epsilon$-smooth for each $\theta \in [-1, 1]$.

We consider the following stochastic first order oracle for each $F_\theta$. For a queried point $x$, the oracle outputs $g_\theta(x)$ defined as

$$g_\theta(x) = \begin{cases} \frac{1}{200\varepsilon}\nabla F_\theta(x) + z & \text{for } z \sim N(0, \frac{\delta^2}{2 \cdot 200\epsilon}), \quad \text{w.p. } 200\epsilon, \\ 0, & \text{w.p. } 1 - 200\epsilon. \end{cases} \tag{B.2}$$

Since we define $F_\theta$ as a linear extension outside $[-1, 1]$, below we may assume that all gradient queries are made within $[-1, 1]$.

Assuming that all queries are made in $[-1, 1]$, we can rewrite (B.2) simply as

$$g_\theta(x) = (x - \theta + z) \cdot \mathsf{Sample}, \tag{B.3}$$

where $\mathsf{Sample} \sim \text{Bernoulli}(200\epsilon)$. Let us verify that this is a valid stochastic first order oracle. First, it is clear that

$$\mathbb{E}[g_\theta(x)] = 200\epsilon \cdot (x - \theta + 0) + (1 - 200\epsilon) \cdot 0 = 200\epsilon \cdot (x - \theta) = \nabla F_\theta(x).$$

Next, for the variance, note that

$$\begin{aligned} \mathbb{E}\left[g_\theta(x) - \nabla F_\theta(x)\right]^2 &= \mathbb{E}\left[(x - \theta + z) \cdot \mathsf{Sample} - \nabla F_\theta(x)\right]^2 \\ &= 200\epsilon \cdot \mathbb{E}\left[x - \theta + z - 200\epsilon \cdot (x - \theta)\right]^2 + (1 - 200\epsilon) \cdot \left[-200\epsilon \cdot (x - \theta)\right]^2 \\ &= 200\epsilon(1 - 200\epsilon)^2 \cdot (x - \theta)^2 + (200\epsilon)^2(1 - 200\epsilon) \cdot (x - \theta)^2 + 200\epsilon \cdot \mathbb{E}[z^2] \\ &= 200\epsilon(1 - 200\epsilon) \cdot (x - \theta)^2 + \delta^2/2. \end{aligned}$$

Hence, the variance is always upper bounded by $\delta^2/2 + 200\epsilon(1 - 200\epsilon)2^2 \le \delta^2$, as long as $\varepsilon \lesssim \delta^2$.

We now prove the theorem. From the fact that the output $x_T^\theta$ is $\varepsilon$-accurate, we have

$$\forall \theta \in [-1, 1], \quad 100\varepsilon \cdot \mathbb{E}\left[x_T^\theta - \theta\right]^2 \le \varepsilon.$$

Using Jensen's inequality, we know $\left|\mathbb{E}[x_T^\theta] - \theta\right|^2 = \left|\mathbb{E}[x_T^\theta - \theta]\right|^2 \le \mathbb{E}[x_T^\theta - \theta]^2$ which implies the following condition:

$$\forall \theta \in [-1, 1], \quad \left|\mathbb{E}[x_T^\theta] - \theta\right| \le 0.1. \tag{B.4}$$

This condition says if we regard $A$ as an estimator of $\theta$ for each $\theta$, the bias is less than equal to $0.1$. For the following argument, we hence change our perspective and regard $A$ as an estimator of $\theta$ based on $T$ inexact gradient queries rather than an optimization algorithm.

Let us fix $\theta \in [-1, 1]$. We first that we may assume that all gradient queries are made at the point $0$. Indeed, from the expression for the stochastic oracle (B.3), we know

$$g_\theta(x) \stackrel{d}{=} g_\theta(0) + x \cdot \mathsf{Sample}.$$

In particular, this implies that one can reconstruct a gradient query at any point $x$ based on a gradient query at the point $0$. Hence, without loss of generality, we may assume that all gradient queries are made at $x = 0$.

Hence, we can regard $A$ as an estimator of $\theta$ based on $T$ independent measurements $y_1, \ldots, y_T$ of form

$$y_i = (-\theta + z_i) \cdot \mathsf{Sample}_i, \quad i = 1, \ldots, T, \tag{B.5}$$

where $z_i \sim N(0, \frac{\delta^2}{200\epsilon})$ and $\mathsf{Sample}_i \sim \mathrm{Bernoulli}(200\epsilon)$. Now with this new perspective in mind, we can lower bound the variance of the estimator $\mathrm{Var}(x_T^\theta)$ using the Cramer-Rao lower bound. To that end, we first calculate the fisher information of the measurement distribution.

Recall that each measurement is of form

$$y = (-\theta + z) \cdot \mathsf{Sample},$$

where $z \sim N(0, \frac{\delta^2}{2 \cdot 200\epsilon})$ and $\mathsf{Sample} \sim \mathrm{Bernoulli}(200\epsilon)$. Then, the log likelihood is

$$\ell(\theta; y) = \ln \left[ \frac{1}{\sqrt{2\pi \cdot \frac{\delta^2}{2 \cdot 200\epsilon}}} \exp\left( -\frac{2 \cdot 200\epsilon}{\delta^2} \cdot \frac{(y+\theta)^2}{2} \right) \right] \mathbb{1}\{\mathsf{Sample} = 1\} + \delta_{[y=0]} \cdot \mathbb{1}\{\mathsf{Sample} = 0\}.$$

Taking derivatives of the log likelihood, we get

$$\nabla_\theta \ell(\theta; y) = -\frac{2 \cdot 200\epsilon}{\delta^2} \cdot (\theta + y) \cdot \mathbb{1}\{\mathsf{Sample} = 1\},$$

$$\nabla_\theta^2 \ell(\theta; y) = -\frac{2 \cdot 200\epsilon}{\delta^2} \cdot \mathbb{1}\{\mathsf{Sample} = 1\}.$$

Hence, the Fisher information is equal to

$$I(\theta) = \underset{\theta}{\mathrm{Cov}} \, \nabla_\theta \ell(\theta; y) = -\underset{\theta}{\mathbb{E}} \, \nabla_\theta^2 \ell(\theta; y) = \frac{2 \cdot 200\epsilon}{\delta^2} \cdot \Pr[\mathsf{Sample} = 1] = \frac{2 \cdot (200\epsilon)^2}{\delta^2}.$$

Hence, the fisher information $I_T(\theta)$ for the $T$ independent measurements is equal to $T \frac{2 \cdot (200\epsilon)^2}{\delta^2}$.

Let us recall the Cramér-Rao lower bound for biased estimators. (see, e.g., [Eldar, 2004, (3)]).

**Proposition 1.** *Let $b(\theta) := \mathbb{E}[\hat{\theta}] - \theta$ be the bias of an estimator $\hat{\theta}$. Then, the following bounds hold:*

$$\mathrm{Var}(\hat{\theta}) \geq \frac{[1 + b'(\theta)]^2}{I(\theta)}.$$

From (B.4), it must be that $b'(\theta_{\mathsf{bad}}) \geq -\frac{1}{2}$ for some $\theta_{\mathsf{bad}} \in [-1, 1]$. To see this, suppose to the contrary that $b'(\theta) < -\frac{1}{2}$ for all $\theta \in [-1, 1]$. Then, it must be that

$$b(1) \leq -\frac{1}{2} \cdot 2 + b(-1) \overset{(B.4)}{\leq} -1 + 0.1 = -0.9,$$

which is a contradiction since (B.4) ensures that $b(1) \geq -0.1$. Thus, Proposition 1 gives

$$\mathrm{Var}(x_T^{\theta_{\mathsf{bad}}}) \geq \frac{(1 + b'(\theta_{\mathsf{bad}}))^2}{I_T(\theta_{\mathsf{bad}})} \geq \Omega\left( \frac{\delta^2}{T\varepsilon^2} \right).$$

This concludes the proof of the lower bound. $\qquad\square$

## B.2 Proof of lower bound (stochastic global oracle)

Recall Theorem 5 from the main text.

**Theorem 5.** *Assume that $\varepsilon < 1/200$ and $\varepsilon \lesssim \delta^2$. Then there exists a family of population costs $F_\theta(x) = \mathbb{E}_{\xi \sim P_\theta} f_\theta(x, \xi)$ parametrized by $\theta \in [1, 2]$, and a $\delta$-bounded stochastic global oracle for $F_\theta$, satisfying the following property. Suppose that $A$ is any algorithm that for each $\theta \in [1, 2]$ uses at most $T$ queries to the stochastic global oracle and outputs $x_T^\theta$ that is $\varepsilon$-accurate, i.e., $\mathbb{E} \, F_\theta(x_T^\theta) - \inf_x F_\theta(x) \leq \varepsilon$. Then, there exists $\theta_{\mathsf{bad}} \in [1, 2]$ such that $\mathrm{Var}(x_T^{\theta_{\mathsf{bad}}}) \geq \Omega(\delta^2/(T\varepsilon^2))$.*

*Proof.* The construction and argument are analogous to the proof of Theorem 6 (Subsection B.1). Fix $\varepsilon > 0$ and consider the following family of cost functions $\{F_\theta : \mathbb{R} \to \mathbb{R}\}$ parametrized by unknown $\theta \in [1, 2]$:

$$F_\theta(x) = \begin{cases} 200\varepsilon \cdot \left\{ \frac{1}{2}x^2 - \theta x \right\}, & \text{for } x \in [1, 2] \\ \text{linear extension} & \text{for } x \notin [1, 2]. \end{cases} \tag{B.6}$$

Note that $F_\theta$ is $200\epsilon$-smooth for all $\theta \in [-1, 1]$ and the minimum is achieved at $x = \theta$. Below, let us fix a ground truth parameter $\theta$ and let $F(x) = F_\theta(x)$.

Now define $f(x, \xi)$ as follows: with probability $200\varepsilon$,

$$f(x, \xi) = \begin{cases} \frac{1}{200\varepsilon}F(x) + zx & \text{for } z \sim N(0, \frac{\delta^2}{2 \cdot 200\epsilon}), & \text{if } x \in [1, 2], \\ \frac{1}{200\varepsilon}F(x) + 2z. & & \text{if } x > 2, \\ \frac{1}{200\varepsilon}F(x) + z & & \text{if } x < 1, \end{cases} \tag{B.7}$$

and $f(x, \xi) = 0$ with probability $1 - 200\varepsilon$. Then clearly we have $\mathbb{E}_\xi f(x, \xi) = F(x)$.

We first check that this construction satisfies Definition 5. It is sufficient to check the condition for $x \in [1, 2]$ since outside the interval the cost is defined as the linear extension. For $x \in [1, 2]$, we have

$$\nabla f(x, \xi) = \begin{cases} x - \theta + z & \text{for } z \sim N(0, \frac{\delta^2}{2 \cdot 200\epsilon}), & \text{w.p. } 200\epsilon, \\ 0, & & \text{w.p. } 1 - 200\epsilon. \end{cases} \tag{B.8}$$

This is precisely the expression (B.2) in the proof of Theorem 6 (Subsection B.1), and hence, this clearly satisfies Definition 5.

Now the key fact of the proof is that one can reconstruct the complete specification of the function $f(\cdot, \xi)$ based on a gradient query, provided that it is nonzero. This is because if nonzero, the gradient query at $x$ is equal to $(x - \theta + z)$. This reveals $(\theta - z)$, from which one can reconstruct the complete characterization $f(x, \xi) = \frac{1}{2}x^2 - (\theta - z)x$.

Hence, the information revealed by a single query to the stochastic global oracle is as good as that revealed by a single query to the stochastic global oracle. Thus, the setting is reduced to that of Theorem 6 (Subsection B.1), and using the same argument, the proof follows. $\qquad\square$

## C  Proof of lower bounds (smooth costs)

We first introduce a helper function we will use throughout the proofs of FOI lower bounds in the remaining sections.

**Helper function for smooth costs lower bounds.**    We will frequently use the following function for the FOI lower bounds for smooth cost. Let $\mathcal{F} : \mathbb{R} \to \mathbb{R}$ be an one-dimensional function defined as

$$\mathcal{F}(x) := \begin{cases} x^2 & \text{if } x \in [0, 1], \\ 2x - 1 & \text{if } x \geq 1, \\ 0 & \text{if } x \leq 0. \end{cases} \tag{C.1}$$

For reader's convenience, we illustrate the helper function $\mathcal{F}$ in Figure 1 below.

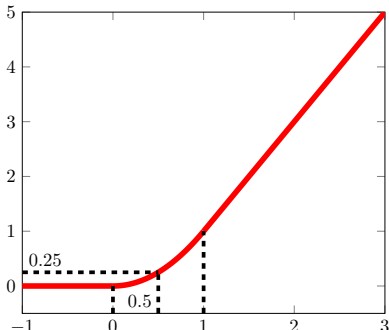

Figure 1: Illustration of the helper function $\mathcal{F}$ for the smooth costs lower bounds.

### C.1  Stochastic inexact gradient model

The proofs of FOI lower bounds are quite technical and rely on delicate constructions, and to illustrate our general proof strategy, we first present a proof that is relatively simpler, yet captures the essence

of the later complicated constructions. More specifically, in this section, we will prove the following FOI lower bounds for smooth costs against stochastic inexact oracle.

**Theorem 7. (Lower Bound)** *Let $\varepsilon > 0$, and $T$ be a given number of iterations. There exists an $O(1)$-smooth convex function $f : \mathbb{R}^{O(T)} \to \mathbb{R}$ and a stochastic inexact gradient oracle such that any FOI algorithm $\mathcal{A}$ that starts at $\boldsymbol{x}_0 = \boldsymbol{0}$ has its $(\varepsilon, \delta)$-deviation lower bounded by $\Omega(\frac{\delta^2}{T\varepsilon^2})$.*

*Proof.* Now consider the cost $f : \boldsymbol{x} = (\boldsymbol{x}^{\mathsf{dum}}, y) \in \mathbb{R}^T \times \mathbb{R} \to \mathbb{R}$ defined as

$$f(\boldsymbol{x}) = 4\varepsilon \cdot \mathcal{F}(y+1), \tag{C.2}$$

where $\mathcal{F}$ is defined in (C.1). Here note that $\boldsymbol{x}^{\mathsf{dum}} \in \mathbb{R}^T$ is dummy coordinates which do not appear in the cost $f$. Next, we define the stochastic inexact oracle for $t = 0, 1, \ldots, T-1$ as

$$g(\boldsymbol{x}_t) = \nabla f(\boldsymbol{x}_t) + \delta r_t \boldsymbol{e}_{1+t} \quad \text{where } r_t \sim \mathrm{Unif}\{\pm 1\}, \tag{C.3}$$

where $\boldsymbol{e}_j$ is the $j$-th coordinate vector. Then, clearly this stochastic gradient fulfills the definition of stochastic inexact gradient oracle. Here the stochastic gradient noises are designed such that during the $t$-th iteration, the noise is added to the coordinate $\boldsymbol{x}^{\mathsf{dum}}[1 + t]$. In other words, the noises will be added to each coordinate of $u$ incrementally.

From here one, let us write iterates $\boldsymbol{x}_t = (\boldsymbol{x}_t^{\mathsf{dum}}, y_t)$. Note first that for $\varepsilon$-accuracy, it must be that $|y_T - y_0| \geq 1/2$; otherwise $f(\boldsymbol{x}_T) > 4\varepsilon \cdot \mathcal{F}(0.5) = \varepsilon$; see Figure 1. Based on the definition of FOI (see (FOI)), let us write

$$\boldsymbol{x}_T = \boldsymbol{x}_0 - \sum_{t=0}^{T-1} \lambda_t^{(T)} g(\boldsymbol{x}_t).$$

Then from the construction (C.2), we know that for any $\boldsymbol{x}$, we know $\frac{\partial f}{\partial y}(\boldsymbol{x}) \in [0, 8\varepsilon]$. On the other hand, as we discussed, we need $|y_T - y_0| \geq 1/2$. Hence, in order for iterates to move far enough from the starting point, the coefficients have to add up to a sufficiently large number:

$$\sum_{t=0}^{T-1} |\lambda_t^{(T)}| \geq \frac{1}{16\varepsilon}, \tag{C.4}$$

since otherwise, $|y_T - y_0| < 8\varepsilon \cdot \frac{1}{16\varepsilon} = 1/2$.

Now we will make use of the condition (C.4) to show that there is a large deviation in the coordinates $\boldsymbol{x}^{\mathsf{dum}}$. More specifically, let us lower bound $\mathbb{E} \left\| \boldsymbol{x}_T^{\mathsf{dum}} - \mathbb{E}[\boldsymbol{x}_T^{\mathsf{dum}}] \right\|^2$. From the construction of inexact oracle (C.3), it follows that

$$\mathbb{E} \left\| \boldsymbol{x}_T^{\mathsf{dum}} - \mathbb{E}[\boldsymbol{x}_T^{\mathsf{dum}}] \right\|^2 = \mathbb{E} \left\| \sum_{t=0}^{T-1} \lambda_t^{(T)} \delta r_t \cdot \boldsymbol{e}_{1+t} \right\|^2 = \sum_{t=0}^{T-1} (\lambda_t^{(T)})^2 \delta^2 \, \mathbb{E}[r_t^2]$$

$$= \sum_{t=0}^{T-1} (\lambda_t^{(T)})^2 \delta^2 \overset{(a)}{\geq} \delta^2 \cdot \frac{1}{T} \cdot \left( \sum_{t=0}^{T-1} |\lambda_t^{(T)}| \right)^2 \gtrsim \frac{\delta^2}{T\varepsilon^2},$$

where $(a)$ is due to Cauchy-Schwarz inequality. This concludes the proof. $\qquad\square$

## C.2 Non-stochastic inexact gradient model

**Theorem 8. (Lower Bound)** *Let $\varepsilon > 0$ be a small constant, and $T$ be a given number of iterations. There exists a $O(1)$-smooth convex function $f : \mathbb{R}^2 \to \mathbb{R}$ with a non-stochastic inexact gradient model such that for any FOI algorithm $\mathcal{A}$ that starts at $\boldsymbol{x}_0 = \boldsymbol{0}$ has the $(\varepsilon, \delta)$-deviation lower bounded by $\Omega(\frac{\delta^2}{\varepsilon^2})$.*

*Proof.* With $\mathcal{F}$ defined as (C.1), this time we consider a simpler construction: the cost $f : \boldsymbol{x} = (x, y) \in \mathbb{R} \times \mathbb{R} \to \mathbb{R}$ is defined as

$$f(x, y) = 4\varepsilon \cdot \mathcal{F}(y+1). \tag{C.5}$$

Let us write the iterate as $\boldsymbol{x}_t = (x_t, y_t)$. Note that for $\varepsilon$-accuracy, it must be that $y_T \geq 1/2$; otherwise $f(y_T) > \varepsilon$. This means that in order to achieve $\varepsilon$-suboptimality, the $y$-component of the iterate has to move at least constant distance away from the starting point.

Now consider the following non-stochastic inexact oracle

$$g(\boldsymbol{x}_t) = \nabla f(\boldsymbol{x}_t) + \delta \cdot \frac{\partial}{\partial y} \mathcal{F}(y_t) \cdot \boldsymbol{e}_1, \tag{C.6}$$

where $\boldsymbol{e}_1$ is the first coordinate vector. Note that this is a valid oracle because $0 \leq \frac{\partial}{\partial y} \mathcal{F}(y) \in [0, 8\varepsilon] \in [0, 1]$ for all $y$. Then from the construction of the inexact gradient oracle (C.6), it follows that

$$\frac{\delta}{4\varepsilon} y_T = x_T \tag{C.7}$$

Letting $\boldsymbol{x}_T^{\mathsf{exact}} = (x_t^{\mathsf{exact}}, y_t^{\mathsf{exact}})$ be the iterate with exact gradients (without the gradient noises), since we know $x_T^{\mathsf{exact}} = 0$, (C.7) implies

$$\left\| \boldsymbol{x}_T - \boldsymbol{x}_T^{\mathsf{exact}} \right\|^2 \geq \left| x_T - x_T^{\mathsf{exact}} \right|^2 = x_T^2 \gtrsim \frac{\delta^2}{\varepsilon^2}.$$

This is the desired lower bound. $\qquad\square$

### C.3 Inexact initialization model

**Theorem 9. (Lower Bound)** *Let $\varepsilon > 0$ be a small constant, and $T$ be a given number of iterations. There exists a $O(1)$-smooth convex function $f : \mathbb{R}^{O(T)} \to \mathbb{R}$ such that for any FOI algorithm $\mathcal{A}$ the $(\varepsilon, \delta)$-deviation lower bounded by $\Omega(\delta^2)$ w.r.t. the reference point $\boldsymbol{x}_0^{\mathsf{ref}} = \boldsymbol{0}$.*

*Proof.* Consider $f : \mathbb{R}^2 \to \mathbb{R}$ defined as $f(x, y) = (y - 1)^2$. Choose $x_0^{\mathsf{ref}} = 0$ and the inexact initialization to be $x_0 = (\delta, 0)$. Then, any first order algorithm only updates the second coordinate, which implies that after $T$ iterations, we still have $\left\| x_T - x_T^{\mathsf{ref}} \right\| \geq \delta^2$. $\qquad\square$

### C.4 Stochastic inexact gradient model (strongly convex costs)

**Theorem 10. (Lower Bound)** *Let $\varepsilon > 0$ and $T$ be a given number of iterations. There exists a $O(1)$-smooth and $\mu$-strongly convex function $f : \mathbb{R}^{O(T)} \to \mathbb{R}$ with a stochastic inexact gradient model such that any FOI algorithm $\mathcal{A}$ that starts at $\boldsymbol{x}_0 = \boldsymbol{0}$ has its $(\varepsilon, \delta)$-deviation lower bounded by $\Omega\left(\frac{\delta^2}{T\mu^2} \wedge \frac{\varepsilon}{\mu}\right)$.*

*Proof.* For $\boldsymbol{x} = (\boldsymbol{x}^{\mathsf{dum}}, y)$ where $\boldsymbol{x}^{\mathsf{dum}} \in \mathbb{R}^T$ and $y \in \mathbb{R}$, consider the cost

$$f(x, y) := y + \frac{\mu}{2} y^2 + \frac{\mu}{2} \left\| \boldsymbol{x} \right\|^2 .$$

We consider the initialization $\boldsymbol{x}_0 = (0, 0, 0, \dots, 0)$.

Next, we define the stochastic inexact oracle for $t = 0, 1, \dots, T - 1$ as

$$\boldsymbol{g}(\boldsymbol{x}_t) = \nabla f(\boldsymbol{x}_t) + \delta r_t \cdot \boldsymbol{e}_{1+t} \quad \text{for } r_t \sim \mathrm{Unif}\{\pm 1\}, \tag{C.8}$$

where $\boldsymbol{e}_j$ is the $j$-th coordinate vector. Also, throughout the proof we use the notation:

$$\boldsymbol{g}_t^{\mathsf{dum}} := \boldsymbol{g}(\boldsymbol{x}_t^{\mathsf{dum}}, y_t)[1, 2, \dots, T] \quad \text{and} \quad g_t^y := \boldsymbol{g}(\boldsymbol{x}_t^{\mathsf{dum}}, y_t)[T + 1] .$$

**Warm-up: the case of simplified gradient noises.** For a moment, we assume that the inexact gradient oracle is non-stochastic with

$$\boldsymbol{g}(\boldsymbol{x}_t) = \nabla f(\boldsymbol{x}_t) + \delta \cdot \boldsymbol{e}_{1+t} .$$

We consider this case first to build the key intuition of the proof. The first prove the following result that is crucial for the proof.

**Lemma 1.** *For each $t = 0, 1, \ldots, T-1$, the output of a FOI algorithm satisfies*

$$\delta \cdot y_t = \sum_{i=1}^{T} x_t[i] \quad and \quad \delta \cdot g_t^y = \sum_{i=1}^{T} \boldsymbol{g}_t^{\mathsf{dum}}[i], \quad for\ each\ t = 0, 1, 2, \ldots, T. \tag{C.9}$$

*Proof.* We prove by induction on $t$. The statement trivially holds for $t = 0$. Assume that the conclusion holds for some $t$. We will first show that

$$\delta \cdot y_{t+1} = \sum_{i=1}^{T} x_{t+1}[i]. \tag{C.10}$$

By the definition of FOI (see (FOI)), we have

$$\delta \cdot y_{t+1} = -\delta \cdot \sum_{j=0}^{t} \lambda_j^{(t+1)} g_j^y$$

$$= -\sum_{j=0}^{t} \lambda_j^{(t+1)} \left( \sum_{i=1}^{T} \boldsymbol{g}_j^{\mathsf{dum}}[i] \right) = -\sum_{i=1}^{T} \sum_{j=0}^{t} \lambda_j^{(t+1)} \boldsymbol{g}_j^{\mathsf{dum}}[i]$$

$$= \sum_{i=1}^{T} x_{t+1}[i].$$

This completes the proof of (C.10). Next, we will show that

$$\delta \cdot g_{t+1}^y = \sum_{i=1}^{T} \boldsymbol{g}_{t+1}^{\mathsf{dum}}[i]. \tag{C.11}$$

This follows because

$$\delta \cdot g_{t+1}^y = \delta(1 + \mu y_{t+1}) = \delta + \mu \left( \sum_{i=1}^{T} \boldsymbol{x}_{t+1}^{\mathsf{dum}}[i] \right)$$

$$\overset{(a)}{=} \boldsymbol{g}_{t+1}^{\mathsf{dum}}[t+2] + \sum_{i=1}^{t+1} \boldsymbol{g}_{t+1}^{\mathsf{dum}}[i] = \sum_{i=1}^{T} \boldsymbol{g}_{t+1}^{\mathsf{dum}}[i],$$

where $(a)$ uses the fact that $\boldsymbol{x}_{t+1}^{\mathsf{dum}}[i] = 0$ for all $i > t + 1$. $\qquad\square$

By Lemma 1, it holds that

$$\delta \cdot y_T = \sum_{i=1}^{T} \boldsymbol{x}_T^{\mathsf{dum}}[i] \tag{C.12}$$

On the other hand, in order to achieve $\varepsilon$-suboptimality, we need $y_T^2 \gtrsim \frac{1}{\mu^2}$. Moreover, for $\varepsilon$-suboptimality, we also need $\sum_{i=1}^{T} \boldsymbol{x}_T^{\mathsf{dum}}[i]^2 \lesssim \frac{\varepsilon}{\mu}$. Therefore, letting $\boldsymbol{x}_T^{\mathsf{exact}} = ((\boldsymbol{x}_T^{\mathsf{dum}})^{\mathsf{exact}}, y_T^{\mathsf{exact}})$ be the iterate with exact gradients, since $(\boldsymbol{x}_T^{\mathsf{dum}})^{\mathsf{exact}} = 0$, the condition (C.12) implies the following: whenever $\sum_{i=1}^{T} \boldsymbol{x}_T^{\mathsf{dum}}[i]^2 \lesssim \frac{\varepsilon}{\mu}$,

$$\left\| \boldsymbol{x}_T - \boldsymbol{x}_T^{\mathsf{exact}} \right\|^2 \geq \left\| \boldsymbol{x}_T^{\mathsf{dum}} - (\boldsymbol{x}_T^{\mathsf{dum}})^{\mathsf{exact}} \right\|^2 = \sum_{i=1}^{T} \boldsymbol{x}_T^{\mathsf{dum}}[i]^2$$

$$\geq \frac{1}{T} \left( \sum_{i=1}^{T} \boldsymbol{x}_T^{\mathsf{dum}}[i] \right)^2 = \frac{1}{T} \cdot \delta^2 \cdot (y_T)^2 \gtrsim \frac{\delta^2}{T\mu^2}.$$

This is precisely equal to the desired lower bound.

**Actual proof for the stochastic noise case.** Now coming back to the stochastic inexact gradient (C.8), one can prove the following analog of Lemma 1:

$$\delta \cdot |y_t| \leq \sum_{i=1}^{T} |\boldsymbol{x}_t^{\mathsf{dum}}[i]| \quad \text{and} \quad \delta \cdot |g_t^y| \leq \sum_{i=1}^{T} |\boldsymbol{g}_t^{\mathsf{dum}}[i]|, \quad \text{for each } t = 0, 1, 2, \ldots, T. \quad \text{(C.13)}$$

Here we note that the construction ensures that $|\boldsymbol{x}_T^{\mathsf{dum}}[i]|$'s are deterministic quantities (because regardless of whether $r_t = \pm 1$ the absolute value is the same), and that is why we do not write the expectation operators next to them.

The above result holds for the following reason, when $r_t = +1$ for all $t$, the stochastic inexact gradient reduces to the non-stochastic inexact gradient, in which case the equality holds in (C.13) without absolute values. With $r_t = \pm 1$, one can no longer argue this. On the other hand, one can apply triangle inequalities to obtain (C.13).

Again, in order to achieve $\varepsilon$-suboptimality, we need $y_T^2 \gtrsim \frac{1}{\mu^2}$ and $\sum_{i=1}^{T} \boldsymbol{x}_T^{\mathsf{dum}}[i]^2 \lesssim \frac{\varepsilon}{\mu}$. Therefore, whenever $\sum_{i=1}^{T} \boldsymbol{x}_T^{\mathsf{dum}}[i]^2 \lesssim \frac{\varepsilon}{\mu}$, we have

$$\mathbb{E}\left\|\boldsymbol{x}_T - \mathbb{E}[\boldsymbol{x}_T]\right\|^2 \geq \sum_{i=1}^{T} |\boldsymbol{x}_T^{\mathsf{dum}}[i]|^2 \geq \frac{1}{T}\left(\sum_{i=1}^{T} |\boldsymbol{x}_T^{\mathsf{dum}}[i]|\right)^2 \geq \frac{1}{T} \cdot \delta^2 \cdot |y_T|^2 \gtrsim \frac{\delta^2}{T\mu^2}.$$

This completes the proof. □

### C.5 Non-stochastic inexact gradient model

**Theorem 11. (Lower Bound)** *Let $\varepsilon > 0$ be a small constant, and $T$ be a given number of iterations. There exists a $O(1)$-smooth $\mu$-strongly convex function $f : \mathbb{R}^2 \to \mathbb{R}$ with a non-stochastic inexact gradient model such that for any FOI algorithm $\mathcal{A}$ that starts at $\boldsymbol{x}_0 = \boldsymbol{0}$ has the $(\varepsilon, \delta)$-deviation lower bounded by $\Omega(\frac{\delta^2}{\mu^2} \wedge \frac{\varepsilon}{\mu})$.*

*Proof.* For simplicity, we assume throughout the proof that $D = 1$ and for $\boldsymbol{x} = (x, y)$ where $x, y \in \mathbb{R}$, consider the cost

$$f(x, y) := y + \frac{\mu}{2}y^2 + \frac{\mu}{2}x^2$$

We consider the initialization $\boldsymbol{x}_0 = (0, 0)$. Next, consider the following non-stochastic inexact oracle

$$g(\boldsymbol{x}_t) = \nabla f(\boldsymbol{x}_t) + \delta \boldsymbol{e}_1 \quad \text{(C.14)}$$

where $\boldsymbol{e}_1$ is the first coordinate vector. Then from this construction, one can verify similarly to Lemma 1 that

$$g(\boldsymbol{x}_t)[1] = \delta \cdot g(\boldsymbol{x}_t)[2] \quad \text{and} \quad x_t = \delta \cdot y_t$$

for $t = 0, 1, \ldots, T$.

Now from the $\varepsilon$-suboptimality, it must be that $y_T^2 \gtrsim \frac{1}{\mu^2}$ and $x_T^2 \lesssim \frac{\varepsilon}{\mu}$. Hence, whenever $x_T^2 \lesssim \frac{\varepsilon}{\mu}$ holds, we have the following deviation bound since $x_T^{\mathsf{exact}} = 0$:

$$\left\|\boldsymbol{x}_T - \boldsymbol{x}_T^{\mathsf{exact}}\right\|^2 \geq |x_T - x_T^{\mathsf{exact}}|^2 = \delta^2 \cdot y_t^2 \gtrsim \frac{\delta^2}{\mu^2}.$$

This completes the proof. □

### C.6 Inexact initialization model (strongly convex costs)

**Theorem 12. (Lower Bound)** *Let $\varepsilon > 0$ be a small constant, and $T$ be a given number of iterations. There exists a $O(1)$-smooth $\mu$-strongly convex function $f : \mathbb{R}^{\Omega(T)} \to \mathbb{R}$ such that for any FOI algorithm $\mathcal{A}$ the $(\varepsilon, \delta)$-deviation lower bounded by $\Omega(\exp(-\Omega(T))\delta^2 \wedge \frac{\varepsilon}{\mu})$ w.r.t. the reference point $\boldsymbol{x}_0^{\mathsf{ref}} = \boldsymbol{0}$.*

*Proof.* We use the construction in [Nesterov, 2018, Theorem 2.1.13]. In particular, for simplicity, we consider the construction for the infinite dimensional Hilbert space $\ell_2$ as it simplifies the proof; in fact, a similar argument works for $\mathbb{R}^{\Omega(T)}$. Let us recall the construction (we follow the presentation in [Bubeck, 2014, Theorem 3.15]). Let $A : \ell_2 \to \ell_2$ be the linear operator that corresponds to the infinite tri-diagonal matrix with $2$ on the diagonal and $-1$ on the upper and lower diagonals. For some constant $\kappa \geq 1$, consider the following $\mu$-strongly convex cost:

$$f^{\mathsf{Nes}}(x) = \frac{\mu(\kappa - 1)}{8} \left( \langle Ax, x \rangle - 2\langle \boldsymbol{e}_1, x \rangle \right) + \frac{\mu}{2} \|x\|^2 \quad \text{and} \quad q := \frac{\sqrt{\kappa} - 1}{\sqrt{\kappa} + 1}.$$

For the zero initialization $x_0 = (0, 0, \dots) \in \ell_2$, the cost satisfies the following properties (see the proof of [Nesterov, 2018, Theorem 2.1.13]):

- Output of any FOI satisfies $x_t[i] = 0, \forall i \geq t$.

- $x^*[i] = q^i$.

- $\|x_0 - x^*\|^2 = \sum_{i=1}^{\infty} (x^*[i])^2 = \sum_{i=1}^{\infty} q^{2i} = \frac{q^2}{1-q^2}$.

- $\|x_t - x^*\|^2 \geq \sum_{i=k+1}^{\infty} q^{2i} = \frac{q^{2(t+1)}}{1-q^2} = q^{2t} \|x_0 - x^*\|^2$.

Now we consider the following cost function: for $\boldsymbol{x} = (x, y) \in \ell_2 \times \ell_2$

$$f(\boldsymbol{x}) = f^{\mathsf{Nes}}(x) + f^{\mathsf{Nes}}(y),$$

and we consider the two initializations:

$$\boldsymbol{x}_0^{\mathsf{ref}} = \left( (0, 0, 0, \dots), (q, q^2, q^3, \dots) \right)$$

$$\boldsymbol{x}_0 = \left( (0, 0, 0, \dots), (q, q^2, q^3, \dots, q^L, 0, 0, \dots) \right)$$

for $L = \Omega(\log(1/\delta))$ is chosen such that $q^{L+1}/(1 - q) \leq \delta$. Then, it follow that $\left\| \boldsymbol{x}_0 - \boldsymbol{x}_0^{\mathsf{ref}} \right\| = \frac{q^{L+1}}{1-q} \leq \delta$. On the other hand, it follows from the above property that

$$\left\| \boldsymbol{x}_t - \boldsymbol{x}_t^{\mathsf{ref}} \right\|^2 \geq q^t \frac{q^{2L+1}}{1 - q^2} \approx q^t \cdot \delta^2.$$

Hence, as long as $\|\boldsymbol{x}_t - \boldsymbol{x}^*\|^2, \left\| \boldsymbol{x}_t^{\mathsf{ref}} - \boldsymbol{x}^* \right\|^2 \lesssim \frac{\varepsilon}{\mu}$, the deviation lower bound follows. $\qquad \square$

# D  Proof of lower bounds (nonsmooth costs)

In this section, we present the proofs of lower bounds for nonsmooth costs. The proof will be based on more complicated constructions than those for the case of smooth costs, so before we dive into the proofs, we first present some intuition behind the constructions.

## D.1  Warm-up: lower bound against GD

In this section, as a warm-up, we will prove a (weaker) lower bound for a simplified setting. In particular, we prove the lower bound against gradient descent (GD). Formally, in the definition of FOI, we restrict that $\lambda_i^{(t)} \equiv \lambda_i$ (i.e., the coefficient is a positive number does not depend on the iterations $t$). In other words, for $\lambda_i, i = 0, 1, \dots, T - 1$,

$$\boldsymbol{x}_t = \boldsymbol{x}_0 - \sum_{i=0}^{t-1} \lambda_i g(\boldsymbol{x}_i) \quad \text{for each } t = 1, 2, \dots, T. \tag{D.1}$$

Note that this is precisely GD with step sizes $\lambda_t$'s. For the lower bound construction, let $\boldsymbol{x} = (\boldsymbol{x}^{\mathsf{err}}, w) \in \mathbb{R}^T \times \mathbb{R}$ and consider the cost

$$f(\boldsymbol{x}^{\mathsf{err}}, w) = \underbrace{\max \left\{ 0, \max_{i=1,\dots,T} \{\boldsymbol{x}^{\mathsf{err}}[i]\} \right\}}_{=: G(\boldsymbol{x}^{\mathsf{err}})} + \underbrace{2\varepsilon \cdot \max\{w + 1, 0\}}_{=: \ell(w)}.$$

Since the above cost function is nonsmooth, we specify the subgradient oracle as follows: for both max terms above, we consider the subgradient oracle that outputs the subgradient corresponding to the first argument that achieves the maximum. Note that $\inf_{\boldsymbol{x}} f(\boldsymbol{x}) = 0$. Consider the zero initialization, i.e., $(\boldsymbol{x}_0^{\mathrm{err}}, w_0) = (\boldsymbol{0}, 0)$, and we write the iterates as $\boldsymbol{x}_t := (\boldsymbol{x}_t^{\mathrm{err}}, w_t) \in \mathbb{R}^T \times \mathbb{R}$.

For intuition, we describe the role of each coordinate:

- The first $T$ coordinates, $\boldsymbol{x}^{\mathrm{err}} \in \mathbb{R}^T$, correspond to the part where the errors due to inexact oracle are added.
- The last coordinate, $w \in \mathbb{R}$, governs the overall cost; in order to achieve $\varepsilon$-accuracy, the optimization algorithm has to decrease coordinate $w$ by at least $1/2$.

The proof proceeds by considering two different scenarios:

**Scenario 1 (exact gradients).** Consider the case where there is no noise in the gradients, i.e., $g(\boldsymbol{x}_t) = \nabla f(\boldsymbol{x}_t)$ for all $t$. Then, since $\boldsymbol{x}_0^{\mathrm{err}} = \boldsymbol{0}$, it follows that $\nabla G(\boldsymbol{x}_t^{\mathrm{err}}) = \boldsymbol{0}$ for all $t$. Hence the algorithm will only update coordinate $w_t$. Note that $\ell(w_0) = 2\varepsilon$, and hence in order to achieve $\ell(w_T) \leq \varepsilon$, it must be that $w_T \leq -1/2$. On the other hand, we have

$$\frac{\partial}{\partial w} \ell(w) = 0 \ \text{ or } \ 2\varepsilon \quad \text{for any } \ w \in \mathbb{R}.$$

Hence, in order to achieve $w_T \leq -1/2$, it must be that

$$\sum_{t=0}^{T-1} \lambda_t \geq \Omega(1/\varepsilon). \tag{D.2}$$

This condition is analogous to (C.4) from the lower bound proof for smooth costs (Subsection C.1).

**Scenario 2 (inexact gradients).** Now let us consider the case where the gradient error during the $t$-th iteration is non-stochastic and equal to $-\delta \boldsymbol{e}_t$, i.e.,

$$g(\boldsymbol{x}_t) = \nabla f(\boldsymbol{x}_t) - \delta \boldsymbol{e}_{t+1} \quad \text{for all } t = 0, 1, \ldots, T-1.$$

Here $\boldsymbol{e}_t$ denotes the $t$-th coordinate vector. Let us assume that $\delta$ is much smaller than all the step sizes $\lambda_t$, in particular, such that $\lambda_i \delta \ll \lambda_{i+1}$ for all $i = 0, \ldots, T-2$. Then from GD iterations defined as (D.1), one can deduce that

$$\boldsymbol{x}_t^{\mathrm{err}} = (-\lambda_1 + \lambda_0 \delta, \ -\lambda_2 + \lambda_1 \delta, \ \cdots, \ -\lambda_{t-1} + \lambda_{t-2}\delta, \ +\lambda_t \delta, 0, \ldots, 0).$$

Thus, the following estimate on the deviation holds:

$$\|\boldsymbol{x}_T^{\mathrm{err}}\|^2 = \left\| \sum_{t=1}^{T-1} (\lambda_t - \lambda_{t-1}\delta)\boldsymbol{e}_t + \lambda_{T-1}\delta \boldsymbol{e}_T \right\|^2 = \sum_{t=1}^{T-1} (\lambda_t - \lambda_{t-1}\delta)^2 \approx \sum_{t=1}^{T-1} \lambda_t^2. \tag{D.3}$$

**Combining the two scenarios.** Thus far, we have obtained (D.2) and (D.3) from the two different scenarios. The condition (D.2) shows that in order to achieve $\varepsilon$-suboptimality, stepsizes have to add up to a large number, more precisely, $\sum_{t=0}^{T-1} \lambda_t = \Omega(1/\varepsilon)$. On the other hand, (D.3) characterizes that the deviation is on the order of the quantity $\sum_{t=1}^{T-1} \lambda_t^2$. In order to formally connect these two conditions, we make the following assumption:

$$\lambda_0 \leq O\left( \sum_{t=1}^{T-1} \lambda_t \right). \tag{D.4}$$

Then with this assumption, one obtain the following deviation bound:

$$\|\boldsymbol{x}_T^{\mathrm{err}}\|^2 \approx \sum_{t=1}^{T-1} (\lambda_t^{(T)})^2 \overset{(a)}{\gtrsim} \frac{1}{T-1} \cdot \left( \sum_{t=1}^{T-1} \lambda_t^{(T)} \right)^2 \gtrsim \frac{1}{T-1} \cdot \left( \sum_{t=0}^{T-1} \lambda_t^{(T)} \right)^2 \gtrsim \frac{1}{T\varepsilon^2},$$

where $(a)$ is due to the Cauchy-Schwartz inequality. This is precisely the desired lower bound.

For the lower bound against the entire class of FOI, there are some other technical challenges arising from the fact that the coefficients $\lambda_i^{(t)}$'s not only depend on $t$, but also could take negative values. We need a more elaborate lower bound construction, as we explain in the subsequent subsections.

## D.2 Helper function

Before actual proofs, we introduce a helper function that we will use throughout the proofs of FOI lower bounds. Let $\chi : \mathbb{R} \to \mathbb{R}$ is a non-smooth convex function defined as $\chi(x) := \max\{x, 0\}$ and the subgradients are defined as

$$\nabla_x \chi(x) := \begin{cases} +1, & \text{if } x \geq 0, \\ 0, & \text{if } x < y. \end{cases} \tag{D.5}$$

The choice of subgradient $+1$ at the origin will play a crucial role in the later proofs. For $\boldsymbol{x}, \boldsymbol{y}, \boldsymbol{z} \in \mathbb{R}^T$ and $v \in \mathbb{R}$, let $\mathcal{G} : (\boldsymbol{x}, \boldsymbol{y}, \boldsymbol{z}) \in \mathbb{R}^{3T} \to \mathbb{R}$ be defined as

$$\mathcal{G}(\boldsymbol{x}, \boldsymbol{y}, \boldsymbol{z}) := \max\{0, \ \mathcal{K}(\boldsymbol{x}, \boldsymbol{y}, \boldsymbol{z})\} \quad \text{and} \tag{D.6}$$

$$\mathcal{K}(\boldsymbol{x}, \boldsymbol{y}, \boldsymbol{z}) := \max_{i=1,\dots,T} \left\{ \chi(\boldsymbol{y}[i]) + \sum_{j=1}^{i-1} \frac{|\boldsymbol{x}[j]|}{2^{j-1}} + \frac{\boldsymbol{x}[i]}{2^{i-1}}, \ \chi(\boldsymbol{z}[i]) + \sum_{j=1}^{i-1} \frac{|\boldsymbol{x}[j]|}{2^{j-1}} - \frac{\boldsymbol{x}[i]}{2^{i-1}} \right\}. \tag{D.7}$$

Then $\mathcal{G}$ is clearly convex, as it is the maximum of convex functions.

We specify the subgradients of $\mathcal{G}$ as follows: for all max terms in (D.8), we get the subgradient of the first argument that achieves the maximum. Then $\mathcal{G}$ is $O(1)$-Lipschitz: for any $\boldsymbol{x}, \boldsymbol{y}, \boldsymbol{z} \in \mathbb{R}^T$,

$$\|\nabla \mathcal{G}(\boldsymbol{x}, \boldsymbol{y}, \boldsymbol{z})\|^2 \leq 1 + \sum_{j=1}^{T} \left(\frac{1}{2^{j-1}}\right)^2 \leq 1 + \sum_{j=1}^{\infty} \frac{1}{4^{j-1}} \leq 1 + 4/3 \,.$$

## D.3 Stochastic inexact gradient model

**Theorem 13. (Lower Bound)** *Let $\varepsilon > 0$ and $T$ be a given number of iterations. There exists a $O(1)$-Lipschitz (nonsmooth) convex function $f : \mathbb{R}^{O(T)} \to \mathbb{R}$ with a stochastic inexact gradient model such that any FOI algorithm $\mathcal{A}$ that satisfies $|\lambda_0^{(T)}| \leq O\left(\left|\sum_{t=1}^{T-1} \lambda_t^{(T)}\right|\right)$ and starts at $\boldsymbol{x}_0 = \boldsymbol{0}$ has its $(\varepsilon, \delta)$-deviation lower bounded by $\Omega(\frac{1}{T\varepsilon^2})$.*

*Proof.* For $\boldsymbol{x} = (\boldsymbol{x}^{\mathrm{err}}, \boldsymbol{y}, \boldsymbol{z}, w)$ where $\boldsymbol{x}^{\mathrm{err}}, \boldsymbol{y}, \boldsymbol{z} \in \mathbb{R}^T$ and $w \in \mathbb{R}$, consider the cost

$$f(\boldsymbol{x}^{\mathrm{err}}, \boldsymbol{y}, \boldsymbol{z}, w) = \mathcal{G}(\boldsymbol{x}^{\mathrm{err}}, \boldsymbol{y}, \boldsymbol{z}) + \underbrace{2\varepsilon \cdot \max\{w + 1, 0\}}_{=:\ell(w)}, \tag{D.8}$$

where $\mathcal{G}$ is defined in (D.6). Then, $f$ is convex since both $\mathcal{G}$ and $\ell$ are convex, and $O(1)$-Lipschitz since both $\mathcal{G}$ and $\ell$ are $O(1)$-Lipschitz.

For intuition, we describe the role of each coordinate as we did in the warm-up section (Subsection D.1):

- The first $T$ coordinates, $\boldsymbol{x}^{\mathrm{err}} \in \mathbb{R}^T$, correspond to the part where the errors due to inexact oracle are added.

- The next $2T$ coordinates, $\boldsymbol{y}, \boldsymbol{z} \in \mathbb{R}^T$ will contribute to large deviation when there are errors in the gradients.

- The last coordinate, $w \in \mathbb{R}$, governs the overall cost; in order to achieve $\varepsilon$-accuracy, the optimization algorithm has to decrease coordinate $w$ by at least $1/2$.

We use the following notation throughout the proof: $\boldsymbol{x}_t = (\boldsymbol{x}_t^{\mathrm{err}}, \boldsymbol{y}_t, \boldsymbol{z}_t, w_t) \in (\mathbb{R}^T)^3 \times \mathbb{R}$. Consider the zero initialization $\boldsymbol{x}_0 = (\boldsymbol{0}, \boldsymbol{0}, \boldsymbol{0}, 0)$ and the following inexact gradient error for $t = 0, 1, 2, \dots, T-1$:

$$g(\boldsymbol{x}_t) = \nabla f(\boldsymbol{x}_t) + \delta r_t \cdot \boldsymbol{e}_{1+t} \quad \text{for } r_t \overset{iid}{\sim} \mathrm{Unif}\{\pm 1\}, \tag{D.9}$$

where $\boldsymbol{e}_j$ is the $j$-th coordinate vector. The following lemma characterizes the key feature of the above construction.

*Remark* 1. Note that Lemma 2 is the place where we use the following additional assumption that we made for the case of nonsmooth costs: "*for the case of nonsmooth costs, we additionally assume that the coefficient of the latest gradient is nonzero, i.e., $\lambda_{t-1}^{(t)} \neq 0$ for all t.*"

**Lemma 2.** *Under the inexact gradient (D.9), the subgradient $\nabla \mathcal{G}$ has the following properties:*

- *For each $t = 1, 2, \ldots, T-1$, there exists $i_t \in \{1, \ldots, t\}$ such that the following holds:*

$$
\begin{cases}
\frac{\partial}{\partial \boldsymbol{y}[i_t]} \mathcal{G}(\boldsymbol{x}_t^{\mathrm{err}}, \boldsymbol{y}_t, \boldsymbol{z}_t) = 1, & \frac{\partial}{\partial \boldsymbol{z}[i_t]} \mathcal{G}(\boldsymbol{x}_t^{\mathrm{err}}, \boldsymbol{y}_t, \boldsymbol{z}_t) = 0 \quad \text{with probability 1/2,} \\
\frac{\partial}{\partial \boldsymbol{y}[i_t]} \mathcal{G}(\boldsymbol{x}_t^{\mathrm{err}}, \boldsymbol{y}_t, \boldsymbol{z}_t) = 0, & \frac{\partial}{\partial \boldsymbol{z}[i_t]} \mathcal{G}(\boldsymbol{x}_t^{\mathrm{err}}, \boldsymbol{y}_t, \boldsymbol{z}_t) = 1 \quad \text{with probability 1/2.}
\end{cases}
\tag{D.10}
$$

  *Moreover, for $i \neq i_t$, $\frac{\partial}{\partial \boldsymbol{y}[i]} \mathcal{G}(\boldsymbol{x}_t^{\mathrm{err}}, \boldsymbol{y}_t, \boldsymbol{z}_t) = \frac{\partial}{\partial \boldsymbol{z}[i]} \mathcal{G}(\boldsymbol{x}_t^{\mathrm{err}}, \boldsymbol{y}_t, \boldsymbol{z}_t) = 0$.*

- *If $i_t \neq t$ (i.e., $i_t < t$), then $i_t = i_{t'}$ for some $t' < t$, and it holds that $\frac{\partial}{\partial \boldsymbol{y}[i_t]} \mathcal{G}(\boldsymbol{x}_t^{\mathrm{err}}, \boldsymbol{y}_t, \boldsymbol{z}_t) = \frac{\partial}{\partial \boldsymbol{y}[i_{t'}]} \mathcal{G}(\boldsymbol{x}_{t'}^{\mathrm{err}}, \boldsymbol{y}_{t'}, \boldsymbol{z}_{t'})$ and $\frac{\partial}{\partial \boldsymbol{z}[i_t]} \mathcal{G}(\boldsymbol{x}_t^{\mathrm{err}}, \boldsymbol{y}_t, \boldsymbol{z}_t) = \frac{\partial}{\partial \boldsymbol{z}[i_{t'}]} \mathcal{G}(\boldsymbol{x}_{t'}^{\mathrm{err}}, \boldsymbol{y}_{t'}, \boldsymbol{z}_{t'})$.*

*Proof of Lemma 2.* Let us recall the definition of $\mathcal{G}(\boldsymbol{x}, \boldsymbol{y}, \boldsymbol{z})$:

$$
\max \left\{ 0 \;,\; \max_{i=1,\ldots,T} \left\{ \chi(\boldsymbol{y}[i]) + \sum_{j=1}^{i-1} \frac{|\boldsymbol{x}[j]|}{2^{j-1}} + \frac{\boldsymbol{x}[i]}{2^{i-1}}, \;\; \chi(\boldsymbol{z}[i]) + \sum_{j=1}^{i-1} \frac{|\boldsymbol{x}[j]|}{2^{j-1}} - \frac{\boldsymbol{x}[i]}{2^{i-1}} \right\} \right\} .
\tag{D.11}
$$

From this, it is clear that there must be at most one $i \in \{1, \ldots, T\}$ for which either $\frac{\partial}{\partial \boldsymbol{y}[i]} \mathcal{G}(\boldsymbol{x}_t^{\mathrm{err}}, \boldsymbol{y}_t, \boldsymbol{z}_t) \neq 0$ or $\frac{\partial}{\partial \boldsymbol{z}[i]} \mathcal{G}(\boldsymbol{x}_t^{\mathrm{err}}, \boldsymbol{y}_t, \boldsymbol{z}_t) \neq 0$. This proves the "*Moreover, for $i \neq i_t$, $\frac{\partial}{\partial \boldsymbol{y}[i]} \mathcal{G}(\boldsymbol{x}_t^{\mathrm{err}}, \boldsymbol{y}_t, \boldsymbol{z}_t) = \frac{\partial}{\partial \boldsymbol{z}[i]} \mathcal{G}(\boldsymbol{x}_t^{\mathrm{err}}, \boldsymbol{y}_t, \boldsymbol{z}_t) = 0$*" part of the first bullet point.

Next, we prove the expression (D.10). We begin with $t = 1$. Since $g(\boldsymbol{x}_0) = \nabla f(\boldsymbol{x}_0) + \delta r_0 \cdot \boldsymbol{e}_1$ and $\lambda_0^{(1)} \neq 0$, it follows that $\boldsymbol{x}_1^{\mathrm{err}}[1] \neq 0$. Then, we claim that the first bullet point holds for $t = 1$. Since we know $\boldsymbol{x}_1^{\mathrm{err}}[1] \neq 0$ and $\boldsymbol{x}_1^{\mathrm{err}}[2], \ldots, \boldsymbol{x}_1^{\mathrm{err}}[T] = 0$, for $\boldsymbol{x}_1$, the maximum in (D.11) is achieved by $i = 1$. This implies that $i_1 = 1$. Moreover, depending on the sign of $r_0$, we either have $\frac{\partial}{\partial \boldsymbol{y}[1]} \mathcal{G}(\boldsymbol{x}_1^{\mathrm{err}}, \boldsymbol{y}_1, \boldsymbol{z}_1) = 1$ or $\frac{\partial}{\partial \boldsymbol{z}[1]} \mathcal{G}(\boldsymbol{x}_1^{\mathrm{err}}, \boldsymbol{y}_1, \boldsymbol{z}_1) = 1$ with equal probability. Thus, (D.10) holds for $t = 1$.

Next, consider $t > 1$. Since $g(\boldsymbol{x}_{t-1}) = \nabla f(\boldsymbol{x}_{t-1}) + \delta r_{t-1} \cdot \boldsymbol{e}_t$ and $\lambda_{t-1}^{(t)} \neq 0$, it follows that $\boldsymbol{x}_t^{\mathrm{err}}[t] \neq 0$. Moreover, we know $\boldsymbol{x}_t^{\mathrm{err}}[t+1], \ldots, \boldsymbol{x}_t^{\mathrm{err}}[T] = 0$. Hence, we have the following two scenarios:

- **Case 1:** $\boldsymbol{y}[i], \boldsymbol{z}[i] \leq 0$ for all $i \in \{1, 2, \ldots, t-1\}$. Note that this hold—for instance—if the coefficients FOI are all non-negative, i.e., $\lambda_i^{(t)} \geq 0$ of for all $i \in \{1, 2, \ldots, t-1\}$ (most first order optimization algorithms usually follow this). In that case, the maximum in (D.11) is achieved by $i = t$ This implies that $i_t = t$. Moreover, depending on the sign of $r_{t-1}$, we either have $\frac{\partial}{\partial \boldsymbol{y}[t]} \mathcal{G}(\boldsymbol{x}_t^{\mathrm{err}}, \boldsymbol{y}_t, \boldsymbol{z}_t) = 1$ or $\frac{\partial}{\partial \boldsymbol{z}[t]} \mathcal{G}(\boldsymbol{x}_t^{\mathrm{err}}, \boldsymbol{y}_t, \boldsymbol{z}_t) = 1$ with equal probability. Thus, again (D.10) holds for $t$.

- **Case 2:** Somehow FOI chooses to follow positive gradient directions (which is unlikely in practice) and it happens that $\boldsymbol{y}[i] > 0$ or $\boldsymbol{z}[i] > 0$ for some $i \in \{1, 2, \ldots, t-1\}$. In such a case, the maximum in (D.11) could be achieved by $i \in \{1, 2, \ldots, t-1\}$, i.e., $i_t \in \{1, 2, \ldots, t-1\}$. Then it must be that $\boldsymbol{y}[i_t] > 0$ or $\boldsymbol{z}[i_t] > 0$. This can happen only if $i_t = i_{t'}$ for some $t' < t$. Hence, it follows that $\frac{\partial}{\partial \boldsymbol{y}[i_t]} \mathcal{G}(\boldsymbol{x}_t^{\mathrm{err}}, \boldsymbol{y}_t, \boldsymbol{z}_t) = \frac{\partial}{\partial \boldsymbol{y}[i_{t'}]} \mathcal{G}(\boldsymbol{x}_{t'}^{\mathrm{err}}, \boldsymbol{y}_{t'}, \boldsymbol{z}_{t'})$ and $\frac{\partial}{\partial \boldsymbol{z}[i_t]} \mathcal{G}(\boldsymbol{x}_t^{\mathrm{err}}, \boldsymbol{y}_t, \boldsymbol{z}_t) = \frac{\partial}{\partial \boldsymbol{z}[i_{t'}]} \mathcal{G}(\boldsymbol{x}_{t'}^{\mathrm{err}}, \boldsymbol{y}_{t'}, \boldsymbol{z}_{t'})$. This proves the second bullet point in the statement. In particular, (D.10) holds for $t$.

This completes the proof of Lemma 2. $\square$

Now we use Lemma 2 to prove Theorem 13. From the construction (D.8), we know that $f(\boldsymbol{x}_0) = 2\varepsilon$. In order to achieve $\varepsilon$-accuracy, we need $w_T \leq -1/2$. Note that $\frac{\partial}{\partial w} \ell(w_t) = 2\varepsilon$ for all $t \geq 0$. From

the fact that $w_T \leq -\frac{1}{2}$, it follows that

$$\sum_{t=0}^{T-1} \lambda_t^{(T)} \frac{\partial}{\partial w} \ell(v_t, w_t) \geq \frac{1}{2} \quad \Longleftrightarrow \quad \sum_{t=0}^{T-1} \lambda_t^{(T)} \geq \frac{1}{4\varepsilon} .$$

Now using the assumption that $|\lambda_0^{(T)}| \leq O\left(\left|\sum_{t=1}^{T-1} \lambda_t^{(T)}\right|\right)$, we obtain

$$\left|\sum_{t=1}^{T-1} \lambda_t^{(T)}\right| \gtrsim |\lambda_0^{(T)}| + \left|\sum_{t=1}^{T-1} \lambda_t^{(T)}\right| \geq \left|\sum_{t=0}^{T-1} \lambda_t^{(T)}\right| \geq \frac{1}{4\varepsilon} ,$$

which leads to the following conditon:

$$\left|\sum_{t=1}^{T-1} \lambda_t^{(T)}\right| \geq \Omega\left(\frac{1}{\varepsilon}\right) \tag{D.12}$$

This condition is analogous to (D.2) from Subsection D.1. Now to better illustrate our proof strategy for the remaining part, we first consider a special case.

**Warm-up: proof for the special case.** As a warm-up, we first consider the special case where in the definition of FOI, all the coefficients $\lambda_i^{(t)}$ are non-negative, i.e.,

$$\boldsymbol{x}_t = \boldsymbol{x}_0 - \sum_{i=0}^{t-1} \lambda_i^{(t)} g(\boldsymbol{x}_i) \quad \text{for some } \lambda_i^{(t)} \geq 0, \ i = 0, \ldots, t-1, \tag{D.13}$$

Then this case belongs to **Case 1** in the proof of Lemma 2. As a consequence, $i_t = t$ in Lemma 2 and the following conclusion holds:

- For each $t = 1, 2, \ldots, T-1$,

$$\begin{cases} \frac{\partial}{\partial \boldsymbol{y}[t]} \mathcal{G}(\boldsymbol{x}_t^{\text{err}}, \boldsymbol{y}_t, \boldsymbol{z}_t) = 1, & \frac{\partial}{\partial \boldsymbol{z}[t]} \mathcal{G}(\boldsymbol{x}_t^{\text{err}}, \boldsymbol{y}_t, \boldsymbol{z}_t) = 0 \quad \text{with probability } 1/2, \\ \frac{\partial}{\partial \boldsymbol{y}[t]} \mathcal{G}(\boldsymbol{x}_t^{\text{err}}, \boldsymbol{y}_t, \boldsymbol{z}_t) = 0, & \frac{\partial}{\partial \boldsymbol{z}[t]} \mathcal{G}(\boldsymbol{x}_t^{\text{err}}, \boldsymbol{y}_t, \boldsymbol{z}_t) = 1 \quad \text{with probability } 1/2. \end{cases} \tag{D.14}$$

Moreover, for $i \neq t$, $\frac{\partial}{\partial \boldsymbol{y}[i]} \mathcal{G}(\boldsymbol{x}_t^{\text{err}}, \boldsymbol{y}_t, \boldsymbol{z}_t) = \frac{\partial}{\partial \boldsymbol{z}[i]} \mathcal{G}(\boldsymbol{x}_t^{\text{err}}, \boldsymbol{y}_t, \boldsymbol{z}_t) = 0$.

We use (D.14) to lower bound $\mathbb{E}\left\|\boldsymbol{x}_T - \mathbb{E}[\boldsymbol{x}_T]\right\|^2$. From (D.14), it holds that for $t = 1, 2, \ldots, T$,

$$\boldsymbol{y}_T[t] = \begin{cases} \lambda_t^{(T)}, & \text{with probability } 1/2, \\ 0, & \text{with probability } 1/2 . \end{cases}$$

Hence, we have

$$\mathbb{E}\left\|\boldsymbol{y}_T - \mathbb{E}[\boldsymbol{y}_T]\right\|^2 \geq \sum_{t=1}^{T-1} \mathbb{E}\left(\boldsymbol{y}_T[t] - \mathbb{E}\,\boldsymbol{y}_T[t]\right)^2 = \sum_{t=1}^{T-1} \mathbb{E}\left(\boldsymbol{y}_T[t] - \frac{1}{2}\lambda_t^{(T)}\right)^2 = \frac{1}{4}\sum_{t=1}^{T-1}(\lambda_t^{(T)})^2$$

$$\overset{(a)}{\geq} \frac{1}{4} \cdot \frac{1}{T-1} \cdot \left(\sum_{t=1}^{T-1} \lambda_t^{(T)}\right)^2 \overset{\text{(D.12)}}{\gtrsim} \frac{1}{4} \cdot \frac{1}{T-1} \cdot \left(\frac{1}{\varepsilon}\right)^2 \gtrsim \frac{1}{T\varepsilon^2} .$$

Here $(a)$ follows from Cauchy-Schwarz inequality. Therefore, we get the desired deviation bound as follows:

$$\mathbb{E}\left\|\boldsymbol{x}_T - \mathbb{E}[\boldsymbol{x}_T]\right\|^2 \geq \mathbb{E}\left\|\boldsymbol{y}_T - \mathbb{E}[\boldsymbol{y}_T]\right\|^2 \gtrsim \frac{1}{T\varepsilon^2} .$$

**The proof for the general case.**  Now we consider the case of general FOI where the coefficients $\lambda_i^{(t)}$ are not necessarily non-negative. With $i_t$ defined in the statement of Lemma 2, let

$$\mathcal{I} := \{i_t \in \{1, 2, \ldots, T\} \ : \ t = 1, 2, \ldots, T-1\} .$$

Then Lemma 2 ensures that that for each $i \in \mathcal{I}$,

$$\boldsymbol{y}_T[i] = \begin{cases} \sum_{\substack{t=1,2,\ldots,T-1 \\ \text{s.t. } i_t=i}} \lambda_t^{(T)}, & \text{with probability } 1/2, \\ 0, & \text{with probability } 1/2 . \end{cases}$$

Hence, we have

$$\mathbb{E}\left\|\boldsymbol{y}_T - \mathbb{E}[\boldsymbol{y}_T]\right\|^2 \geq \sum_{t=1}^{T-1} \mathbb{E}\left(\boldsymbol{y}_T[t] - \mathbb{E}\,\boldsymbol{y}_T[t]\right)^2 = \frac{1}{4} \sum_{i \in \mathcal{I}} \left( \sum_{\substack{t=1,2,\ldots,T-1 \\ \text{s.t. } i_t=i}} \lambda_t^{(T)} \right)^2$$

$$\overset{(a)}{\geq} \frac{1}{4} \cdot \frac{1}{|\mathcal{I}|} \cdot \left( \sum_{i \in \mathcal{I}} \sum_{\substack{t=1,2,\ldots,T-1 \\ \text{s.t. } i_t=i}} \lambda_t^{(T)} \right)^2 = \frac{1}{4} \cdot \frac{1}{|\mathcal{I}|} \cdot \left( \sum_{t=1}^{T-1} \lambda_t^{(T)} \right)^2 \overset{(\text{D.12})}{\gtrsim} \frac{1}{T\varepsilon^2}.$$

Here $(a)$ follows from Cauchy-Schwarz inequality. Therefore, we get the desired deviation bound as follows:

$$\mathbb{E}\left\|\boldsymbol{x}_T - \mathbb{E}[\boldsymbol{x}_T]\right\|^2 \geq \mathbb{E}\left\|\boldsymbol{y}_T - \mathbb{E}[\boldsymbol{y}_T]\right\|^2 \gtrsim \frac{1}{T\varepsilon^2}.$$

$$\square$$

## D.4  Non-stochastic inexact gradient model

**Theorem 14. (Lower Bound)** *Let $\varepsilon > 0$, and $T$ be a given number of iterations. There exists a $O(1)$-Lipschitz and nonsmooth convex function $f : \mathbb{R}^{O(T)} \to \mathbb{R}$ with a non-stochastic inexact gradient oracle such that for any FOI algorithm $\mathcal{A}$ that satisfies $|\lambda_0^{(T)}| \leq O\left(\left|\sum_{t=1}^{T-1} \lambda_t^{(T)}\right|\right)$ and starts at $\boldsymbol{x}_0 = \mathbf{0}$ has a minimum of $(\varepsilon, \delta)$-deviation of $\Omega(\frac{1}{T\varepsilon^2} + \frac{\delta^2}{\varepsilon^2})$.*

*Proof.* We consider an almost identical construction to the one considered in the proof of Theorem 13, namely (D.8). The only difference is that now we add an extra dummy coordinate, namely the $(3T+2)$-th coordinate, which does not appear in the cost. Let us denote this dummy coordinate by $u$. Concretely, we consider the following cost: For $\boldsymbol{x} = (\boldsymbol{x}^{\text{err}}, \boldsymbol{y}, \boldsymbol{z}, w, u)$ where $\boldsymbol{x}^{\text{err}}, \boldsymbol{y}, \boldsymbol{z} \in \mathbb{R}^T$ and $w, u \in \mathbb{R}$, consider the cost

$$f(\boldsymbol{x}^{\text{err}}, \boldsymbol{y}, \boldsymbol{z}, w, u) = \mathcal{G}(\boldsymbol{x}^{\text{err}}, \boldsymbol{y}, \boldsymbol{z}) + \underbrace{2\varepsilon \cdot \max\{w+1, 0\}}_{=:\ell(w)} . \tag{D.15}$$

We denote the iterates of FOI due to *exact* gradients by

$$\boldsymbol{x}_t^{\text{exact}} = ((\boldsymbol{x}_t^{\text{err}})^{\text{exact}}, \boldsymbol{y}_t^{\text{exact}}, \boldsymbol{z}_t^{\text{exact}}, w_t^{\text{exact}}, u_t^{\text{exact}}) .$$

Now we define the inexact gradient oracle. The noise in the inexact gradient oracle consists of two parts. For $t = 0, 1, 2, \ldots, T-1$,

$$g(\boldsymbol{x}_t) = \nabla f(\boldsymbol{x}_t) + \frac{\delta}{\sqrt{2}} \boldsymbol{e}_{1+t} + \frac{\delta}{\sqrt{2}} \boldsymbol{e}_{3T+2} , \tag{D.16}$$

where $\boldsymbol{e}_j$ is the $j$-th coordinate vector. Note that the first part of the error is similar to the error for the stochastic error case, and the second part of the error is added to the dummy coordinate $u$.

Then analogous to Lemma 2, one can establish the following result. We skip the proof since it is very analogous to that of Lemma 2.

**Corollary 1.** *Under the inexact gradient* (D.9)*, the subgradient* $\nabla \mathcal{G}$ *has the following properties:*

- *For each $t = 1, 2, \ldots, T-1$, there exists $i_t \in \{1, \ldots, t\}$ such that either one of the following holds:*

$$
\begin{cases}
\frac{\partial}{\partial \boldsymbol{y}[i_t]} \mathcal{G}(\boldsymbol{x}_t^{\mathrm{err}}, \boldsymbol{y}_t, \boldsymbol{z}_t) = 1, & \frac{\partial}{\partial \boldsymbol{z}[i_t]} \mathcal{G}(\boldsymbol{x}_t^{\mathrm{err}}, \boldsymbol{y}_t, \boldsymbol{z}_t) = 0, \quad \text{or} \\
\frac{\partial}{\partial \boldsymbol{y}[i_t]} \mathcal{G}(\boldsymbol{x}_t^{\mathrm{err}}, \boldsymbol{y}_t, \boldsymbol{z}_t) = 0, & \frac{\partial}{\partial \boldsymbol{z}[i_t]} \mathcal{G}(\boldsymbol{x}_t^{\mathrm{err}}, \boldsymbol{y}_t, \boldsymbol{z}_t) = 1.
\end{cases}
$$

  *Moreover, for $i \neq i_t$, $\frac{\partial}{\partial \boldsymbol{y}[i]} \mathcal{G}(\boldsymbol{x}_t^{\mathrm{err}}, \boldsymbol{y}_t, \boldsymbol{z}_t) = \frac{\partial}{\partial \boldsymbol{z}[i]} \mathcal{G}(\boldsymbol{x}_t^{\mathrm{err}}, \boldsymbol{y}_t, \boldsymbol{z}_t) = 0.$*

- *If $i_t \neq t$ (i.e., $i_t < t$), then $i_t = i_{t'}$ for some $t' < t$, and it holds that $\frac{\partial}{\partial \boldsymbol{y}[i_t]} \mathcal{G}(\boldsymbol{x}_t^{\mathrm{err}}, \boldsymbol{y}_t, \boldsymbol{z}_t) = \frac{\partial}{\partial \boldsymbol{y}[i_{t'}]} \mathcal{G}(\boldsymbol{x}_{t'}^{\mathrm{err}}, \boldsymbol{y}_{t'}, \boldsymbol{z}_{t'})$ and $\frac{\partial}{\partial \boldsymbol{z}[i_t]} \mathcal{G}(\boldsymbol{x}_t^{\mathrm{err}}, \boldsymbol{y}_t, \boldsymbol{z}_t) = \frac{\partial}{\partial \boldsymbol{z}[i_{t'}]} \mathcal{G}(\boldsymbol{x}_{t'}^{\mathrm{err}}, \boldsymbol{y}_{t'}, \boldsymbol{z}_{t'})$.*

The rest of the proof is similar to that of Theorem 13. From the construction (D.15), we know that $f(\boldsymbol{x}_0) = 2\varepsilon$. In order to achieve $\varepsilon$-accuracy, one can similarly deduce that (D.12) holds.

With $i_t$ defined in the statement of Corollary 1, let

$$
\mathcal{I} := \{i_t \in \{1, 2, \ldots, T\} \,:\, t \in \mathcal{T}\} \,.
$$

Then one can similarly argue using Corollary 1 that for each $i \in \mathcal{I}$, either

$$
\boldsymbol{y}_T[i] = \sum_{\substack{t=1,2,\ldots,T-1 \\ \text{s.t. } i_t=i}} \lambda_t^{(T)} \quad \text{or} \quad \boldsymbol{z}_T[i] = \sum_{\substack{t=1,2,\ldots,T-1 \\ \text{s.t. } i_t=i}} \lambda_t^{(T)} \,.
$$

This, together with the fact that $\boldsymbol{y}_T^{\mathrm{exact}} = \boldsymbol{0}$ and $\boldsymbol{z}_T^{\mathrm{exact}} = \boldsymbol{0}$, implies that

$$
\left\| \boldsymbol{y}_T - \boldsymbol{y}_T^{\mathrm{exact}} \right\|^2 + \left\| \boldsymbol{z}_T - \boldsymbol{z}_T^{\mathrm{exact}} \right\|^2 \geq \sum_{i \in \mathcal{I}} \left( \sum_{\substack{t=1,2,\ldots,T-1 \\ \text{s.t. } i_t=i}} \lambda_t^{(T)} \right)^2
$$

$$
\overset{(a)}{\geq} \frac{1}{|\mathcal{I}|} \cdot \left( \sum_{i \in \mathcal{I}} \sum_{\substack{t=1,2,\ldots,T-1 \\ \text{s.t. } i_t=i}} \lambda_t^{(T)} \right)^2 = \frac{1}{4} \cdot \frac{1}{|\mathcal{I}|} \cdot \left( \sum_{t=1}^{T-1} \lambda_t^{(T)} \right)^2 \overset{\text{(D.12)}}{\gtrsim} \frac{1}{T\varepsilon^2} \,.
$$

On the other hand, since $u_T^{\mathrm{exact}} = 0$, the deviation in the dummy coordinate can be lower bounded as follows:

$$
|u_T - u_T^{\mathrm{exact}}|^2 = \left( \frac{\delta}{\sqrt{2}} \cdot \sum_{t=0}^{T-1} \lambda_t^{(T)} \right)^2 \gtrsim \frac{\delta^2}{\varepsilon^2} \,.
$$

Therefore, combining all together, we get

$$
\left\| \boldsymbol{x}_T - \boldsymbol{x}_T^{\mathrm{exact}} \right\|^2 \geq \left\| \boldsymbol{y}_T - \boldsymbol{y}_T^{\mathrm{exact}} \right\|^2 + \left\| \boldsymbol{z}_T - \boldsymbol{z}_T^{\mathrm{exact}} \right\|^2 + \left\| u_T - u_T^{\mathrm{exact}} \right\|^2
$$

$$
\gtrsim \frac{1}{T\varepsilon^2} + \frac{\delta^2}{\varepsilon^2} \,,
$$

as desired. $\qquad \square$

## D.5 Inexact initialization model

**Theorem 15. (Lower Bound)** *Let $\varepsilon > 0$ be a small constant, and $T$ be a given number of iterations. There exists a $O(1)$-Lipschitz (nonsmooth) convex function $f : \mathbb{R}^{O(T)} \to \mathbb{R}$ such that for any FOI algorithm $\mathcal{A}$ the $(\varepsilon, \delta)$-deviation lower bounded by $\Omega(\frac{1}{T\varepsilon^2} + \delta^2)$ w.r.t. the reference point $\boldsymbol{x}_0^{\mathrm{ref}} = \boldsymbol{0}$.*

*Proof.* For the initialization error model, we use a simpler construction. For $\boldsymbol{x} = (\boldsymbol{x}^{\mathrm{err}}, \boldsymbol{y}, w, u) \in (\mathbb{R}^T)^2 \times \mathbb{R}^2 \to \mathbb{R}$, consider the cost defined as

$$
f(\boldsymbol{x}^{\mathrm{err}}, \boldsymbol{y}, w) = \max\{0, \, \boldsymbol{x}^{\mathrm{err}}[1] + \boldsymbol{y}[1], \, \ldots, \, \boldsymbol{x}^{\mathrm{err}}[T] + \boldsymbol{y}[T]\} + 2\varepsilon \cdot \max\{w+1, 0\} \,.
$$

Here $u$ is a dummy coordinate that does not appear in the cost. For both max terms above, we consider the subgradient that outputs the gradient of the first argument that achieves the maximum. Then, $f$ is clearly $O(1)$-Lipschitz.

We set the reference and inexact intializations as follows:

$$\boldsymbol{x}_0^{\mathsf{ref}} = (0, 0, \ldots, 0) \quad \text{and} \quad \boldsymbol{x}_0 = (\underbrace{\delta/\sqrt{2T}, \cdots, \delta/\sqrt{2T}}_{\text{first } T}, \underbrace{0, \cdots, 0}_{\text{second } T}, 0, \delta/\sqrt{2}). \tag{D.17}$$

Following the previous notations, we write the iterates as $\boldsymbol{x}_t = (\boldsymbol{x}_t^{\mathsf{err}}, \boldsymbol{y}_t, w_t) \in (\mathbb{R}^T)^2 \times \mathbb{R}$. Moreover, we will write the iterates corresponding to the reference initialization as $\boldsymbol{x}_t^{\mathsf{ref}} = ((\boldsymbol{x}_t^{\mathsf{err}})^{\mathsf{ref}}, \boldsymbol{y}_t^{\mathsf{ref}}, w_t^{\mathsf{ref}})$.

From the fact that the algorithm has to achieve $\varepsilon$-suboptimality, it follows that $w_T \leq -1/2$. On the other hand, from the inexact initialization (D.17), it holds that for all $t = 0, \ldots, T-1$,

$$\exists i_t \in \{1, \ldots, T\} \text{ s.t. } \begin{cases} \frac{\partial}{\partial \boldsymbol{y}[i_t]} f(\boldsymbol{x}_t^{\mathsf{err}}, \boldsymbol{y}_t, w_t) = 1 \quad \text{and} \\ \frac{\partial}{\partial \boldsymbol{y}[i]} f(\boldsymbol{x}_t^{\mathsf{err}}, \boldsymbol{y}_t, w_t) = 0 \quad \text{for } i \neq i_t. \end{cases}$$

Hence, it holds that $\frac{1}{2\varepsilon} w_T = \sum_{i=1}^T \boldsymbol{y}_T[i]$. Since we know $\boldsymbol{y}_T^{\mathsf{ref}} = (0, 0, \ldots, 0)$ and $u^{\mathsf{ref}} = 0$, the following deviation lower bound holds:

$$\left\| \boldsymbol{x}_T - \boldsymbol{x}_T^{\mathsf{ref}} \right\|^2 \geq \left\| \boldsymbol{y}_T - \boldsymbol{y}_T^{\mathsf{ref}} \right\|^2 + \left\| u_T - u_T^{\mathsf{ref}} \right\|^2 = \sum_{i=1}^T (\boldsymbol{y}_T[i])^2 + u_T^2$$

$$\geq \frac{1}{T} \cdot \left( \sum_{i=1}^T \boldsymbol{y}_T[i] \right)^2 + \frac{\delta^2}{2} = \frac{1}{T} \cdot \frac{1}{4\varepsilon^2} \cdot (w_T)^2 + \frac{\delta^2}{2} \gtrsim \frac{1}{T\varepsilon^2} + \delta^2$$

This completes the proof. $\qquad \square$

## D.6 Stochastic inexact gradient model (strongly-convex costs)

**Theorem 16. (Lower Bound)** *Let $\varepsilon > 0$ and $T$ be a given number of iterations. There exists a $O(1)$-Lipschitz (nonsmooth) and $\mu$-strongly convex function $f : \mathbb{R}^{O(T)} \to \mathbb{R}$ with a stochastic inexact gradient model such that any FOI algorithm $\mathcal{A}$ that starts at $\boldsymbol{x}_0 = \boldsymbol{0}$ has its $(\varepsilon, \delta)$-deviation lower bounded by $\Omega(\frac{1}{T\mu^2} \wedge \frac{\varepsilon}{\mu})$.*

*Proof.* For $\boldsymbol{x} = (\boldsymbol{x}^{\mathsf{err}}, \boldsymbol{y}, \boldsymbol{z}, w) \in (\mathbb{R}^T)^3 \times \mathbb{R}^2$, consider the cost defined as

$$f(\boldsymbol{x}^{\mathsf{err}}, \boldsymbol{y}, \boldsymbol{z}, w) = \underbrace{\mathcal{G}(\boldsymbol{x}^{\mathsf{err}} + \delta \boldsymbol{e}_1, \boldsymbol{y}, \boldsymbol{z}) + \frac{\mu}{2} \left\| (\boldsymbol{x}^{\mathsf{err}}, \boldsymbol{y}, \boldsymbol{z}) \right\|^2}_{=:\mathcal{G}^\mu(\boldsymbol{x}^{\mathsf{err}}, \boldsymbol{y}, \boldsymbol{z})} + \underbrace{w + \frac{\mu}{2} w^2}_{=:\ell^\mu(w)}. \tag{D.18}$$

We consider the same inexact gradient oracle defined in (D.9). Then similarly to Lemma 2, it holds that for each $t = 1, 2, \ldots, T-1$, there exists $i_t \in \{1, \ldots, t\}$ such that the following holds:

$$\begin{cases} \frac{\partial}{\partial \boldsymbol{y}[i_t]} \mathcal{G}(\boldsymbol{x}_t^{\mathsf{err}}, \boldsymbol{y}_t, \boldsymbol{z}_t, v_t) = 1, \quad \frac{\partial}{\partial \boldsymbol{z}[i_t]} \mathcal{G}(\boldsymbol{x}_t^{\mathsf{err}}, \boldsymbol{y}_t, \boldsymbol{z}_t, v_t) = 0 \quad \text{with probability } 1/2, \\ \frac{\partial}{\partial \boldsymbol{y}[i_t]} \mathcal{G}(\boldsymbol{x}_t^{\mathsf{err}}, \boldsymbol{y}_t, \boldsymbol{z}_t, v_t) = 0, \quad \frac{\partial}{\partial \boldsymbol{z}[i_t]} \mathcal{G}(\boldsymbol{x}_t^{\mathsf{err}}, \boldsymbol{y}_t, \boldsymbol{z}_t, v_t) = 1 \quad \text{with probability } 1/2. \end{cases} \tag{D.19}$$

Moreover, for $i \neq i_t$, $\frac{\partial}{\partial \boldsymbol{y}[i]} \mathcal{G}(\boldsymbol{x}_t^{\mathsf{err}}, \boldsymbol{y}_t, \boldsymbol{z}_t, v_t) = \frac{\partial}{\partial \boldsymbol{z}[i]} \mathcal{G}(\boldsymbol{x}_t^{\mathsf{err}}, \boldsymbol{y}_t, \boldsymbol{z}_t, v_t) = 0$. Throughout the rest of the proof, we use the following notations:

$$\begin{cases} \boldsymbol{g}_t^{\boldsymbol{y}} = (\boldsymbol{g}_t^{\boldsymbol{y}}[1], \boldsymbol{g}_t^{\boldsymbol{y}}[2], \ldots, \boldsymbol{g}_t^{\boldsymbol{y}}[T]) := \nabla_{\boldsymbol{y}} \mathcal{G}^\mu(\boldsymbol{x}_t^{\mathsf{err}}, \boldsymbol{y}_t, \boldsymbol{z}_t, v_t), \\ \boldsymbol{g}_t^{\boldsymbol{z}} = (\boldsymbol{g}_t^{\boldsymbol{z}}[1], \boldsymbol{g}_t^{\boldsymbol{z}}[2], \ldots, \boldsymbol{g}_t^{\boldsymbol{z}}[T]) := \nabla_{\boldsymbol{z}} \mathcal{G}^\mu(\boldsymbol{x}_t^{\mathsf{err}}, \boldsymbol{y}_t, \boldsymbol{z}_t, v_t) \\ g_t^w := \frac{\partial \ell^\mu}{\partial w}(w_t) \end{cases}$$

We prove the following crucial result for the proof.

**Lemma 3.** *For each $t = 0, 1, 2, \ldots, T$, the output of a FOI algorithm satisfies the following:*

$$w_t = \sum_{i=1}^T \boldsymbol{y}_t[i] + \boldsymbol{z}_t[i] \quad \text{and} \quad g_t^w = g_t^v + \sum_{i=1}^T \boldsymbol{g}_t^{\boldsymbol{y}}[i] + \boldsymbol{g}_t^{\boldsymbol{z}}[i]. \tag{D.20}$$

*Proof.* We prove by induction on $t$. We first prove the statement for $t = 0$. Recall the definition $\mathcal{G}^\mu(\boldsymbol{x}^{\mathrm{err}}, \boldsymbol{y}, \boldsymbol{z}) := \mathcal{G}(\boldsymbol{x}^{\mathrm{err}} + \delta \boldsymbol{e}_1, \boldsymbol{y}, \boldsymbol{z}) + \frac{\mu}{2} \|(\boldsymbol{x}^{\mathrm{err}}, \boldsymbol{y}, \boldsymbol{z})\|^2$. Since the first coordinate of $\boldsymbol{x}^{\mathrm{err}}$ is $\delta$, it follows that $g_0^w = 1$, $\boldsymbol{g}_0^{\boldsymbol{y}} = \boldsymbol{e}_1$ and $\boldsymbol{g}_0^{\boldsymbol{z}} = \boldsymbol{0}$. Hence, the statement holds for $t = 0$.

Assume that the conclusion holds for some $t$. We will first show that $w_{t+1} = \sum_{i=1}^{T} \boldsymbol{y}_{t+1}[i] + \boldsymbol{z}_{t+1}[i]$. Using the definition of FOI together with the inductive hypothesis, we have

$$
\begin{aligned}
w_{t+1} &= -\sum_{j=0}^{t} \lambda_j^{(t+1)} g_j^w = -\sum_{j=0}^{t} \lambda_j^{(t+1)} \left( \sum_{i=1}^{T} \boldsymbol{g}_j^{\boldsymbol{y}}[i] + \boldsymbol{g}_j^{\boldsymbol{z}}[i] \right) \\
&= -\sum_{i=1}^{T} \sum_{j=0}^{t} \lambda_j^{(t+1)} \left( \boldsymbol{g}_j^{\boldsymbol{y}}[i] + \boldsymbol{g}_j^{\boldsymbol{z}}[i] \right) = \sum_{i=1}^{T} \boldsymbol{y}_{t+1}[i] + \boldsymbol{z}_{t+1}[i].
\end{aligned}
$$

Next, we show that $g_{t+1}^w = \sum_{i=1}^{T} \boldsymbol{g}_{t+1}^{\boldsymbol{y}}[i] + \boldsymbol{g}_{t+1}^{\boldsymbol{z}}[i]$. Using the conclusion we just proved, we obtain

$$
g_{t+1}^w = 1 + \mu w_{t+1} = 1 + \mu \left( \sum_{i=1}^{T} \boldsymbol{y}_{t+1}[i] + \boldsymbol{z}_{t+1}[i] \right) = \sum_{i=1}^{T} \left( \boldsymbol{g}_{t+1}^{\boldsymbol{y}}[i] + \boldsymbol{g}_{t+1}^{\boldsymbol{z}}[i] \right) ,
$$

where in the last equality, we used the fact that $\nabla \mathcal{G}^\mu(\boldsymbol{x}_t^{\mathrm{err}}, \boldsymbol{y}_t, \boldsymbol{z}_t)$ is zero except for a single coordinate that is equal to 1. $\qquad \square$

Note that in order for $f(\boldsymbol{x}_T)$ to achieve $\varepsilon$-accuracy, it must be that

$$
w_T \in \left[ -\frac{1}{2\mu} - O(\sqrt{\varepsilon}), -\frac{1}{2\mu} + O(\sqrt{\varepsilon}) \right] .
$$

Note that by symmetry, $\boldsymbol{y}_T[i] + \boldsymbol{z}_T[i]$ is a deterministic quantity. Lemma 2 ensures that

$$
\boldsymbol{y}_T[i] = \begin{cases} \boldsymbol{y}_T[i] + \boldsymbol{z}_T[i] & \text{with probability } 1/2, \\ 0 & \text{with probability } 1/2, \end{cases} \quad \text{for all } i = 2, \dots, T.
$$

Hence, the deviation $\mathbb{E} \|\boldsymbol{x}_T - \mathbb{E}[\boldsymbol{x}_T]\|^2$ is again lower bounded by $\frac{1}{4} \sum_{i=2}^{T} (\boldsymbol{y}_T[i] + \boldsymbol{z}_T[i])^2$. Since either $\boldsymbol{y}_T[i]$ or $\boldsymbol{z}_T[i]$ has to be zero, it follows that $(\boldsymbol{y}_T[i] + \boldsymbol{z}_T[i])^2 = (\boldsymbol{y}_T[i])^2 + (\boldsymbol{z}_T[i])^2$. From the $\varepsilon$-suboptimality of $\boldsymbol{x}_T$, it also holds that $\sum_{i=2}^{T} (\boldsymbol{y}_T[i])^2 + (\boldsymbol{z}_T[i])^2 \leq \frac{2}{\mu} \varepsilon$. Hence either $\sum_{i=2}^{T} (\boldsymbol{y}_T[i])^2 + (\boldsymbol{z}_T[i])^2 = \Omega(\frac{\varepsilon}{\mu})$ (in which case the deviation is lower bounded by $\Omega(\frac{\varepsilon}{\mu})$), or we use Lemma 3 to conclude that

$$
\begin{aligned}
\sum_{i=2}^{T} (\boldsymbol{y}_T[i] + \boldsymbol{z}_T[i])^2 &\overset{(a)}{\geq} \frac{1}{T-1} \cdot \left( \sum_{i=2}^{T} \boldsymbol{y}_T[i] + \boldsymbol{z}_T[i] \right)^2 \\
&= \frac{1}{T-1} \cdot (w_T - (\boldsymbol{y}_T[1] + \boldsymbol{z}_T[1]))^2 \gtrsim \frac{1}{T\mu^2} ,
\end{aligned}
$$

where $(a)$ follows form the Cauchy-Schwartz inequality, and the last inequality is due to the fact that $(\boldsymbol{y}_T[1] + \boldsymbol{z}_T[1])^2 \lesssim \varepsilon$. This completes the proof. $\qquad \square$

### D.7 Non-stochastic inexact gradient model (strongly-convex costs)

**Theorem 17.** **(Lower Bound)** *Let $\varepsilon > 0$ and $T$ be a given number of iterations. There exists a $O(1)$-Lipschitz (nonsmooth) and $\mu$-strongly convex function $f : \mathbb{R}^{O(T)} \to \mathbb{R}$ with a non-stochastic inexact gradient model such that any FOI algorithm $\mathcal{A}$ that starts at $\boldsymbol{x}_0 = \boldsymbol{0}$ has its $(\varepsilon, \delta)$-deviation lower bounded by $\Omega((\frac{1}{T\mu^2} + \frac{\delta^2}{\mu^2}) \wedge \frac{\varepsilon}{\mu})$.*

*Proof.* We consider the same construction as the one considered in the proof of Theorem 16. The only difference is that now we add an extra dummy coordinate $u$ (the $(3T + 2)$-th coordinate), and add $\frac{\mu}{2} u^2$ to the overall cost (so that the overall cost is still $\mu$-strongly convex). Following the previous

convention, we will write the iterate as $\boldsymbol{x}_t = (\boldsymbol{x}_t^{\text{err}}, \boldsymbol{y}_t, \boldsymbol{z}_t, w_t, u_t) \in (\mathbb{R}^T)^3 \times \mathbb{R}^3$. We define the inexact gradient as:

$$g(\boldsymbol{x}_t) = \nabla f(\boldsymbol{x}_t) + \frac{\delta}{\sqrt{2}} \boldsymbol{e}_{1+t} + \frac{\delta}{\sqrt{2}} \boldsymbol{e}_{3T+3} \,,$$

where $\boldsymbol{e}_j$ is the $j$-th coordinate vector. Then following the same argument as Lemma 3, it holds that

$$w_T = \sum_{i=1}^{T} \boldsymbol{y}_T[i] + \boldsymbol{z}_T[i] \quad \text{and} \quad \delta/\sqrt{2} \cdot w_T = u_T \,. \tag{D.21}$$

From the fact that $\boldsymbol{x}_T$ achieves $\varepsilon$-suboptimality, it must be that

$$w_T \in \left[ -\frac{1}{2\mu} - O(\sqrt{\varepsilon}), -\frac{1}{2\mu} + O(\sqrt{\varepsilon}) \right] \,.$$

Now the rest of the proof follows similarly to that of Theorem 16. Using the facts $\boldsymbol{y}_T^{\text{exact}} = 0$, $\boldsymbol{z}_T^{\text{exact}} = 0$ and $u_T^{\text{exact}} = 0$, we have the following lower bound on the deviation:

$$\left\| \boldsymbol{x}_T - \boldsymbol{x}_T^{\text{exact}} \right\|^2 \geq \left\| \boldsymbol{y}_T - \boldsymbol{y}_T^{\text{exact}} \right\|^2 + \left\| \boldsymbol{z}_T - \boldsymbol{z}_T^{\text{exact}} \right\|^2 + \left\| u_T - u_T^{\text{exact}} \right\|^2$$

$$\geq u_T^2 + \sum_{i=2}^{T} [(\boldsymbol{y}_T[i])^2 + (\boldsymbol{z}_T[i])^2]$$

From the $\varepsilon$-suboptimality of $\boldsymbol{x}_T$, it must be that $u_T^2 + \sum_{i=2}^{T} (\boldsymbol{y}_T[i])^2 + (\boldsymbol{z}_T[i])^2 \leq \frac{2}{\mu} \varepsilon$. Thus, either $u_T^2 + \sum_{i=2}^{T} (\boldsymbol{y}_T[i])^2 + (\boldsymbol{z}_T[i])^2 = \Omega(\frac{\varepsilon}{\mu})$ (in which case the deviation is lower bounded by $\Omega(\frac{\varepsilon}{\mu})$), or we use the fact that $(\sum_{i=2}^{T} \boldsymbol{y}_T[i] + \boldsymbol{z}_T[i])^2 \gtrsim \frac{1}{\mu^2}$ to conclude that

$$u_T^2 + \sum_{i=2}^{T} [(\boldsymbol{y}_T[i])^2 + (\boldsymbol{z}_T[i])^2] \geq \frac{\delta^2}{2} w_T^2 + \frac{1}{2(T-1)} \left( \sum_{i=1}^{T} \boldsymbol{y}_T[i] + \boldsymbol{z}_T[i] \right)^2$$

$$\gtrsim \delta^2 w_T^2 + \frac{1}{T} w_T^2 \gtrsim \frac{\delta^2}{\mu^2} + \frac{1}{T\mu^2} \,.$$

This completes the proof.

$\square$

## D.8 Inexact initialization model (strongly-convex costs)

**Theorem 18. (Lower Bound)** *Let $\varepsilon > 0$ and $T$ be a given number of iterations. There exists a $O(1)$-Lipschitz (nonsmooth) and $\mu$-strongly convex function $f : \mathbb{R}^{O(T)} \to \mathbb{R}$ with an inexact initialization model such that any FOI algorithm $\mathcal{A}$ that starts at $\boldsymbol{x}_0 = \boldsymbol{0}$ has its $(\varepsilon, \delta)$-deviation lower bounded by $\Omega(\frac{1}{T\mu^2} \wedge \frac{\varepsilon}{\mu})$.*

*Proof.* For the initialization error model, we use a simpler construction. For $\boldsymbol{x} = (\boldsymbol{x}^{\text{err}}, \boldsymbol{y}, w) \in (\mathbb{R}^T)^2 \times \mathbb{R} \to \mathbb{R}$, consider the cost defined as

$$f(\boldsymbol{x}^{\text{err}}, \boldsymbol{y}, w) = \max\{0, \ \boldsymbol{x}^{\text{err}}[1] + \boldsymbol{y}[1], \ \ldots, \ \boldsymbol{x}^{\text{err}}[T] + \boldsymbol{y}[T]\} + w + \frac{\mu}{2} \left\| (\boldsymbol{x}^{\text{err}}, \boldsymbol{y}, w) \right\|^2 \,.$$

Then, clearly $f$ is clearly $O(1)$-Lipschitz. We set the reference and inexact intializations as follows:

$$\boldsymbol{x}_0^{\text{ref}} = (0, 0, \ldots, 0) \quad \text{and} \quad \boldsymbol{x}_0 = (\underbrace{\delta, \cdots, \delta}_{\text{first } T}, \underbrace{0, \cdots, 0}_{\text{second } T}, 0) \,. \tag{D.22}$$

Following the previous proofs, we will write $\boldsymbol{x}_t = (\boldsymbol{x}_t^{\text{err}}, \boldsymbol{y}_t, w_t) \in (\mathbb{R}^T)^2 \times \mathbb{R}$ and the iterates corresponding to the reference initialization as $\boldsymbol{x}_t^{\text{ref}} = ((\boldsymbol{x}_t^{\text{err}})^{\text{ref}}, \boldsymbol{y}_t^{\text{ref}}, w_t^{\text{ref}})$. From the fact that the algorithm has to achieve $\varepsilon$-suboptimality, it follows that

$$w_T \in \left[ -\frac{1}{2\mu} - O(\sqrt{\varepsilon}), -\frac{1}{2\mu} + O(\sqrt{\varepsilon}) \right] \,.$$

Then following the same argument as Lemma 3 it holds that $w_T = \sum_{i=1}^{T} \boldsymbol{y}_T[i]$. Using the fact that $\boldsymbol{y}_T^{\text{ref}} = \boldsymbol{0}$, we have the following deviation lower bound:

$$\left\|\boldsymbol{x}_T - \boldsymbol{x}_T^{\text{ref}}\right\|^2 \geq \left\|\boldsymbol{y}_T - \boldsymbol{y}_T^{\text{ref}}\right\|^2 = \sum_{i=1}^{T}(\boldsymbol{y}_T[i])^2$$

Moreover, from the $\varepsilon$-suboptimality of $\boldsymbol{x}_T$, it must be that $\sum_{i=1}^{T}(\boldsymbol{y}_T[i])^2 \leq \frac{2}{\mu}\varepsilon$. Thus, either $\sum_{i=1}^{T}(\boldsymbol{y}_T[i])^2 = \Omega(\frac{\varepsilon}{\mu})$ (in which case the deviation is lower bounded by $\Omega(\frac{\varepsilon}{\mu})$), or the following deviation lower bound holds:

$$\sum_{i=1}^{T}(\boldsymbol{y}_T[i])^2 \geq \frac{1}{T} \cdot \left(\sum_{i=1}^{T} \boldsymbol{y}_T[i]\right)^2 = \frac{1}{T} \cdot (w_T)^2 \gtrsim \frac{1}{T\mu^2}.$$

This completes the proof. □

## E  Proof of upper bounds (smooth costs)

### E.1  Stochastic inexact gradient model

**Theorem 19. (Upper Bound)** *Let $f$ be an $O(1)$-smooth convex cost function. Let $\varepsilon > 0$, and $T$ be a given number of iterations. Under the stochastic inexact gradient model, the $(\varepsilon, \delta)$-deviation of standard SGD with an appropriately chosen step size is $O(\frac{\delta^2}{T\varepsilon^2})$, provided that $T = \Omega(1/\varepsilon^2)$.*

*Proof.* Throughout the proof, let $L$ be the smoothness constant of $f$. We first derive the deviation bound. Let $\{\boldsymbol{x}_t\}$ be the GD iterates with stochastic inexact gradients and $\{\boldsymbol{y}_t\}$ be the GD iterates with exact gradients. Assuming that $\eta_t \leq \frac{2}{L}$, the standard convex analysis yields the following one-step deviation inequality ($\mathbb{E}$ denotes the conditional expectation over the randomness in $g(\boldsymbol{x}_t)$)

$$
\begin{aligned}
\mathbb{E}\left\|\boldsymbol{x}_{t+1} - \boldsymbol{y}_{t+1}\right\|^2 &= \mathbb{E}\left\|(\boldsymbol{x}_t - \eta_t g(\boldsymbol{x}_t) - (\boldsymbol{y}_t - \eta_t \nabla f(\boldsymbol{y}_t))\right\|^2 \\
&= \left\|\boldsymbol{x}_t - \boldsymbol{y}_t\right\|^2 - 2\eta_t \underbrace{\mathbb{E}\left\langle \boldsymbol{x}_t - \boldsymbol{y}_t, g(\boldsymbol{x}_t) - \nabla f(\boldsymbol{y}_t)\right\rangle}_{=\langle \boldsymbol{x}_t - \boldsymbol{y}_t, \nabla f(\boldsymbol{x}_t) - \nabla f(\boldsymbol{y}_t)\rangle} + \eta_t^2 \mathbb{E}\left\|g(\boldsymbol{x}_t) - \nabla f(\boldsymbol{y}_t)\right\|^2 \\
&= \left\|\boldsymbol{x}_t - \boldsymbol{y}_t\right\|^2 - 2\eta_t \left\langle \boldsymbol{x}_t - \boldsymbol{y}_t, \nabla f(\boldsymbol{x}_t) - \nabla f(\boldsymbol{y}_t)\right\rangle + \eta_t^2 \mathbb{E}\left\|g(\boldsymbol{x}_t) - \nabla f(\boldsymbol{x}_t)\right\|^2 \\
&\quad + \underbrace{2\eta_t^2 \mathbb{E}\left\langle g(\boldsymbol{x}_t) - \nabla f(\boldsymbol{x}_t), \nabla f(\boldsymbol{x}_t) - \nabla f(\boldsymbol{y}_t)\right\rangle}_{=0} + \eta_t^2 \left\|\nabla f(\boldsymbol{x}_t) - \nabla f(\boldsymbol{y}_t)\right\|^2 \\
&= \left\|\boldsymbol{x}_t - \boldsymbol{y}_t\right\|^2 \underbrace{-2\eta_t \left\langle \boldsymbol{x}_t - \boldsymbol{y}_t, \nabla f(\boldsymbol{x}_t) - \nabla f(\boldsymbol{y}_t)\right\rangle + \eta_t^2 \left\|\nabla f(\boldsymbol{x}_t) - \nabla f(\boldsymbol{y}_t)\right\|^2}_{\leq 0} \\
&\quad + \eta_t^2 \mathbb{E}\left\|\nabla f(\boldsymbol{x}_t) - g(\boldsymbol{x}_t)\right\|^2 \\
&\leq \left\|\boldsymbol{x}_t - \boldsymbol{y}_t\right\|^2 + \eta_t^2 \delta^2.
\end{aligned}
$$

Here the last inequality is due to the standard fact about smooth and convex function that for any $\boldsymbol{x}, \boldsymbol{y}$, $\frac{1}{L}\left\|\nabla f(\boldsymbol{x}) - \nabla f(\boldsymbol{y})\right\|^2 \leq \langle \nabla f(\boldsymbol{x}) - \nabla f(\boldsymbol{y}), \boldsymbol{x} - \boldsymbol{y}\rangle$ (see, e.g., [Nesterov, 2018, (2.1.11)]), together with the fact $\eta_t \leq \frac{2}{L}$. Hence, we have proved

$$\mathbb{E}\left\|\boldsymbol{x}_T - \boldsymbol{y}_T\right\|^2 \leq \delta^2 \sum_t \eta_t^2. \tag{E.1}$$

Now for the upper bound, we consider variants of SGD. From the standard convergence result (see, e.g., [Bubeck, 2014, Thm. 6.3]), with step size $\eta_t \equiv \frac{1}{L + 1/\eta}$ for some $\eta > 0$,

$$\mathbb{E} f\left(\frac{1}{T}\sum_{t=1}^{T} \boldsymbol{x}_t\right) - f(\boldsymbol{x}_*) \leq \frac{L\left\|\boldsymbol{x}_0 - \boldsymbol{x}_*\right\|^2}{2T} + \frac{\left\|\boldsymbol{x}_0 - \boldsymbol{x}_*\right\|^2}{2\eta T} + \frac{\eta \delta^2}{2}. \tag{E.2}$$

For simplicity, let $\bar{\boldsymbol{x}}_T := \frac{1}{T}\sum_t \boldsymbol{x}_t$, and $\bar{\boldsymbol{y}}_T := \frac{1}{T}\sum_t \boldsymbol{y}_t$. From the convexity of $\|\cdot\|^2$, we have

$$\mathbb{E}\left\|\bar{\boldsymbol{x}}_T - \bar{\boldsymbol{y}}_T\right\|^2 \leq \frac{1}{T}\sum_t \mathbb{E}\left\|\boldsymbol{x}_t - \boldsymbol{y}_t\right\|^2. \tag{E.3}$$

Now let us combine above results to upper bound $(\varepsilon, \delta)$-deviation.

As a warm-up, let us first consider SGD with $\eta = O(1/\sqrt{T})$. From (E.2), it follows that the convergence rate reads $\mathbb{E} f(\bar{\boldsymbol{x}}_T) - f(\boldsymbol{x}_*) \leq O(1/\sqrt{T})$. With such a choice of $\eta$, the stepsize is

$$\eta_t \equiv \frac{1}{L + \Omega(\sqrt{T})} = O(\frac{1}{\sqrt{T}}).$$

Hence, for the deviation bound, using (E.1) together with (E.3), we have

$$\mathbb{E}\left\|\bar{\boldsymbol{x}}_T - \bar{\boldsymbol{y}}_T\right\|^2 \leq \frac{1}{T}\sum_t \mathbb{E}\left\|\boldsymbol{x}_t - \boldsymbol{y}_t\right\|^2 \lesssim \frac{1}{T}\sum_{t=1}^{T}\left[t \cdot \frac{1}{T} \cdot \delta^2\right] \lesssim \delta^2.$$

This shows that with $T = \Omega(1/\varepsilon^2)$, the $(\varepsilon, \delta)$-deviation is $O(\delta^2)$.

In order to recover the bound in the theorem statement, we consider a mini-batch SGD. In particular, the above calculation shows that using a mini-batch of size $b$ at each iteration, it follows that with $O(\frac{b}{\varepsilon^2})$ gradient queries, the deviation is upper bounded by $O(\frac{\delta^2}{b})$. This precisely corresponds to the $(\varepsilon, \delta)$-deviation bound of $O(\frac{\delta^2}{\varepsilon^2 T})$.

An alternative way is to let the learning rate $\eta_t = 1/(\varepsilon T)$. It then follows from (E.2) that

$$\mathbb{E} f(\bar{\boldsymbol{x}}_T) - f(\boldsymbol{x}_*) \leq O\left(\frac{1}{T} + \varepsilon + \frac{\delta^2}{\varepsilon T}\right) = O(\varepsilon),$$

since $T = \Omega(1/\varepsilon^2)$. Moreover, (E.1) and (E.3) imply

$$\mathbb{E}\left\|\bar{\boldsymbol{x}}_T - \bar{\boldsymbol{y}}_T\right\|^2 \leq \frac{1}{T}\sum_t \mathbb{E}\left\|\boldsymbol{x}_t - \boldsymbol{y}_t\right\|^2 \leq \frac{1}{T}\sum_{t=1}^{T}\left[t \cdot \delta^2 \cdot \frac{1}{\varepsilon^2 T^2}\right] \leq \frac{\delta^2}{T\varepsilon^2},$$

which is the desired upper bound. $\qquad\square$

## E.2 Non-stochastic gradient errors

**Theorem 20. (Upper Bound)** *For $L = O(1)$ and $D = O(1)$, let $f$ be an $L$-smooth convex cost function whose optimum lies in a ball of radius $D$. Let $\varepsilon > 0$ and $\delta > 0$ are such that $\delta \leq \frac{\varepsilon}{2LD}$. Let $T$ be a given number of iterations. Under the non-stochastic inexact gradient model, there exists a FOI algorithm whose $(\varepsilon, \delta)$-deviation is $O(\frac{\delta^2}{\varepsilon^2})$, provided that $T = \Omega(\frac{1}{\varepsilon})$.*

*Proof.* Throughout the proof, let $L$ be the smoothness constant of $f$. We consider the projected gradient descent with step size $\eta_t = \frac{1}{L}$ onto the ball of radius $D$ that contains the optimum $\boldsymbol{x}_*$. It is important to note that this algorithm is a FOI because the projection onto the ball of radius $D$ is a re-scaling, and hence after the projection, the coefficients $\lambda_i^{(t)}$ are still positive.

The proximal inequality (e.g., [Bauschke et al., 2011, Proposition 12.26]) implies that

$$\left\|\boldsymbol{x}_{t+1} - \boldsymbol{x}_*\right\|^2 - \left\|\boldsymbol{x}_t - \boldsymbol{x}_*\right\|^2 \leq -\left\|\boldsymbol{x}_{t+1} - \boldsymbol{x}_t\right\|^2 - 2\eta_t \langle g(\boldsymbol{x}_t), \boldsymbol{x}_{t+1} - \boldsymbol{x}_*\rangle$$

Let $\Delta_t$ denote the error due to the non-stochastic inexact gradient model at iteration $t$, i.e., $\Delta_t := g(\boldsymbol{x}_t) - \nabla f(\boldsymbol{x}_t)$. Then we have

$$\left\|\boldsymbol{x}_{t+1} - \boldsymbol{x}_*\right\|^2 - \left\|\boldsymbol{x}_t - \boldsymbol{x}_*\right\|^2$$
$$\leq -\left\|\boldsymbol{x}_{t+1} - \boldsymbol{x}_t\right\|^2 - 2\eta_t \langle \nabla f(\boldsymbol{x}_t) + \Delta_t, \boldsymbol{x}_{t+1} - \boldsymbol{x}_*\rangle$$
$$\leq -\left\|\boldsymbol{x}_{t+1} - \boldsymbol{x}_t\right\|^2 + 2\eta_t \langle \nabla f(\boldsymbol{x}_t), \boldsymbol{x}_* - \boldsymbol{x}_{t+1}\rangle + 2\delta D$$
$$= -\underbrace{\left(\left\|\boldsymbol{x}_{t+1} - \boldsymbol{x}_t\right\|^2 + 2\eta_t \langle \nabla f(\boldsymbol{x}_t), \boldsymbol{x}_{t+1} - \boldsymbol{x}_t\rangle\right)}_{(a)} + \underbrace{2\eta_t \langle \nabla f(\boldsymbol{x}_t), \boldsymbol{x}_* - \boldsymbol{x}_t\rangle}_{(b)} + 2\delta D$$
$$\leq -2\eta_t(f(\boldsymbol{x}_{t+1}) - f(\boldsymbol{x}_t)) + 2\eta_t(f(\boldsymbol{x}_*) - f(\boldsymbol{x}_t)) + 2\delta D$$
$$= 2\eta_t(f(\boldsymbol{x}_*) - f(\boldsymbol{x}_{t+1})) + 2\delta D,$$

where $(a)$ is upper bounded using the $L$-smoothness together with $\eta_t = \frac{1}{L}$ as follows:

$$f(\boldsymbol{x}_{t+1}) - f(\boldsymbol{x}_t) \leq \langle \nabla f(\boldsymbol{x}_t), \boldsymbol{x}_{t+1} - \boldsymbol{x}_t \rangle + \frac{L}{2} \|\boldsymbol{x}_{t+1} - \boldsymbol{x}_t\|^2$$

$$= \langle \nabla f(\boldsymbol{x}_t), \boldsymbol{x}_{t+1} - \boldsymbol{x}_t \rangle + \frac{1}{2\eta_t} \|\boldsymbol{x}_{t+1} - \boldsymbol{x}_t\|^2 \, ,$$

and $(b)$ is handled using convexity.

Summing this over all $t = 0, 1, \ldots, T-1$ gives

$$\sum_{t=0}^{T-1} \frac{2}{L} (f(\boldsymbol{x}_{t+1}) - f(\boldsymbol{x}_*)) \leq \|\boldsymbol{x}_0 - \boldsymbol{x}_*\|^2 + 2\delta T D,$$

which implies the following average-iterate guarantee:

$$f(\bar{\boldsymbol{x}}_T) - f(\boldsymbol{x}_*) \leq \frac{LD^2}{2T} + \delta L D.$$

Thus, in order to achieve $\varepsilon$-suboptimality, we need $\Omega(1/\varepsilon)$ iterations, since the theorem statement assumed that $\delta \leq \frac{\varepsilon}{2LD}$. Next, let us bound the deviation.

**Lemma 4.** *Suppose that $f$ is $L$-smooth. Let $\Delta_t$ and $\Delta'_t$ denote noises in the gradients. If $\eta_t \leq \frac{2}{L}$, then the following one-step deviation inequality holds*

$$\|\boldsymbol{x}_t - \eta_t(\nabla f(\boldsymbol{x}_t) + \Delta_t) - (\boldsymbol{x}'_t - \eta_t(\nabla f(\boldsymbol{x}'_t) + \Delta'_t))\| \leq \|\boldsymbol{x}_t - \boldsymbol{x}'_t\| + \eta_t \|\Delta_t\| + \eta_t \|\Delta'_t\| \, .$$

*Proof.* The proof follows from the following inequality:

$$\|\boldsymbol{x}_{t+1} - \boldsymbol{x}'_{t+1}\| \leq \|\boldsymbol{x}_t - \boldsymbol{x}'_t - \eta_t(g(\boldsymbol{x}_t) - g(\boldsymbol{x}'_t))\|$$

$$\leq \|\boldsymbol{x}_t - \boldsymbol{x}'_t - \eta_t(\nabla f(\boldsymbol{x}_t) - \nabla f(\boldsymbol{x}'_t))\| + \eta_t \|\Delta_t\| + \eta_T \|\Delta_t\|'$$

$$\overset{(a)}{\leq} \|\boldsymbol{x}_t - \boldsymbol{x}'_t\| + \eta_t \|\Delta_t\| + \eta_t \|\Delta_t\|' \, ,$$

Here $(a)$ follows from the following fact:

$$\|\boldsymbol{x}_t - \boldsymbol{x}'_t - \eta_t(\nabla f(\boldsymbol{x}_t) - \nabla f(\boldsymbol{x}'_t))\|^2$$

$$= \|\boldsymbol{x}_t - \boldsymbol{x}'_t\|^2 - 2\eta_t \langle \boldsymbol{x}_t - \boldsymbol{x}'_t, \nabla f(\boldsymbol{x}_t) - \nabla f(\boldsymbol{x}'_t) \rangle + \eta_t^2 \|\nabla f(\boldsymbol{x}_t) - \nabla f(\boldsymbol{x}'_t)\|^2$$

$$\overset{(b)}{\leq} \|\boldsymbol{x}_t - \boldsymbol{x}'_t\|^2 - \frac{2}{L}\eta_t \|\nabla f(\boldsymbol{x}_t) - \nabla f(\boldsymbol{x}'_t)\|^2 + \eta_t^2 \|\nabla f(\boldsymbol{x}_t) - \nabla f(\boldsymbol{x}'_t)\|^2$$

$$\overset{(c)}{\leq} \|\boldsymbol{x}_t - \boldsymbol{x}'_t\|^2 \, ,$$

where $(b)$ is due to the fact that for a $L$-smooth and convex function $f$, it holds that

$$\frac{1}{L} \|\nabla f(\boldsymbol{x}) - \nabla f(\boldsymbol{y})\|^2 \leq \langle \nabla f(\boldsymbol{x}) - \nabla f(\boldsymbol{y}), \boldsymbol{x} - \boldsymbol{y} \rangle \quad \text{for any } \boldsymbol{x}, \boldsymbol{y},$$

and $(c)$ is because $\eta_t^2 \leq \frac{2}{L}\eta_t$. $\qquad\square$

Now given the convergence rate and the deviation inequality, we are ready to prove the desired upper bound on $(\varepsilon, \delta)$-deviation. From the triangle inequality, we get

$$\|\bar{\boldsymbol{x}}_T - \bar{\boldsymbol{x}}'_T\| = \left\| \frac{1}{T} \sum_{t=1}^{T} \boldsymbol{x}_t - \frac{1}{T} \sum_{t=1}^{T} \boldsymbol{x}'_t \right\| \leq \frac{1}{T} \sum_{t=1}^{T} \|\boldsymbol{x}_t - \boldsymbol{x}'_t\|$$

$$\leq \frac{1}{T} \sum_{t=1}^{T} O(t\delta) \leq O(T\delta) \, ,$$

where the second line follows from Lemma 4. Thus, the $(\varepsilon, \delta)$-deviation is bounded by $O(\frac{\delta^2}{\varepsilon^2})$ using the averaged iterate with $T = \Theta(1/\varepsilon)$. $\qquad\square$

One limitation of Theorem 20 is that it requires the optimum to lie in a bounded domain. Next we show that without this requirement, the gradient descent iterate is still bounded when it first attains $\varepsilon$-accuracy.

**Theorem 21.** *Suppose $f$ is $L$-smooth with optimum $x^*$. Let $\varepsilon, \delta > 0$ be given such that $\varepsilon \le 2L\|\boldsymbol{x}_0 - \boldsymbol{x}_*\|^2$ and $\delta \le \frac{\varepsilon}{\|\boldsymbol{x}_0 - \boldsymbol{x}_*\|}$. Consider gradient descent with a constant learning rate $\eta = \frac{1}{L}$. Under the non-stochastic inexact gradient model, for the first iterate $\boldsymbol{x}_T$ with $f(\boldsymbol{x}_T) - f(\boldsymbol{x}_*) \le \varepsilon$, it holds that $\|\boldsymbol{x}_T - \boldsymbol{x}_*\| \le 2\|\boldsymbol{x}_0 - \boldsymbol{x}_*\|$.*

*Proof.* Let $D_t := \|\boldsymbol{x}_t - \boldsymbol{x}_*\|$. Following the proof of Theorem 20 (which still holds when the domain is unbounded), we can show that

$$D_{t+1}^2 - D_t^2 \le 2\eta(f(\boldsymbol{x}_*) - f(\boldsymbol{x}_{t+1})) + 2\eta\delta D_{t+1}. \tag{E.4}$$

Let $T$ denote the first step with $f(\boldsymbol{x}_T) \le f(\boldsymbol{x}_*) + \varepsilon$. We claim that for all $0 \le t \le T - 2$,

$$D_{t+1} \le D_t \le D_0. \tag{E.5}$$

We will prove (E.5) by induction. Given $0 \le t \le T - 2$, suppose $D_t \le D_0$, and note that the definition of $T$ implies $f(\boldsymbol{x}_{t+1}) > f(\boldsymbol{x}_*) + \varepsilon$. It then follows from (E.4) that

$$D_{t+1}^2 - 2\eta\delta D_{t+1} + 2\eta\varepsilon - D_t^2 < 0.$$

Let $h(z) := z^2 - 2\eta\delta z + 2\eta\varepsilon - D_t^2$. First,

$$h(D_t) = -2\eta\delta D_t + 2\eta\varepsilon \ge 0,$$

since $\delta D_t \le \delta D_0 \le \varepsilon$ by the condition of Theorem 21. Moreover, $D_t$ is larger than $\eta\delta$, the minimum of $h$, because if it is not true, then

$$f(\boldsymbol{x}_t) - f(\boldsymbol{x}_*) \le \frac{L}{2}D_t^2 \le \frac{L\eta^2\delta^2}{2} \le \frac{L\eta^2\varepsilon^2}{2D_0^2} = \frac{\varepsilon^2}{2LD_0^2} \le \varepsilon,$$

which contradicts the definition of $T$. Since $h(D_{t+1}) < 0$, it then follows that $D_{t+1} \le D_t \le D_0$, and in particular $D_{T-1} \le D_0$. Finally, note that smoothness implies

$$\|\nabla f(\boldsymbol{x}_{T-1})\| \le LD_{T-1} \le LD_0,$$

and thus

$$D_T \le \|\boldsymbol{x}_T - \boldsymbol{x}_{T-1}\| + D_{T-1} \le \eta\|\nabla f(\boldsymbol{x}_{T-1})\| + D_{T-1} \le D_0 + D_{T-1} \le 2D_0.$$

This completes the proof $\qquad\square$

## E.3 Inexact initialization model

**Theorem 22. (Upper Bound)** *Let $f$ be an $O(1)$-Lipschitz convex cost function. Let $\varepsilon > 0$ be a small constant, and $T$ be a given number of iterations. Then there exists a FOI algorithm whose $(\varepsilon, \delta)$-deviation is $O(\delta^2)$, provided that $T = \Omega(1/\varepsilon)$.*

*Proof.* In view of Lemma 4, two different runs $\boldsymbol{x}_T, \boldsymbol{x}_T'$ of gradient descent satisfies

$$\|\boldsymbol{x}_T - \boldsymbol{x}_T'\| \le \|\boldsymbol{x}_{T-1}' - \boldsymbol{x}_{T-1}'\| \le \cdots \le \|\boldsymbol{x}_0 - \boldsymbol{x}_0'\| \le 2\delta.$$

Hence, the statement follows. $\qquad\square$

In this section, we consider smooth and strongly-convex costs.

## E.4 Stochastic inexact gradient model (strongly convex costs)

We first show an upper bound for the stochastic inexact gradient oracle.

**Theorem 23. (Upper Bound)** *Let $f$ be an $O(1)$-smooth $\mu$-strongly convex cost function. Let $\varepsilon > 0$ be a small constant, and $T$ be a given number of iterations. Under the stochastic inexact gradient model, there exists a FOI algorithm whose $(\varepsilon, \delta)$-deviation is $O\left(\frac{\delta^2}{T\mu^2} \wedge \frac{\varepsilon}{\mu}\right)$, provided that $T = \Omega(\frac{1}{\varepsilon\mu})$.*

*Proof.* The following proof is based on the proof of [Bubeck, 2014, Theorem 6.3], but here we further make use of strong convexity.

Let $C \subset \mathbb{R}^n$ denote the domain of $f$, and assume it is convex and closed. We simply run stochastic gradient descent: starting from some $\boldsymbol{x}_0 \in C$, let

$$\boldsymbol{x}_{t+1} := \Pi_C \left[ \boldsymbol{x}_t - \eta_t g(\boldsymbol{x}_t) \right] \quad \text{where} \quad \eta_t := \frac{1}{L + 1/\lambda_t}.$$

We will pick a value for each $\lambda_t$ below. Note that $C$ can just be $\mathbb{R}^n$, in which case no projection is needed, but our analysis can also handle a bounded domain.

Let $\boldsymbol{x}_*$ denote the optimal solution, and suppose $f$ is $L$-smooth. It follows that

$$
\begin{aligned}
f(\boldsymbol{x}_{t+1}) - f(\boldsymbol{x}_t) &\leq \langle \nabla f(\boldsymbol{x}_t), \boldsymbol{x}_{t+1} - \boldsymbol{x}_t \rangle + \frac{L}{2} \|\boldsymbol{x}_{t+1} - \boldsymbol{x}_t\|^2 \\
&= \langle g(\boldsymbol{x}_t), \boldsymbol{x}_{t+1} - \boldsymbol{x}_t \rangle + \langle \nabla f(\boldsymbol{x}_t) - g(\boldsymbol{x}_t), \boldsymbol{x}_{t+1} - \boldsymbol{x}_t \rangle + \frac{L}{2} \|\boldsymbol{x}_{t+1} - \boldsymbol{x}_t\|^2 \\
&\leq \langle g(\boldsymbol{x}_t), \boldsymbol{x}_{t+1} - \boldsymbol{x}_t \rangle + \frac{\lambda_t}{2} \|\nabla f(\boldsymbol{x}_t) - g(\boldsymbol{x}_t)\|^2 + \frac{L + 1/\lambda_t}{2} \|\boldsymbol{x}_{t+1} - \boldsymbol{x}_t\|^2.
\end{aligned}
$$

Moreover, the projection step ensures

$$
\begin{aligned}
\frac{1}{L + 1/\lambda_t} \langle g(\boldsymbol{x}_t), \boldsymbol{x}_{t+1} - \boldsymbol{x}_* \rangle &\leq \langle \boldsymbol{x}_t - \boldsymbol{x}_{t+1}, \boldsymbol{x}_{t+1} - \boldsymbol{x}_* \rangle \\
&= \frac{1}{2} \left( \|\boldsymbol{x}_t - \boldsymbol{x}_*\|^2 - \|\boldsymbol{x}_t - \boldsymbol{x}_{t+1}\|^2 - \|\boldsymbol{x}_{t+1} - \boldsymbol{x}_*\|^2 \right).
\end{aligned}
$$

Consequently,

$$
\begin{aligned}
f(\boldsymbol{x}_{t+1}) - f(\boldsymbol{x}_t) &\leq \langle g(\boldsymbol{x}_t), \boldsymbol{x}_* - \boldsymbol{x}_t \rangle + \frac{\lambda_t}{2} \|\nabla f(\boldsymbol{x}_t) - g(\boldsymbol{x}_t)\|^2 \\
&\quad + \frac{L + 1/\lambda_t}{2} \left( \|\boldsymbol{x}_t - \boldsymbol{x}_*\|^2 - \|\boldsymbol{x}_{t+1} - \boldsymbol{x}_*\|^2 \right).
\end{aligned}
$$

Taking expectation with respect to $g(\boldsymbol{x}_t)$, we have

$$\mathbb{E}[f(\boldsymbol{x}_{t+1})] \leq f(\boldsymbol{x}_t) + \langle \nabla f(\boldsymbol{x}_t), \boldsymbol{x}_* - \boldsymbol{x}_t \rangle + \frac{\lambda_t}{2} \delta^2 + \frac{L + 1/\lambda_t}{2} \mathbb{E}\left[ \|\boldsymbol{x}_t - \boldsymbol{x}_*\|^2 - \|\boldsymbol{x}_{t+1} - \boldsymbol{x}_*\|^2 \right].$$

Further invoking strong convexity, we have

$$\mathbb{E}[f(\boldsymbol{x}_{t+1})] \leq f(\boldsymbol{x}_*) - \frac{\mu}{2} \|\boldsymbol{x}_t - \boldsymbol{x}_*\|^2 + \frac{\lambda_t \delta^2}{2} + \frac{L + 1/\lambda_t}{2} \mathbb{E}\left[ \|\boldsymbol{x}_t - \boldsymbol{x}_*\|^2 - \|\boldsymbol{x}_{t+1} - \boldsymbol{x}_*\|^2 \right].$$
(E.6)

Pick a small enough $k$ which also satisfies $k \geq 4L/\mu$, and let

$$\lambda_t = \frac{2}{(t+k)\mu - 2L}.$$

It follows that $\lambda_t > 0$ by construction, and also

$$L + \frac{1}{\lambda_t} = \frac{t+k}{2}\mu, \quad \text{and} \quad L + \frac{1}{\lambda_t} - \mu = \frac{t+k-2}{2}\mu,$$

and that

$$(t+k-1)\lambda_t = \frac{2}{\mu} \cdot \frac{t+k-1}{t+k-2L/\mu} \leq \frac{2}{\mu} \cdot \frac{t+k-1}{(t+k)/2} \leq \frac{4}{\mu},$$

since by the definition of $k$, we have $2L/\mu \leq k/2 \leq (t+k)/2$. Therefore if we multiply both sides of (E.6) by $(t+k-1)$, we get

$$
\begin{aligned}
(t+k-1) \mathbb{E}\left[ f(\boldsymbol{x}_{t+1}) - f(\boldsymbol{x}_*) \right] &\leq \frac{(t+k-1)(t+k-2)}{4} \mu \, \mathbb{E}[\|\boldsymbol{x}_t - \boldsymbol{x}_*\|^2] \\
&\quad - \frac{(t+k)(t+k-1)}{4} \mu \, \mathbb{E}[\|\boldsymbol{x}_{t+1} - \boldsymbol{x}_*\|^2] + \frac{2\delta^2}{\mu}.
\end{aligned}
$$

Now taking the sum from $t = 0$ to $T - 1$, we have

$$\sum_{t=0}^{T-1} (t + k - 1) \, \mathbb{E} \left[ f(\boldsymbol{x}_{t+1}) - f(\boldsymbol{x}_*) \right] \leq \frac{(k-1)(k-2)\mu}{4} \|\boldsymbol{x}_0 - \boldsymbol{x}_*\|^2 + \frac{2\delta^2 T}{\mu}.$$

Define

$$\tilde{\boldsymbol{x}}_T := \sum_{t=0}^{T-1} \frac{t + k - 1}{\sum_{j=0}^{T-1} (j + k - 1)} \boldsymbol{x}_{t+1} \, .$$

Then, we have

$$\mathbb{E} \left[ f(\tilde{\boldsymbol{x}}_T) - f(\boldsymbol{x}_*) \right] \leq O \left( \frac{(k-1)(k-2)\mu}{4T^2} \|\boldsymbol{x}_0 - \boldsymbol{x}_*\|^2 + \frac{2\delta^2}{T\mu} \right).$$

Since $k = \Theta(L/\mu)$, it follows that as long as $T = \Omega(\frac{1}{\varepsilon\mu})$, we have

$$\mathbb{E} \left[ f(\tilde{\boldsymbol{x}}_T) - f(\boldsymbol{x}_*) \right] \leq \varepsilon.$$

Next we analyze the deviation bound. Similarly to the proof of Theorem 19, let $\{\boldsymbol{x}_t\}$ denote GD iterates with stochastic inexact gradients, and let $\{\boldsymbol{y}_t\}$ denote GD iterates with exact gradients, we can show

$$\mathbb{E} \left[ \|\boldsymbol{x}_{t+1} - \boldsymbol{y}_{t+1}\|^2 \right] \leq \|\boldsymbol{x}_t - \boldsymbol{y}_t\|^2 - 2\eta_t \langle \boldsymbol{x}_t - \boldsymbol{y}_t, \nabla f(\boldsymbol{x}_t) - \nabla f(\boldsymbol{y}_t) \rangle + \eta_t^2 \|\nabla f(\boldsymbol{x}_t) - \nabla f(\boldsymbol{y}_t)\|^2$$
$$+ \eta_t^2 \, \mathbb{E} \left[ \|\nabla f(\boldsymbol{x}_t) - \nabla g(\boldsymbol{x}_t)\|^2 \right].$$

Next we need the following lemma.

**Lemma 5.** *Suppose $f$ is $L$-smooth and $\mu$-strongly convex. For $\eta \leq 1/L$, it holds for any $\boldsymbol{x}, \boldsymbol{y}$ that*

$$\|\boldsymbol{x} - \boldsymbol{y} - \eta(\nabla f(\boldsymbol{x}) - \nabla f(\boldsymbol{y}))\|^2 \leq (1 - \eta\mu) \|\boldsymbol{x} - \boldsymbol{y}\|^2.$$

*Proof.* First we have

$$\|\boldsymbol{x} - \boldsymbol{y} - \eta(\nabla f(\boldsymbol{x}) - \nabla f(\boldsymbol{y}))\|^2$$
$$= \|\boldsymbol{x} - \boldsymbol{y}\|^2 - 2\eta \langle \boldsymbol{x} - \boldsymbol{y}, \nabla f(\boldsymbol{x}) - \nabla f(\boldsymbol{y}) \rangle + \eta^2 \|\nabla f(\boldsymbol{x}) - \nabla f(\boldsymbol{y})\|^2$$
$$\leq \|\boldsymbol{x} - \boldsymbol{y}\|^2 - 2\eta \langle \boldsymbol{x} - \boldsymbol{y}, \nabla f(\boldsymbol{x}) - \nabla f(\boldsymbol{y}) \rangle + \eta \frac{1}{L} \cdot L \langle \boldsymbol{x} - \boldsymbol{y}, \nabla f(\boldsymbol{x}) - \nabla f(\boldsymbol{y}) \rangle$$
$$= \|\boldsymbol{x} - \boldsymbol{y}\|^2 - \eta \langle \boldsymbol{x} - \boldsymbol{y}, \nabla f(\boldsymbol{x}) - \nabla f(\boldsymbol{y}) \rangle,$$

where the inequality is due to smoothness and $\eta \leq 1/L$. Strong convexity then implies

$$\|\boldsymbol{x} - \boldsymbol{y} - \eta(\nabla f(\boldsymbol{x}) - \nabla f(\boldsymbol{y}))\|^2 \leq \|\boldsymbol{x} - \boldsymbol{y}\|^2 - \eta\mu\|\boldsymbol{x} - \boldsymbol{y}\|^2.$$

$\square$

Note that in the current setting, $\eta_t = 1/(L + 1/\lambda_t) \leq 1/L$, therefore we can invoke Lemma 5 and obtain

$$\mathbb{E} \left[ \|\boldsymbol{x}_{t+1} - \boldsymbol{y}_{t+1}\|^2 \right] \leq (1 - \eta_t\mu) \, \mathbb{E} \left[ \|\boldsymbol{x}_t - \boldsymbol{y}_t\|^2 \right] + \eta_t^2 \delta^2,$$

which further implies

$$\mathbb{E} \left[ \|\boldsymbol{x}_T - \boldsymbol{y}_T\|^2 \right] \leq \delta^2 \sum_{t=0}^{T-1} \eta_t^2 \prod_{j=t+1}^{T-1} (1 - \eta_j\mu),$$

since $\boldsymbol{x}_0 = \boldsymbol{y}_0$. Note that

$$\eta_t = \frac{2}{(t + k)\mu}, \quad \text{and} \quad 1 - \eta_t\mu = \frac{t + k - 2}{t + k},$$

therefore

$$\mathbb{E}\left[\|\boldsymbol{x}_T - \boldsymbol{y}_T\|^2\right] \leq \delta^2 \sum_{t=0}^{T-1} \frac{4}{(t+k)^2\mu^2} \frac{(t-1+k)(t+k)}{(T-2+k)(T-1+k)}$$

$$\leq \delta^2 \sum_{t=0}^{T-1} \frac{4}{\mu^2} \frac{1}{(T-2+k)(T-1+k)}$$

$$\leq \delta^2 \sum_{t=0}^{T-1} \frac{4}{\mu^2 T^2} \leq \frac{4\delta^2}{T\mu^2}.$$

Now define

$$\tilde{\boldsymbol{y}}_T := \sum_{t=0}^{T-1} \frac{t+k-1}{\sum_{j=0}^{T-1}(j+k-1)} \boldsymbol{y}_{t+1}.$$

Since $\tilde{\boldsymbol{x}}_T$ and $\tilde{\boldsymbol{y}}_T$ are weighted averages of $\boldsymbol{x}_t$ and $\boldsymbol{y}_t$ respectively, we have

$$\mathbb{E}\left[\|\tilde{\boldsymbol{x}}_T - \tilde{\boldsymbol{y}}_T\|^2\right] \leq \frac{4\delta^2}{T\mu^2},$$

Moreover, since $\tilde{\boldsymbol{y}}_T$ is deterministic, an $O(\frac{\delta^2}{T\mu^2})$ deviation bound also follows. □

### E.5 Non-stochastic inexact gradient model (strongly convex costs)

Next we consider the non-stochastic inexact gradient oracle.

**Theorem 24. (Upper Bound)** *For $L = O(1)$ and $D = O(1)$, let $f$ be an $L$-smooth $\mu$-strongly convex cost function whose optimum lies in a ball of radius $D$. Let $\varepsilon > 0$ and $\delta > 0$ are such that $\delta \leq \frac{\varepsilon}{2LD}$, and let $T$ be a given number of iterations. Under the non-stochastic inexact gradient model, there exists a FOI algorithm whose $(\varepsilon, \delta)$-deviation is $O\left(\frac{\delta^2}{\mu^2} \wedge \frac{\varepsilon}{\mu}\right)$, provided that $T = \Omega(1/\varepsilon)$.*

*Proof.* We run projected gradient descent with a constant learning rate $\eta = 1/L$. For the upper bound on excess error, we simply invoke Theorem 20: as long as $T = \Omega(1/\varepsilon)$, it holds that

$$f(\bar{\boldsymbol{x}}_T) \leq \frac{1}{T} \sum_{t=1}^{T} f(\boldsymbol{x}_t) \leq f(\boldsymbol{x}_*) + \varepsilon.$$

To bound the deviation, we follow a similar analysis as in the proof of Lemma 4. Consider two gradient descent runs $\{\boldsymbol{x}_t\}$ and $\{\boldsymbol{x}'_t\}$, and let $g(\boldsymbol{x}_t) = \nabla f(\boldsymbol{x}_t) + \Delta_t$, and $g(\boldsymbol{x}'_t) = \nabla f(\boldsymbol{x}'_t) + \Delta'_t$. First we have

$$\left\|\boldsymbol{x}_{t+1} - \boldsymbol{x}'_{t+1}\right\| \leq \|\boldsymbol{x}_t - \boldsymbol{x}'_t - \eta_t(g(\boldsymbol{x}_t) - g(\boldsymbol{x}'_t))\|$$

$$\leq \|\boldsymbol{x}_t - \boldsymbol{x}'_t - \eta_t(\nabla f(\boldsymbol{x}_t) - \nabla f(\boldsymbol{x}'_t))\| + \eta_t \|\Delta_t\| + \eta_t \|\Delta'_t\|$$

$$\leq \|\boldsymbol{x}_t - \boldsymbol{x}'_t - \eta_t(\nabla f(\boldsymbol{x}_t) - \nabla f(\boldsymbol{x}'_t))\| + \frac{2\delta}{L}.$$

Moreover, Lemma 5 implies

$$\|\boldsymbol{x}_t - \boldsymbol{x}'_t - \eta_t(\nabla f(\boldsymbol{x}_t) - \nabla f(\boldsymbol{x}'_t))\| \leq \sqrt{1 - \frac{\mu}{L}} \|\boldsymbol{x}_t - \boldsymbol{x}'_t\|.$$

Therefore

$$\|\boldsymbol{x}_{t+1} - \boldsymbol{x}'_{t+1}\| \leq \sqrt{1 - \frac{\mu}{L}} \|\boldsymbol{x}_t - \boldsymbol{x}'_t\| + \frac{2\delta}{L},$$

and for all $t \geq 1$,

$$\|\boldsymbol{x}_t - \boldsymbol{x}_t'\| \leq \left(1 - \frac{\mu}{L}\right)^{t/2} \|\boldsymbol{x}_0 - \boldsymbol{x}_0'\| + \frac{2\delta}{L} \cdot \frac{1}{1 - \sqrt{1 - \mu/L}}$$

$$= \left(1 - \frac{\mu}{L}\right)^{t/2} \|\boldsymbol{x}_0 - \boldsymbol{x}_0'\| + \frac{2\delta}{L} \cdot \frac{1 + \sqrt{1 - \mu/L}}{\mu/L}$$

$$\leq \left(1 - \frac{\mu}{L}\right)^{t/2} \|\boldsymbol{x}_0 - \boldsymbol{x}_0'\| + \frac{4\delta}{\mu}.$$

Finally,

$$\|\bar{\boldsymbol{x}}_T - \bar{\boldsymbol{x}}_T'\| \leq \frac{1}{T} \sum_{t=1}^{T} \|\boldsymbol{x}_t - \boldsymbol{x}_t'\| \leq \frac{4\delta}{\mu} + \frac{1}{T} \underbrace{\|\boldsymbol{x}_0 - \boldsymbol{x}_0'\|}_{=0} \frac{1}{1 - \sqrt{1 - \mu/L}} = O\left(\frac{\delta}{\mu}\right).$$

This completes the proof. $\qquad\square$

### E.6 Inexact initialization model (strongly convex costs)

**Theorem 25. (Upper Bound)** *Let $f$ be an $L$-smooth $\mu$-strongly convex cost function. Let $\varepsilon > 0$ be a small constant, and $T$ be a given number of iterations. Then there exists a FOI algorithm whose $(\varepsilon, \delta)$-deviation is $O(\exp(-\mu T/L)\delta^2 \wedge \frac{\varepsilon}{\mu})$.*

*Proof.* Let $\boldsymbol{x}_0$, $\boldsymbol{x}_0'$ denote two initial iterates. Lemma 5 implies

$$\|\boldsymbol{x}_{t+1} - \boldsymbol{x}_{t+1}'\|^2 = \|\boldsymbol{x}_t - \boldsymbol{x}_t' - \eta_t(\nabla f(\boldsymbol{x}_t) - \nabla f(\boldsymbol{x}_t'))\|^2 \leq \left(1 - \frac{\mu}{L}\right) \|\boldsymbol{x}_t - \boldsymbol{x}_t'\|^2,$$

and

$$\|\boldsymbol{x}_T - \boldsymbol{x}_T'\|^2 \leq \left(1 - \frac{\mu}{L}\right)^T \|\boldsymbol{x}_0 - \boldsymbol{x}_0'\|^2 \leq e^{-\mu T/L} \|\boldsymbol{x}_0 - \boldsymbol{x}_0'\|^2.$$

This completes the proof. $\qquad\square$

## F Proof of upper bounds (nonsmooth costs)

### F.1 Stochastic inexact gradient model

**Theorem 26. (Upper Bound)** *Let $f$ be an $O(1)$-Lipschitz convex cost function. Let $\varepsilon > 0$ be a small constant, and $T$ be a given number of iterations. Under the stochastic inexact gradient model, there exists a FOI algorithm whose $(\varepsilon, \delta)$-deviation is $O(\frac{1}{T\varepsilon^2})$, provided that $T = \Omega(1/\varepsilon^2)$.*

*Proof.* Assume now that $f$ is $G$-Lipschitz but otherwise nonsmooth ($G = O(1)$). Let $\{\boldsymbol{x}_t\}$ be the GD iterates with stochastic inexact gradients and $\{\boldsymbol{y}_t\}$ be the GD iterates with exact gradients. Then the one-step deviation bound can be derived as follows ($\mathbb{E}$ denotes the conditional expectation over the randomness in $g(\boldsymbol{x}_t)$):

$$\mathbb{E} \|\boldsymbol{x}_{t+1} - \boldsymbol{y}_{t+1}\|^2 = \mathbb{E} \|(\boldsymbol{x}_t - \eta_t g(\boldsymbol{x}_t) - (\boldsymbol{y}_t - \eta_t \nabla f(\boldsymbol{y}_t))\|^2$$

$$= \|\boldsymbol{x}_t - \boldsymbol{y}_t\|^2 - 2\eta_t \underbrace{\langle \boldsymbol{x}_t - \boldsymbol{y}_t, \nabla f(\boldsymbol{x}_t) - \nabla f(\boldsymbol{y}_t) \rangle}_{\leq 0 \ (\because \text{ convexity})}$$

$$+ \eta_t^2 \mathbb{E} \|g(\boldsymbol{x}_t) - \nabla f(\boldsymbol{x}_t)\|^2 + \eta_t^2 \|\nabla f(\boldsymbol{x}_t) - \nabla f(\boldsymbol{y}_t)\|^2$$

$$\leq \|\boldsymbol{x}_t - \boldsymbol{y}_t\|^2 + \eta_t^2 (4G^2 + \delta^2).$$

Since we consider the regime $\delta^2 \lesssim 1$, the one-step bound leads to the following deviation inequality:

$$\mathbb{E} \|\boldsymbol{x}_T - \boldsymbol{y}_T\|^2 \lesssim \sum_t \eta_t^2. \tag{F.1}$$

Note that the above deviation bound is worse than the smooth case deviation bound (E.1) which reads $\mathbb{E} \|\boldsymbol{x}_T - \boldsymbol{y}_T\|^2 \le \delta^2 \sum_t \eta_t^2$.

For the algorithm, we again consider SGD. Invoking the standard convergence guarantee of SGD for nonsmooth costs (see, e.g., [Bubeck, 2014, Thm. 6.1]), with step size $\eta_t \equiv \eta$ for some $\eta > 0$, we have the following convergence rate:

$$\mathbb{E} f\left(\frac{1}{T} \sum_{t=1}^T \boldsymbol{x}_t\right) - f(\boldsymbol{x}_*) \le \frac{\|\boldsymbol{x}_0 - \boldsymbol{x}_*\|^2}{2\eta T} + \frac{\eta G^2}{2}. \tag{F.2}$$

From (F.2), it follows that with $\eta = O(1/\sqrt{T})$, the convergence rate reads $\mathbb{E} f(\bar{\boldsymbol{x}}_T) - f(\boldsymbol{x}_*) \le O(1/\sqrt{T})$. With such a choice of $\eta$, the deviation can be bounded using (E.1) together with (E.3),

$$\mathbb{E} \|\bar{\boldsymbol{x}}_T - \bar{\boldsymbol{y}}_T\|^2 \le \frac{1}{T} \sum_t \mathbb{E} \|\boldsymbol{x}_t - \boldsymbol{y}_t\|^2 \lesssim \frac{1}{T} \sum_{t=1}^T \left[t \cdot \frac{1}{T} \cdot G^2\right] \lesssim G^2.$$

In fact, by choosing $\eta = \frac{1}{\varepsilon T}$ (since $T = \Omega(1/\varepsilon^2)$, it must be that $\eta = O(\varepsilon)$), it follows that the $(\varepsilon, \delta)$-deviation is upper bounded by

$$O\left(\frac{1}{T} \sum_{t=1}^T \left[t \cdot \frac{1}{T^2 \varepsilon^2} \cdot G^2\right]\right) \lesssim O(\frac{G^2}{T\varepsilon^2}) \lesssim O(\frac{1}{T\varepsilon^2}).$$

This completes the proof. $\qquad\square$

## F.2    Non-stochastic inexact gradient model

We first prove a deviation bound.

**Lemma 6.** *Suppose that $f$ is convex and $G$-Lipschitz. Let $\{y_t\}$ be the iterates of (projected) GD with stepsize $\eta_t$ with exact gradients and $\{x_t\}$ be the iterates of (projected) GD with the same stepsize with inexact gradients with noise $\{\Delta_t\}$. Assuming that $\|\Delta_t\| \le \delta$ for each t, we have*

$$\|\boldsymbol{x}_T - \boldsymbol{y}_T\| \le \sqrt{3(2G^2 + \delta^2) \cdot \sum_{t=0}^{T-1} \eta_t^2 + 2\delta \sum_{t=0}^{T-1} \eta_t}. \tag{F.3}$$

*Proof.* The proof is analogous to [Bassily et al., 2020, Lemma 3.1]. First, note that

$$\begin{aligned}
\|\boldsymbol{y}_{t+1} - \boldsymbol{x}_{t+1}\|^2 &\overset{(a)}{\le} \|\boldsymbol{y}_t - \eta_t \nabla f(\boldsymbol{y}_t) - (\boldsymbol{x}_t - \eta_t(\nabla f(\boldsymbol{x}_t) + \Delta_t))\|^2 \\
&= \|\boldsymbol{x}_t - \boldsymbol{y}_t\|^2 - 2\eta_t \langle \boldsymbol{y}_t - \boldsymbol{x}_t, \nabla f(\boldsymbol{y}_t) - \nabla f(\boldsymbol{x}_t) - \Delta_t \rangle \\
&\quad + \eta_t^2 \|\nabla f(\boldsymbol{y}_t) - \nabla f(\boldsymbol{x}_t) - \Delta_t\|^2 \\
&\overset{(b)}{\le} \|\boldsymbol{x}_t - \boldsymbol{y}_t\|^2 + 2\eta_t \langle \boldsymbol{x}_t - \boldsymbol{y}_t, \Delta_t \rangle + \eta_t^2 \|\nabla f(\boldsymbol{y}_t) - \nabla f(\boldsymbol{x}_t) - \Delta_t\|^2 \\
&\overset{(c)}{\le} \|\boldsymbol{x}_t - \boldsymbol{y}_t\|^2 + 2\delta\eta_t \|\boldsymbol{x}_t - \boldsymbol{y}_t\| + 3\eta_t^2(2G^2 + \delta^2),
\end{aligned}$$

where $(a)$ is due to the non-expansiveness of the projection step, $(b)$ is due to convexity, and $(c)$ is due to the inequality $\|v_1 + v_2 + v_3\| \le 3\|v_1\|^2 + 3\|v_2\|^2 + 3\|v_3\|^2$. Denoting $d_t := \|\boldsymbol{x}_t - \boldsymbol{y}_t\|$, we obtain

$$d_T^2 \le 2\delta \sum_{t=0}^{T-1} \eta_t d_t + 3(2G^2 + \delta^2) \sum_{t=0}^{T-1} \eta_t^2. \tag{F.4}$$

We now prove (F.3) by induction. If $d_T \le \max_{t=0,\dots,T-1} d_t$, then the conclusion follows from the induction hypothesis. Hence we may assume that $d_T > \max_{t=0,\dots,T-1} d_t$. Then the following inequality follows from (F.4):

$$\begin{aligned}
d_T^2 &\le 2\delta \sum_{t=0}^{T-1} \eta_t d_t + 3(2G^2 + \delta^2) \sum_{t=0}^{T-1} \eta_t^2 \\
&\le 2\delta d_T \cdot \sum_{t=0}^{T-1} \eta_t + 3(2G^2 + \delta^2) \sum_{t=0}^{T-1} \eta_t^2.
\end{aligned}$$

Solving this, we obtain the desired conclusion (F.3). $\qquad\square$

**Theorem 27. (Upper Bound)** *For $G = O(1)$ and $D = O(1)$, let $f$ be an $G$-Lipschitz convex cost function whose optimum lies in a ball of radius $D$. Let $\varepsilon > 0$ and $\delta > 0$ are such that $\delta \leq \frac{\varepsilon}{2D}$. Let $\varepsilon > 0$ be a small constant, and $T$ be a given number of iterations. Under the non-stochastic inexact gradient model, there exists a FOI algorithm whose $(\varepsilon, \delta)$-deviation is $O(\frac{1}{T\varepsilon^2} + \frac{\delta^2}{\varepsilon^2})$, provided that $T = \Omega(1/\varepsilon^2)$.*

*Proof.* We consider the projected gradient descent with constant stepsize $\eta$ onto the ball of radius $D$ that contains the optimum $\boldsymbol{x}_*$.

Let $\boldsymbol{y}_t$ denote the iterate before projection. Let $\Delta_t$ denote the error due to the non-stochastic inexact gradient model at iteration $t$, i.e., $\Delta_t := g(\boldsymbol{x}_t) - \nabla f(\boldsymbol{x}_t)$. Then, we have

$$\frac{1}{2}\|\boldsymbol{x}_{t+1} - \boldsymbol{x}_*\|^2 - \frac{1}{2}\|\boldsymbol{x}_t - \boldsymbol{x}_*\|^2 \leq \frac{1}{2}\|\boldsymbol{y}_{t+1} - \boldsymbol{x}_*\|^2 - \frac{1}{2}\|\boldsymbol{x}_t - \boldsymbol{x}_*\|^2$$

$$\leq -\eta \langle \nabla f(\boldsymbol{x}_t) + \Delta_t, \boldsymbol{x}_t - \boldsymbol{x}_* \rangle + \frac{1}{2}\eta^2 \|\nabla f(\boldsymbol{x}_t) + \Delta_t\|^2$$

Hence,

$$f(\boldsymbol{x}_t) - f(\boldsymbol{x}_*) + \frac{1}{2\eta}\|\boldsymbol{x}_{t+1} - \boldsymbol{x}_*\|^2 - \frac{1}{2\eta}\|\boldsymbol{x}_t - \boldsymbol{x}_*\|^2$$

$$= f(\boldsymbol{x}_t) - f(\boldsymbol{x}_*) - \langle \nabla f(\boldsymbol{x}_t) + \Delta_t, \boldsymbol{x}_t - \boldsymbol{x}_* \rangle + \frac{1}{2}\eta \|\nabla f(\boldsymbol{x}_t) + \Delta_t\|^2$$

$$\leq \delta D + \eta(G + \delta)^2.$$

After telescoping the above inequalities from $t = 0, \ldots, T - 1$, we obtain the bound

$$f\left(\frac{1}{T}\sum_{t=0}^{T-1} \boldsymbol{x}_t\right) - f(\boldsymbol{x}_*) \lesssim \frac{D^2}{\eta T} + \eta G^2 + \delta D.$$

Thus, for $\varepsilon$-accuracy, we need $\eta \lesssim \varepsilon$ and $\eta T \gtrsim 1/\varepsilon$, since the theorem statement assumed that $\delta \leq \frac{\varepsilon}{2D}$. Hence, choosing $\eta = \Theta(\frac{1}{\varepsilon T})$, Lemma 6 gives

$$\|\boldsymbol{x}_t - \boldsymbol{x}'_t\|^2 \lesssim \frac{t}{\varepsilon^2 T^2} + (t\delta\eta)^2 = \frac{t}{\varepsilon^2 T^2} + \frac{\delta^2 t^2}{\varepsilon^2 T^2}.$$

Hence,

$$\|\bar{\boldsymbol{x}}_T - \bar{\boldsymbol{x}}'_T\|^2 \leq \frac{1}{T}\sum_t \|x_t - x'_t\|^2$$

$$\lesssim \frac{1}{T}\sum_t \left[\frac{t}{\varepsilon^2 T^2} + \frac{\delta^2 t^2}{\varepsilon^2 T^2}\right] \approx \frac{1}{\varepsilon^2 T} + \frac{\delta^2}{\varepsilon^2},$$

as desired. $\qquad\square$

### F.3 Inexact initialization model

**Theorem 28. (Upper Bound)** *Let $f$ be an $O(1)$-Lipschitz convex cost function. Let $\varepsilon > 0$ be a small constant, and $T$ be a given number of iterations. Then there exists a FOI algorithm whose $(\varepsilon, \delta)$-deviation is $O(\frac{1}{T\varepsilon^2} + \delta^2)$, provided that $T = \Omega(1/\varepsilon^2)$.*

*Proof.* We consider the subgradient descent with constant step size $\eta$. A standard convergence guarantee for GD reads (see, e.g., [Bubeck, 2014, Theorem 3.2])

$$f(\bar{\boldsymbol{x}}_T) - f(\boldsymbol{x}_*) \lesssim \frac{\|\boldsymbol{x}_0 - \boldsymbol{x}_*\|^2}{\eta T} + \eta G^2.$$

Hence, in order to have $\varepsilon$-suboptimality, we need to have $\eta T \approx \frac{1}{\varepsilon}$ and $\eta \lesssim \varepsilon$.

We now derive a deviation bound. A similar calculation to Lemma 6 yields the following:

$$\|\boldsymbol{y}_{t+1} - \boldsymbol{x}_{t+1}\|^2 \leq \|\boldsymbol{x}_t - \boldsymbol{y}_t\|^2 - 2\eta \langle \boldsymbol{y}_t - \boldsymbol{x}_t, \nabla f(\boldsymbol{y}_t) - \nabla f(\boldsymbol{x}_t) \rangle + \eta^2 \|\nabla f(\boldsymbol{y}_t) - \nabla f(\boldsymbol{x}_t)\|^2$$

$$\leq \|\boldsymbol{x}_t - \boldsymbol{y}_t\|^2 + \eta^2 \|\nabla f(\boldsymbol{y}_t) - \nabla f(\boldsymbol{x}_t)\|^2 \leq \|\boldsymbol{x}_t - \boldsymbol{y}_t\|^2 + 4G^2\eta^2.$$

Hence, it holds that

$$\|\boldsymbol{x}_t - \boldsymbol{y}_t\|^2 \le \|\boldsymbol{x}_0 - \boldsymbol{y}_0\|^2 + G^2 \sum_{t=0}^{t-1} \eta^2 \le \delta^2 + G^2 t\eta^2 = \delta^2 + G^2 \frac{t}{T^2}(\eta T)^2 \approx \delta^2 + G^2 \frac{t}{T^2} \cdot \frac{1}{\varepsilon^2}\,.$$

Thus, it follows that

$$\|\bar{\boldsymbol{x}}_t - \bar{\boldsymbol{y}}_T\|^2 \le \frac{1}{T}\sum_{t=0}^{T-1}\|\boldsymbol{x}_t - \boldsymbol{y}_t\|^2 \lesssim \frac{1}{T}\sum_{t=0}^{T-1}\left(\delta^2 + G^2 \frac{t}{T^2}\frac{1}{\varepsilon^2}\right) \lesssim \delta^2 + \frac{1}{\varepsilon^2 T}\,,$$

as desired. $\qquad\square$

### F.4 Stochastic inexact gradient model (strongly convex costs)

**Theorem 29. (Upper Bound)** *Let $f$ be an $O(1)$-Lipschitz $\mu$-strongly convex cost function. Let $\varepsilon > 0$ be a small constant, and $T$ be a given number of iterations. Under the stochastic inexact gradient model, there exists a FOI algorithm whose $(\varepsilon, \delta)$-deviation is $O(\frac{1}{T\mu^2} \wedge \frac{\varepsilon}{\mu})$, provided that $T = \Omega(1/\varepsilon)$.*

*Proof.* The standard convergence rate bound (e.g., [Bubeck, 2014, Theorem 6.2]) implies that SGD with $\eta_t = \frac{2}{\mu(t+1)}$ satisfies

$$\mathbb{E}f\left(\sum_{t=1}^{T}\frac{2t}{T(T+1)}\boldsymbol{x}_t\right) - f(\boldsymbol{x}^*) \lesssim \frac{2G^2}{\mu(T+1)}\,, \tag{F.5}$$

where $G$ is the Lipschitz constant of $f$. Hence, letting $\bar{\boldsymbol{x}}_T := \sum_{t=1}^{T}\frac{2t}{T(T+1)}\boldsymbol{x}_t$, it follows that

$$\mathbb{E}\|\bar{\boldsymbol{x}}_T - \boldsymbol{x}^*\|^2 \lesssim \frac{1}{T\mu^2} \wedge \frac{\varepsilon}{\mu}\,,$$

where $\frac{\varepsilon}{\mu}$ follows from the fact that $\bar{\boldsymbol{x}}_T$ achieves $\varepsilon$-accuracy. $\qquad\square$

### F.5 Non-stochastic inexact gradient model (strongly convex costs)

**Theorem 30. (Upper Bound)** *For $G = O(1)$ and $D = O(1)$, let $f$ be an $G$-Lipschitz $\mu$-strongly convex cost function whose optimum lies in a ball of radius $D$. Let $\varepsilon > 0$ and $\delta > 0$ are such that $\delta \le \frac{\varepsilon}{2D}$. Let $\varepsilon > 0$ be a small constant, and $T$ be a given number of iterations. Under the non-stochastic inexact gradient model, there exists a FOI algorithm whose $(\varepsilon, \delta)$-deviation is $O((\frac{1}{T\mu^2} + \frac{\delta^2}{\mu^2}) \wedge \frac{\varepsilon}{\mu})$, provided that $T = \Omega(1/\varepsilon)$.*

*Proof.* We first prove the convergence rate bound. We run projected gradient descent with a constant learning rate $\eta_t = \frac{1}{\mu(t+1)}$. Then, it follows that

$$\frac{\mu(t+1)}{2}\|\boldsymbol{x}_{t+1} - \boldsymbol{x}^*\|^2 - \frac{\mu t}{2}\|\boldsymbol{x}_t - \boldsymbol{x}^*\|^2$$

$$= \frac{\mu}{2}\|\boldsymbol{x}_t - \boldsymbol{x}^*\|^2 + \frac{1}{2\eta_t}\left(\|\boldsymbol{x}_{t+1} - \boldsymbol{x}^*\|^2 - \|\boldsymbol{x}_t - \boldsymbol{x}^*\|^2\right)$$

$$= \frac{\mu}{2}\|\boldsymbol{x}_t - \boldsymbol{x}^*\|^2 + \langle g(\boldsymbol{x}_t), \boldsymbol{x}^* - \boldsymbol{x}_t\rangle + \frac{\eta_t}{2}\|g(\boldsymbol{x}_t)\|^2$$

$$\le \frac{\mu}{2}\|\boldsymbol{x}_t - \boldsymbol{x}^*\|^2 + \langle \nabla f(\boldsymbol{x}_t), \boldsymbol{x}^* - \boldsymbol{x}_t\rangle + \frac{\eta_t}{2}\|g(\boldsymbol{x}_t)\|^2 + \delta D\,,$$

where the last line follows since every iterate lies in the ball of radius $D$. Hence,

$$f(\boldsymbol{x}_t) - f(\boldsymbol{x}^*) + \frac{\mu(t+1)}{2}\|\boldsymbol{x}_{t+1} - \boldsymbol{x}^*\|^2 - \frac{\mu t}{2}\|\boldsymbol{x}_t - \boldsymbol{x}^*\|^2$$

$$\le \underbrace{f(\boldsymbol{x}_t) - f(\boldsymbol{x}^*) + \frac{\mu}{2}\|\boldsymbol{x}_t - \boldsymbol{x}^*\|^2 + \langle \nabla f(\boldsymbol{x}_t), \boldsymbol{x}^* - \boldsymbol{x}_t\rangle}_{} + \frac{\eta_t}{2}\|g(\boldsymbol{x}_t)\|^2 + \delta D$$

$$\overset{(a)}{\le} \frac{\eta_t}{2}\|g(\boldsymbol{x}_t)\|^2 + \delta D \lesssim \frac{\eta_t}{2}G^2 + \delta D\,,$$

where $(a)$ follows from strong convexity. Therefore, it holds that

$$f\left(\sum_{t=1}^{T}\frac{2(t+1)}{(T+1)(T+2)}\boldsymbol{x}_t\right) - f(\boldsymbol{x}^*) \le \sum_{t=1}^{T}\frac{2(t+1)}{(T+1)(T+2)}(f(\boldsymbol{x}_t) - f(\boldsymbol{x}^*))$$

$$\lesssim \sum_{t=1}^{T}\frac{2(t+1)}{(T+1)(T+2)}(\frac{\eta_t}{2}G^2 + \delta D) \lesssim \frac{G^2}{\mu T} + \delta D.$$

Let $\bar{\boldsymbol{x}}_T := \sum_{t=1}^{T}\frac{2t}{T(T+1)}\boldsymbol{x}_t$. We next bound the deviation. Again, let $\{\boldsymbol{y}_t\}$ be the iterates of (projected) GD with stepsize $\eta_t$ with exact gradients and $\{\boldsymbol{x}_t\}$ be the iterates of (projected) GD with the same stepsize with inexact gradients with noise $\{\Delta_t\}$.

$$\|\boldsymbol{y}_{t+1} - \boldsymbol{x}_{t+1}\|^2 \overset{(a)}{\le} \|\boldsymbol{y}_t - \eta_t\nabla f(\boldsymbol{y}_t) - (\boldsymbol{x}_t - \eta_t(\nabla f(\boldsymbol{x}_t) + \Delta_t))\|^2$$

$$= \|\boldsymbol{x}_t - \boldsymbol{y}_t\|^2 - 2\eta_t\langle \boldsymbol{y}_t - \boldsymbol{x}_t, \nabla f(\boldsymbol{y}_t) - \nabla f(\boldsymbol{x}_t) - \Delta_t\rangle$$

$$+ \eta_t^2\|\nabla f(\boldsymbol{y}_t) - \nabla f(\boldsymbol{x}_t) - \Delta_t\|^2$$

$$\overset{(b)}{\le} (1 - 2\mu\eta_t)\|\boldsymbol{x}_t - \boldsymbol{y}_t\|^2 + 2\eta_t\langle \boldsymbol{x}_t - \boldsymbol{y}_t, \Delta_t\rangle + \eta_t^2\|\nabla f(\boldsymbol{y}_t) - \nabla f(\boldsymbol{x}_t) - \Delta_t\|^2$$

$$\overset{(c)}{\lesssim} (1 - \frac{2}{t+1})\|\boldsymbol{x}_t - \boldsymbol{y}_t\|^2 + 2\delta\eta_t\|\boldsymbol{x}_t - \boldsymbol{y}_t\| + \eta_t^2 G^2,$$

where $(a)$ is due to the non-expansiveness of the projection step, $(b)$ is due to convexity, and $(c)$ is due to the inequality $\|v_1 + v_2 + v_3\| \le 3\|v_1\|^2 + 3\|v_2\|^2 + 3\|v_3\|^2$. Denoting $d_t := \|\boldsymbol{x}_t - \boldsymbol{y}_t\|$, we obtain

$$d_T^2 \lesssim \delta\sum_{t=0}^{T-1}\frac{t^2}{T^2}\eta_t d_t + \sum_{t=0}^{T-1}\frac{t^2}{T^2}\eta_t^2. \tag{F.6}$$

Now similarly to Lemma 6, one can deduce from this inequality that

$$d_T^2 \lesssim \frac{\delta}{\mu}d_T + \frac{1}{T\mu^2} \implies d_T^2 \lesssim \frac{\delta^2}{\mu^2} + \frac{1}{T\mu^2}.$$

Now, after applying the Jensen's inequality, we obtain the desired deviation bound of $\|\bar{\boldsymbol{x}}_T - \bar{\boldsymbol{x}}_T'\|^2 \le (\frac{\delta^2}{\mu^2} + \frac{1}{T\mu^2}) \wedge \frac{\varepsilon}{\mu}$, where $\frac{\varepsilon}{\mu}$ follows from the fact that $\bar{\boldsymbol{x}}_T$ achieves $\varepsilon$-accuracy. $\qquad\square$

## F.6 Inexact initialization model (strongly convex costs)

**Theorem 31. (Upper Bound)** *Let $f$ be an $O(1)$-Lipschitz $\mu$-strongly convex cost function. Let $\varepsilon > 0$ be a small constant, and $T$ be a given number of iterations. Then there exists a FOI algorithm whose $(\varepsilon, \delta)$-deviation is $O(\frac{1}{T\mu^2} \wedge \frac{\varepsilon}{\mu})$, provided that $T = \Omega(1/\varepsilon)$.*

*Proof.* The standard convergence rate bound (e.g., [Bansal and Gupta, 2019, Theorem 2.4]) implies that GD with $\eta_t = \frac{2}{\mu(t+1)}$ satisfies

$$f\left(\sum_{t=1}^{T}\frac{2t}{T(T+1)}\boldsymbol{x}_t\right) - f(\boldsymbol{x}^*) \lesssim \frac{2G^2}{\mu(T+1)}, \tag{F.7}$$

where $G$ is the Lipschitz constant of $f$. Hence, letting $\bar{\boldsymbol{x}}_T := \sum_{t=1}^{T}\frac{2t}{T(T+1)}\boldsymbol{x}_t$, it follows that

$$\|\bar{\boldsymbol{x}}_T - \boldsymbol{x}^*\|^2 \lesssim \frac{1}{T\mu^2} \wedge \frac{\varepsilon}{\mu},$$

where $\frac{\varepsilon}{\mu}$ follows from the fact that $\bar{\boldsymbol{x}}_T$ achieves $\varepsilon$-accuracy. $\qquad\square$

# G    Proof of upper bound for finite-sum setting (Theorem 4)

Recall Theorem 4:

**Theorem 4.** *For $G = O(1)$ and $D = O(1)$, let $f_i$ be an G-Lipschitz convex cost function for each $i \in [m]$, and assume that the optimum of $f$ lies in a ball of radius $D$. Let $\varepsilon, \delta > 0$ be given parameters such that $\delta \leq \varepsilon/(2D)$, and $T = \Omega(1/\varepsilon^2)$ be a given number of iterations. Define the SGD updates as follows: initialize $\boldsymbol{x}_0 = 0$, and for $t = 0, 1, \ldots, T-1$, set $\boldsymbol{x}_{t+1} = \boldsymbol{x}_t - \eta_t g_{i_t}(\boldsymbol{x}_t)$ where $i_t \sim [n]$ uniformly at random. Under the inexact component gradient oracle (Definition 4), the average iterate $\bar{\boldsymbol{x}}_T$ of SGD with stepsize $\eta = \Theta(1/(\varepsilon T))$ satisfies $\mathbb{E} f(\bar{\boldsymbol{x}}_T) - \inf_{\boldsymbol{x} \in domf} f(\boldsymbol{x}) \leq \varepsilon$ and $\mathbb{E} \|\bar{\boldsymbol{x}}_T - \bar{\boldsymbol{x}}'_T\|^2 = O(1/(T\varepsilon^2) + \delta^2/\varepsilon^2)$, where $\bar{\boldsymbol{x}}'_T$ is the output of an independent run of SGD.*

*Proof.* We first prove a deviation bound similar to that of Lemma 6. Let us denote $\Delta_t := g_{i_t}(\boldsymbol{x}_t) - \nabla f_{i_t}(\boldsymbol{x}_t)$. Let $\{\boldsymbol{y}_t\}$ be the iterates of (projected) GD with inexact gradients. Then we have

$$\mathbb{E}_{i_t} \|\boldsymbol{x}_{t+1} - \boldsymbol{y}_{t+1}\|^2$$

$$\overset{(a)}{\leq} \mathbb{E}_{i_t} \|\boldsymbol{x}_t - \eta_t g_{i_t}(\boldsymbol{x}_t) - (y_t - \eta_t(\nabla f(y_t)))\|^2$$

$$= \|\boldsymbol{x}_t - y_t\|^2 - 2\eta_t \mathbb{E}_{i_t} \langle \boldsymbol{x}_t - y_t, g_{i_t}(\boldsymbol{x}_t) - \nabla f(y_t)\rangle + \eta_t^2 \mathbb{E}_{i_t} \|g_{i_t}(\boldsymbol{x}_t) - \nabla f(y_t)\|^2$$

$$= \|\boldsymbol{x}_t - y_t\|^2 - 2\eta_t \mathbb{E}_{i_t} \langle \boldsymbol{x}_t - y_t, \nabla f_{i_t}(\boldsymbol{x}_t) + \Delta_t - \nabla f(y_t)\rangle + \eta_t^2 \mathbb{E}_{i_t} \|g_{i_t}(\boldsymbol{x}_t) - \nabla f(y_t)\|^2$$

$$= \|\boldsymbol{x}_t - y_t\|^2 - 2\eta_t \langle \boldsymbol{x}_t - y_t, \nabla f(\boldsymbol{x}_t) + \Delta_t - \nabla f(y_t)\rangle + \eta_t^2 \mathbb{E}_{i_t} \|g_{i_t}(\boldsymbol{x}_t) - \nabla f(y_t)\|^2$$

$$\overset{(b)}{\leq} \|\boldsymbol{x}_t - y_t\|^2 - 2\eta_t \langle \boldsymbol{x}_t - y_t, \Delta_t\rangle + \eta_t^2 \mathbb{E}_{i_t} \|\nabla f_{i_t}(\boldsymbol{x}_t) + \Delta_t - \nabla f(y_t)\|^2$$

$$\overset{(c)}{\leq} \|\boldsymbol{x}_t - y_t\|^2 + 2\delta\eta_t \|\boldsymbol{x}_t - y_t\| + 3\eta_t^2(2G^2 + \delta^2),$$

where $(a)$ is due to the non-expansiveness of the projection step, and $(b)$ is due to convexity, and $(c)$ is due to the inequality $\|v_1 + v_2 + v_3\| \leq 3\|v_1\|^2 + 3\|v_2\|^2 + 3\|v_3\|^2$. Taking expectations on both sides, we obtain

$$\mathbb{E} \|\boldsymbol{x}_{t+1} - y_{t+1}\|^2 \leq \mathbb{E} \|\boldsymbol{x}_t - y_t\|^2 + 2\delta\eta_t \mathbb{E} \|\boldsymbol{x}_t - y_t\| + 3\eta_t^2(2G^2 + \delta^2)$$

$$\leq \mathbb{E} \|\boldsymbol{x}_t - y_t\|^2 + 2\delta\eta_t \sqrt{\mathbb{E} \|\boldsymbol{x}_t - y_t\|^2} + 3\eta_t^2(2G^2 + \delta^2),$$

Denoting $d_t := \sqrt{\mathbb{E} \|\boldsymbol{x}_t - y_t\|^2}$ and telescoping the above inequality, we obtain

$$d_T^2 \leq 2\delta \sum_{t=0}^{T-1} \eta_t d_t + 3(2G^2 + \delta^2) \sum_{t=0}^{T-1} \eta_t^2. \tag{G.1}$$

This is precisely equal to (F.4) from the proof of Lemma 6. Following the same recursion, we obtain the following bound:

$$d_T \leq \sqrt{3(2G^2 + \delta^2) \cdot \sum_{t=0}^{T-1} \eta_t^2 + 2\delta \sum_{t=0}^{T-1} \eta_t}.$$

Squaring both sides, we obtain

$$\mathbb{E} \|\boldsymbol{x}_T - y_T\|^2 \leq 6(2G^2 + \delta^2) \cdot \sum_{t=0}^{T-1} \eta_t^2 + 8\delta^2 \left(\sum_{t=0}^{T-1} \eta_t\right)^2. \tag{G.2}$$

We next prove the bound on the convergence rate.

**Convergence rate bound.** We consider the projected gradient descent with constant stepsize $\eta$ onto the ball of radius $D$ that contains the optimum $\boldsymbol{x}_*$. Let $z_t$ denote the iterate before projection. As

before, let $\Delta_t := g_{i_t}(\boldsymbol{x}_t) - \nabla f_{i_t}(\boldsymbol{x}_t)$. Then, we have

$$
\begin{aligned}
\frac{1}{2} \mathop{\mathbb{E}}_{i_t} \|\boldsymbol{x}_{t+1} - \boldsymbol{x}_*\|^2 - \frac{1}{2} \|\boldsymbol{x}_t - \boldsymbol{x}_*\|^2 &\leq \frac{1}{2} \mathop{\mathbb{E}}_{i_t} \|z_{t+1} - \boldsymbol{x}_*\|^2 - \frac{1}{2} \|\boldsymbol{x}_t - \boldsymbol{x}_*\|^2 \\
&= \frac{1}{2} \mathop{\mathbb{E}}_{i_t} \|\boldsymbol{x}_t - \eta_t g_{i_t}(\boldsymbol{x}_t) - \boldsymbol{x}_*\|^2 - \frac{1}{2} \|\boldsymbol{x}_t - \boldsymbol{x}_*\|^2 \\
&= -2\eta_t \mathop{\mathbb{E}}_{i_t} \langle \nabla f_{i_t}(\boldsymbol{x}_t) + \Delta_t, \boldsymbol{x}_t - \boldsymbol{x}_* \rangle + \frac{1}{2}\eta_t^2 \mathop{\mathbb{E}}_{i_t} \|\nabla f_{i_t}(\boldsymbol{x}_t) + \Delta_t\|^2 \\
&= -2\eta_t \langle \nabla f(\boldsymbol{x}_t) + \Delta_t, \boldsymbol{x}_t - \boldsymbol{x}_* \rangle + \frac{1}{2}\eta_t^2 \mathop{\mathbb{E}}_{i_t} \|\nabla f_{i_t}(\boldsymbol{x}_t) + \Delta_t\|^2 \\
&\leq -2\eta_t \langle \nabla f(\boldsymbol{x}_t) + \Delta_t, \boldsymbol{x}_t - \boldsymbol{x}_* \rangle + \eta_t^2 (G^2 + \delta^2).
\end{aligned}
$$

Hence,

$$
\begin{aligned}
&f(\boldsymbol{x}_t) - f(\boldsymbol{x}_*) + \frac{1}{2\eta_t} \mathop{\mathbb{E}}_{i_t} \|\boldsymbol{x}_{t+1} - \boldsymbol{x}_*\|^2 - \frac{1}{2\eta_t} \|\boldsymbol{x}_t - \boldsymbol{x}_*\|^2 \\
&\qquad \leq f(\boldsymbol{x}_t) - f(\boldsymbol{x}_*) - \langle \nabla f(\boldsymbol{x}_t) + \Delta_t, \boldsymbol{x}_t - \boldsymbol{x}_* \rangle + \eta_t (G^2 + \delta^2) \\
&\qquad \leq \delta D + \eta_t (G^2 + \delta^2).
\end{aligned}
$$

Choosing $\eta_t \equiv \eta$ and after telescoping the above inequalities from $t = 0, \ldots, T-1$, we obtain the bound

$$
\mathbb{E} f\left(\frac{1}{T} \sum_{t=0}^{T-1} \boldsymbol{x}_t\right) - f(\boldsymbol{x}_*) \lesssim \frac{D^2}{\eta T} + \eta G^2 + \delta D.
$$

Thus, for $\varepsilon$-accuracy, we need $\eta \lesssim \varepsilon$ and $\eta T \gtrsim 1/\varepsilon$, since the theorem statement assumed that $\delta \leq \frac{\varepsilon}{2D}$. Hence, choosing $\eta = \Theta(\frac{1}{\varepsilon T})$, the deviation bound we proved gives

$$
\mathbb{E} \|\boldsymbol{x}_t - \boldsymbol{x}_t'\|^2 \lesssim \frac{t}{\varepsilon^2 T^2} + (t\delta\eta)^2 = \frac{t}{\varepsilon^2 T^2} + \frac{\delta^2 t^2}{\varepsilon^2 T^2}.
$$

Hence,

$$
\begin{aligned}
\mathbb{E} \|\bar{\boldsymbol{x}}_T - \bar{\boldsymbol{x}}_T'\|^2 &\leq \frac{1}{T} \sum_t \mathbb{E} \|x_t - x_t'\|^2 \\
&\lesssim \frac{1}{T} \sum_t \left[\frac{t}{\varepsilon^2 T^2} + \frac{\delta^2 t^2}{\varepsilon^2 T^2}\right] \approx \frac{1}{\varepsilon^2 T} + \frac{\delta^2}{\varepsilon^2},
\end{aligned}
$$

as desired. $\qquad\square$