# OpenReview forum: "Reproducibility in Optimization: Theoretical Framework and Limits"
_NeurIPS.cc/2022/Conference — NeurIPS 2022 Accept_

### Official Review · Reviewer_YXF1 · 2022-07-10

**Rating:** 7
**Confidence:** 4
**Soundness:** 3 good
**Presentation:** 3 good
**Contribution:** 3 good

**Summary:**

This paper considers the reproducibility of convex optimization algorithms, meaning roughly how much the parameters output by an optimization algorithm can change when the algorithm is run a second (on the same data). The paper studies the reproducibility of first-order convex optimization algorithms under three different types of noise/errors: stochastic gradient noise, inexact gradient computations, and inexact initialization. In all cases, the paper shows that such noise/errors can lead the output of two independent runs of the same algorithm to differ by an amount which depends mainly on (1) the amount of noise/errors, (2) the final optimization accuracy, and (3) the number of iterations used by the algorithm. Tight upper and lower bounds are proven in a number of settings, and one of the main upshots is that in order to match the reproducibility lower bounds, it is necessary to run an optimization algorithm for more iterations than are needed just to reach the desired accuracy.

**Questions:**

This work looks at something similar to the classic idea of numerical stability. I am not personally super familiar with this literature, but I would be shocked if the numerical stability of various optimization algorithms had not been studied before, and if it has been, it would be good to discuss similarities/differences in the related work section. Have you searched for this type of paper? Is there just nothing relevant?

**Limitations:**

See weaknesses above.

**Strengths And Weaknesses:**

Strengths:
This is an interesting paper with a somewhat new perspective on convex optimization algorithms, and the paper is well-written and clear in how it presents its ideas. I read through the proofs of the theoretical results, and they appear correct to the best of my knowledge. The identified tradeoff between reproducibility and computation is quite intriguing---I have not seen anything like this before, and it seems like something that is worthy of additional investigation. Overall, I think this paper brings some interesting new ideas to the optimization community.

Weaknesses:
I think that at several points in the paper, it would be useful to make things more concrete with some examples. I feel this most strongly in the discussion of inexact initialization and the (non-stochastic) inexact gradient oracle. In particular, it seems from the motivation that the main source of inexactness would come essentially from numerical errors? If so, it seems to me that \delta in this case should be extremely small, roughly machine precision, or perhaps machine precision times the square root of the dimension. If this is the case, the irreproducibility level shown in Theorems 2/3 in all cases excluding the non-smooth, non-strongly convex case seem like they would be quite small, and it's not clear to me that we would have anything to worry about here.

In the strongly convex settings, I feel that the level of irreproducibility is somewhat exaggerated in the prose below the theorems. E.g. around line 283, I don't think that it is the least bit surprising that irreproducibility can manifest at all---of course deviations in the gradients will affect the solution *some* amount, although presumably not too much. And indeed, the level of irreproducibility shown is at most epsilon, which seems quite small, and probably nothing to worry about!

I am also a little skeptical about the formulation of the stochastic optimization oracle and the "optimizing the population loss" settings. In particular, if you are someone that is worried about reproducibility, then wouldn't you want to fix the order in which you use your samples? In this case, although there still might be some discrepancy between different runs (due to, e.g., non-stochastic inexactness in the gradient computations, inexact initialization, etc), it would likely be at a much smaller level since corresponding gradients calculated by each run of the algorithm would be based on the same sample.

Finally, I have one slightly unfair complaint, which I don't weigh too heavily since it is a little unfair, but I want to mention it in case there might be a way to address this in future versions of the paper. Essentially, I found all of the results in the paper to be fairly obvious. Although there is a bit of a difference between looking at the iterate distance from the solution \| x_f - x* \|^2 and the reproducibility \| x_f - x'_f \|^2, these are very similar quantities, and unless I am mistaken, all of the results in this paper can basically be rephrased as saying that these quantities are within a constant factor of each other in the worst case (e.g. I'm pretty sure that Theorem 1 holds as stated also for the quantity \mathbb{E} \| x_f - x* \|^2).

Typos:
line 249-250: "(epsilon)-deviation bounds for parametter reproducibility immediately transform into (epsilon, delta)- deviation bounds for parameter reproducibility"
line 297: the lower bound is 1/eps + delta^2/eps^2 in the smooth setting, not 1/eps^2

---

> ### Author Response · Authors · 2022-07-30
> **Thank you!**
>
> Thank you for your questions! We answer them one by one below.
>
> 1. We study an exhaustive characterization of $(\epsilon,\delta)$-deviation for different settings in order to identify for which case the irreproducibility might be an issue.  Although your interpretation is largely correct because $\delta$ is quite small, we’d like to mention that in modern giant machine learning models with ~100 billions of parameters, $\delta$ which roughly scales as (machine precision)$\times$(dimension)$^{1/2}$ could be quite large indeed. Moreover, our results show that for non-smooth costs, even very small $\delta$ can lead to large irreproducibility.
>
> 2. For strongly convex costs, we agree that the deviation is much smaller than other cases due to the existence of a unique minimizer. We’ll modify our prose as per your comment. However, we’d also like to highlight that it is still nontrivial to get the exact dependency on $\delta$, $\epsilon$, $T$ and provide matching upper and lower bounds, as we did in our paper. In fact, some of the lower bound proofs require more delicate constructions than those of the non-strongly-convex cases.
>
> 3. Regarding the results in Section 6 “Optimization for Machine Learning”: We agree that fixing the order of minibatches would ensure higher reproducibility. However, in practice, fixing the order is not a practical choice: many works have shown that the stochasticity in minibatch helps improve generalization in practice. Moreover, in applications like federated learning, it is hard to control the order of examples since clients connect to the server at random times that are beyond our control (see e.g. [Kairouz et al.](https://arxiv.org/abs/1912.04977)). Hence, we believe that studying reproducibility under the stochasticity in minibatch is still an important question.
>
> 4. We would like to highlight that $x^*$ may not even be unique without strong convexity,  and many of our analyses do not rely on the relation between $||x_f - x^* ||^2$ and $|| x_f - x'_f ||^2$. For instance, in the proof of Theorem 6, we directly lower bound $var(x_T)$ instead of $|| x_f - x^* ||^2$. For the same reason, we disagree with your assessment “Theorem 1 holds as stated also for the quantity $|| x_f - x^* ||^2$” because again for non-strongly convex costs the minima may not be unique. We are surprised that the reviewer found our results obvious since obtaining tight upper and lower bounds was quite challenging in several cases (for instance, see Appendices B and F) and the form of the deviation bounds was not obvious to us. If the reviewer has an easier way to derive our bounds, we would appreciate it if they could provide the easy analysis in an update to the review.
>
>     Moreover, while the upper bound analyses might seem similar to stability analyses of gradient based algorithms (in the context of generalization bounds and differential privacy), the lower bounds certainly are novel and it is in fact surprising that simple algorithms like GD and SGD are already optimal in terms of the convergence rate vs reproducibility tradeoff.
>
> 5. We haven’t found any results in the numerical analysis literature that theoretically study the deviation in terms of $\epsilon$, $\delta$, and $T$. To the best of our knowledge, the primary focus in the numerical stability community has been on important linear algebraic algorithms (e.g., solving linear systems, matrix inversion, exponentiation, square root etc) and less on general purpose convex optimization.
>
>    Apart from the numerical stability literature, some works from the optimization literatures (e.g.  [d'Aspremont](https://arxiv.org/abs/math/0512344) or [Devolder et al.](https://link.springer.com/article/10.1007/s10107-013-0677-5)) study similar inexact gradient oracle models but their main result is about characterizing the convergence rate under inexact oracle instead of the deviation in the iterates as we did in our work. Also, we noticed a similar formulation of deviation from control/dynamical systems literature in the form of input-to-state stability, but those results usually deal with general dynamical systems with different motivations.

---

> > ### Comment · Reviewer_YXF1 · 2022-08-04
> > **Thanks**
> >
> > Thank you for the response. This response has basically addressed my concerns, and I have upgraded my score accordingly. I think that the paper would benefit from some of this discussion being added (e.g. describing how delta might be very large even just from floating point arithmetic errors if you're training some huge model) to make things more concrete.
> >
> > I also take your point that my claim about Theorem 1 holding for the distance to x^* can't be true exactly, and I didn't mean to suggest that Theorem 1 or its proof are trivial. That said, I think || x_f - x_f' ||^2 is at least qualitatively similar to the conventional wisdom about || x_f - x^* ||^2, i.e. that the latter will be small for strongly convex functions, not necessarily small for non-strongly convex functions, that it can be monotonically non-increasing for smooth objectives (at least when the stepsize is sufficiently small) but can be completely unpredictable for non-smooth objectives, etc. I think it might be interesting for the reader and potentially helpful for building intuition if this analogy was made in the paper, but perhaps the authors disagree, in which case fair enough!

---

> > > ### Author Response · Authors · 2022-08-04
> > > **Thank you for your feedback!**
> > >
> > > Thank you for raising the score, and for giving us very useful feedback for improving the presentation of the paper (specifically, the discussion about $\delta$, and the discussion about the role of $||x_f - x^*||^2$ in various settings). We’ll reflect your comments in our final version.

---

### Official Review · Reviewer_Vi1H · 2022-07-12

**Rating:** 8
**Confidence:** 4
**Soundness:** 4 excellent
**Presentation:** 4 excellent
**Contribution:** 4 excellent

**Summary:**

This paper studies the reproducibility in optimization and provides the first theoretical framework to analyze the irreducibility of optimization algorithms in modern machine learning problems. The authors have provided the $(\epsilon, \delta)$ deviation of first-order stochastic optimization algorithms in different problems (strongly convex, convex smooth, convex non smooth), and support the bounds with information-theoretical lower bounds. The paper is mainly theoretical.

**Questions:**

I do have some questions for the authors though.

1. Does the results in Table 1 imply that reproducibility is better (i.e., the deviation is smaller) when we do not converge to the neighborhoods of the optimum(i.e., when $\epsilon$ is large instead of small)? I find it a little bit hard to interpret the results in this direction and I wonder what the authors think. Please correct me if I think it wrong.

2. Although the authors mention that irreducibility for non convex optimization is another potential direction, I do not think it is quite possible to derive an upper bound in the non convex case since convergence in nonconvex problems is (usually) measured by the stationarity in gradients, and any two stationary points can be far away from each other. Therefore, I cannot think of an easy way to adapt the results in this paper to the non convex case. It's worth discussing though.

**Limitations:**

yes.

**Strengths And Weaknesses:**

As a theory person, I like the paper very much. The fundamental problem of irreducibility in machine learning is common and cannot be ignored. However, in real-life people simple re-train the models and use the one with the highest performance. This paper is, as far as I know, the first one that studies the problem rigorously and I really appreciate the results.

Strengths.
  1. The paper is very well-written, with clear notations, nice results, good presentations and easy-to-understand words and sentences. I thank the author for writing such a good paper.
  2. The results are new and very interesting, which help us to understand why ML models are often irreproducible.
  3. The authors have nice analysis results in different cases (strongly convex, convex smooth, convex non smooth), and support all of them with lower bounds.
  4. I believe this work opens many new directions for future study.

Weaknesses.
  I cannot find any obvious weakness of this paper. I believe as a theory paper, it is already good enough. If the authors can have some simple experiments in convex optimization to support their claims, it would be even better. However, I believe the paper is good as it is.

---

> ### Author Response · Authors · 2022-07-30
> **Thank you!**
>
> Thank you for your very encouraging comment! We agree that our results help us understand the irreproducibility in ML and open many interesting directions for future study.
> Let us answer your questions one by one.
>
> 1. Intuitively, you are correct that reproducibility should improve when $\epsilon$ increases because it gets easier to reach the suboptimality level. For example, consider the extreme case where $\epsilon \approx 1$. Then one can simply output an initialization itself (or an output of gradient descent after some small number of steps). Hence, one would expect high reproducibility.
>
>       On the other hand, we’d like to clarify that as the target accuracy $\epsilon$ tends to zero, this intuition no longer holds: the main reason is the appearance of $T$ in the denominator of the reproducibility guarantee. For example, let’s consider the smooth case bound of $\Theta(\frac{\delta^2}{T\epsilon^2})$. Although it may seem that the appearance of $\epsilon$ in the denominator suggest better reproducibility when $\epsilon$ is larger, here note that $T$ has to be at least $\Omega(\frac{1}{\epsilon^2})$ in order to achieve the $\epsilon$-accuracy in the cost. Hence, merely increasing the target suboptimality $\epsilon$ in such a regime does not improve reproducibility. In that case, one way to improve the reproducibility is to increase the number of iterations $T$ relative to the target accuracy $\epsilon$. This can be achieved for instance by using smaller step sizes.
>
> 2. We agree that an extension to general non-convex setting might require a careful formulation precisely due to the issue of multiple stationary points. In the worst case, we agree that the nonconvexity would result in very large irreproducibility. One possible concrete first step is to consider benign non-convex functions like those satisfying PL inequality. This would help us understand the non-convex case when we initialize near a single basin.

---

> > ### Comment · Reviewer_Vi1H · 2022-08-03
> > **Thank you for the rebuttal**
> >
> > I have read the authors’ rebuttal and I appreciate their answers. I don’t have any further questions. Good work!

---

### Official Review · Reviewer_aZPs · 2022-07-12

**Rating:** 6
**Confidence:** 3
**Soundness:** 3 good
**Presentation:** 3 good
**Contribution:** 2 fair

**Summary:**

This paper studies the notion of reproducibility in convex optimization, defined as the requirement that two runs of the algorithm on the same input should produce the same (or similar) output. The authors define a quantitative notion of irreproducibility (the ($\epsilon,\delta$)-deviation) which measures how much an algorithm (that successfully minimizes the objective up to error $\epsilon$) can change the output when the algorithm has an error of up to $\delta$. The errors can come from stochastic gradient oracles, or numerical error, or inexact initialization. For each source of error, the authors compute tight bounds on the ($\epsilon,\delta$)-deviation for first-order methods, in the cases of smooth and non-smooth, convex and strongly convex functions. The authors also show results for optimization problems motivated by machine learning, including finite-sum optimization and stochastic convex optimization, and show the bounds are tight.

**Questions:**

-

**Limitations:**

Yes

**Strengths And Weaknesses:**

This is a nice paper that studies an interesting conceptual question and proposes a quantitative analysis and answers. The paper is well-written and well-motivated, with good discussion of alternative notions. The setup and definition is clear and concise. The results on the tight bounds on the deviations are very nice, and some of the results surprising.

The paper studies a rather simple setting of convex optimization, but in a comprehensive way.

---

> ### Author Response · Authors · 2022-07-30
> **Thank you!**
>
> Thank you for your encouraging comments!

---

### Meta-Review · Area_Chair_RvD3 · 2022-08-26

**Recommendation:** Accept
**Confidence:** Certain

**Metareview:**

The paper studies how the noise inherent in optimization affects “reproducibility,” which the authors measure by the Euclidean distance between two independent runs of the algorithm. The results of the paper reveal fundamental tradeoffs between computation (in terms of gradient oracle complexity) and the proposed notion of reproducibility. The reviewers have reached a clear consensus toward accepting this paper, citing its novelty and technical depth. I concur, and recommend acceptance as a spotlight presentation.

**Award:**

No

---

### Decision · Program_Chairs · 2022-09-14

Accept